# Simulations of Firn Processes over the Greenland and Antarctic Ice Sheets: 1980–2021

Brooke Medley[1], Thomas A. Neumann[1], H. Jay Zwally[1,2], Benjamin E. Smith[3], and C. Max Stevens[1,4]

[1]Cryospheric Sciences Laboratory, NASA Goddard Space Flight Center, Greenbelt, MD, 20771, USA.
[2]University of Maryland – College Park, College Park, MD, 20742, USA.
[3]Applied Physics Laboratory, University of Washington, Seattle, WA, 98105, USA.
[4]Earth System Science Interdisciplinary Center, University of Maryland, College Park, MD, 20740, USA.

*Correspondence to*: Brooke Medley (brooke.c.medley@nasa.gov)

**Abstract.** Conversion of altimetry-derived ice-sheet volume change to mass requires an understanding of the evolution of
the combined ice and air content within the firn column. In the absence of suitable techniques to observe the changes to the firn column across the entirety of an ice sheet, the firn column processes are typically modelled. Here, we present new simulations of firn processes over the Greenland and Antarctic Ice Sheets using the Community Firn Model and atmospheric reanalysis variables for more than four decades. A dataset of more than 250 measured depth-density profiles from both ice sheets provides the basis of the calibration of the dry-snow densification scheme. The resulting scheme results in a reduction
in the rate of densification, relative to a commonly used semi-empirical model, through a decreased dependence on the accumulation rate, a proxy for overburden stress. The 1980–2020 modelled firn column runoff, when combined with atmospheric variables from MERRA-2, generates realistic mean integrated surface mass balance values for the Greenland (+390 Gt yr$^{-1}$) and Antarctic (+2612 Gt yr$^{-1}$) ice sheets when compared to published model-ensemble means. We find that seasonal volume changes associated with firn air content are on average approximately 2.5 times larger than those associated
with mass fluxes from surface processes for the AIS and 1.5 times larger for the GrIS; however, when averaged over multiple years, ice and air-volume fluctuations within the firn column are of comparable magnitudes. Between 1996 and 2019, the Greenland Ice Sheet lost nearly 5% of its firn air content indicating a reduction in the total meltwater retention capability. Nearly all (94%) of the meltwater produced over the Antarctic Ice Sheet is retained within the firn column through infiltration and refreezing.

## 1 Introduction

One of the most robust methods for measuring ice-sheet mass balance uses satellite altimetry (Shepherd et al., 2012, 2018) to measure changes in surface height through time and ultimately provide ice-sheet-wide volume change estimates (Helm et al., 2014; Paolo et al., 2015; Pritchard et al., 2009; Zwally et al., 2005, 2015). Interpretation of volume changes, however, requires ancillary information because there are several processes that generate height changes observable by satellite
altimeters (Ligtenberg et al., 2011; Smith et al., 2020). The measured surface height change is a combination of signals,

which reflect processes that involve ice or solid-earth mass change, or even no mass change at all. Even if we remove the solid-earth processes, partitioning the remaining ice-sheet-volume change to the appropriate material densities remains a challenge. Specifically, volume change due to ice dynamics represents a change at the density of ice (917 kg m$^{-3}$) whereas surface processes (snowfall, sublimation, melt) typically (but not always) represent change under much lower densities (200

kg m$^{-3}$ – 600 kg m$^{-3}$) (Zwally et al., 2015). Additionally, the role of surface processes in observed volume change varies substantially in space and time, yet remains largely unmeasured. Here, we present techniques that use modelling to constrain surface mass balance and firn processes over both the Greenland and Antarctic Ice Sheets (GrIS and AIS, respectively) for improved mass balance studies. Specifically, we provide details on a new approach to densification model calibration, an investigation of relevant spatial and temporal scales, uncertainty quantification, and a model of initial density.

In our modelling, we divide the ice sheets into two vertical layers of different material density, referred to hereinafter as the firn and ice columns. Typically extending tens to over hundred meters down from the surface (Ligtenberg et al., 2011), the firn column represents snow that has fallen, was subsequently buried, and is undergoing densification, yet remains less dense than ice. The rate at which firn compacts varies and is dependent on its age, the weight of snow pressing down on it from above, temperature, and meltwater infiltration and refreezing. The ice column begins at a depth where material density

becomes approximately constant (917 kg m$^{-3}$) and terminates at the bed. In a constant climate, the annually averaged upward vertical velocity of the surface due to snow accumulation is perfectly balanced by ablation, compaction of the firn column, transformation to solid ice, and finally divergence of the underlying ice column (Zwally and Li, 2002), and the thicknesses of the firn and ice columns remain constant. In this scenario, height change is zero.

The firn column is constantly evolving due to a changing climate, across all timescales, and the deviations in snow

accumulation, meltwater production, and temperature from steady-state conditions drive changes in the firn layer thickness. The goal of this work is to simulate these changes in the firn column over the past 40+ years (1980–2021) using a firn densification model and atmospheric reanalysis climate forcing to determine its manifestation in altimetry-derived ice-sheet height change and the subsequent height change correction for mass balance studies.

## 1.1  Ice-Sheet Height and Mass Change

Changes in ice-sheet surface height reflect the integrated signal of several processes, some of which are related to ice or solid-earth mass change and others that reflect no mass change at all. Thus, we must decompose the full signal into various components in order to derive the quantity of interest; here, we are focusing on ice mass change.

Observed height change ($dh/dt$) is defined as:

$$\frac{dh}{dt} = \frac{dh_f}{dt} + \frac{dh_i}{dt} + \frac{dh_e}{dt}, \tag{1}$$

where $f$, $i$, $e$ represent the component of $dh/dt$ resulting from changes in firn processes, solid-ice processes, and solid-earth movement, respectively. Here, $dh/dt$ is the surface height change; however, this value is not synonymous with actual height fluctuations of the full-ice-sheet column change ($dh_I/dt$). Solid-earth uplift or subsidence impacts measured height

changes, yet reflect changes in bedrock elevation in response to current and past ice-mass changes rather than ice-thickness changes alone. This signal must be removed in order to isolate the height change due to combined firn and ice processes, $dh_I/dt$:

$$\frac{dh_I}{dt} = \frac{dh}{dt} - \frac{dh_e}{dt} = \frac{dh_f}{dt} + \frac{dh_i}{dt}.$$  (2)

Height changes that manifest from solid-ice processes ($dh_i/dt$) result from ice dynamical change over grounded ice, but over floating ice, there is an additional component due to sub-ice-shelf melt. These processes are difficult to observe or quantify; thus, we can approximate the solid-ice changes by further reworking Eq. (2) to remove the firn-column height change signal ($dh_f/dt$) from the total ice-sheet column change ($dh_I/dt$):

$$\frac{dh_i}{dt} = \frac{dh_I}{dt} - \frac{dh_f}{dt},$$  (3)

which provides the groundwork for determining ice-sheet mass balance. If the role of firn processes in ice-sheet height change is adequately modeled, we can isolate the contribution due to ice dynamical changes, which are easily converted to mass because the material is assumed to be solid ice (917 kg m$^{-3}$). Ice sheet mass balance estimates remain highly sensitive to small errors in the height change measurements and the modelled firn thickness signal.

Firn column changes, however, have a complicated relationship with mass change. Height changes due to variable rates of compaction of the firn column do not reflect a change in mass but impact the observed ice-sheet height variations through changes in volume and density. Meltwater production is more ambiguous: when it can infiltrate the firn and refreeze, there is no resulting mass change, but when infiltration is impeded and meltwater runs off, there is mass change. The effect of net snow accumulation always reflects a change in mass and can be positive or negative. As a result, the conversion between height, volume, and ultimately mass change requires understanding the material density of each component, which is neither constant in time nor space.

Rather than partition firn column changes by its individual components (see above), we divide total firn-column height change into changes in the thickness of ice and the air thickness: surface mass balance (SMB) and firn air content (FAC), respectively. Specifically, we define $dh_f/dt$ as:

$$\frac{dh_f}{dt} = \frac{dh_{SMB}}{dt} + \frac{dh_{FAC}}{dt},$$  (4)

where $dh_{SMB}/dt$ and $dh_{FAC}/dt$ represent height change fluctuations due to SMB and FAC. These components are defined below. These two components are not independent of one another: snow accumulates at the surface as a mixture of ice and air. We elect to partition firn height change into ice and air components for two reasons: (1) to better support ice-sheet altimetry studies and allow for removal of non-ice-mass change from the observed volume changes and (2) to partially isolate the firn modeling effort presented here from the reanalysis-generated surface mass balance variables used as forcing. Apart from surface runoff, the latter ensures that we take the SMB signal directly from the reanalysis model without

modification, so the focus of the modeling work presented is almost entirely on $dh_{FAC}/dt$; however, we do provide analysis of $dh_{SMB}/dt$ for completeness.

### 1.1.1 Surface Mass Balance

The SMB is the summation of mass fluxes at the surface, including precipitation (solid and liquid), evaporation/sublimation, and runoff (Lenaerts et al., 2019). Here, we do not account for blowing snow processes that likely impact local-scale SMB; however, these processes comprise an overall small percentage of total SMB (Van Wessem et al., 2018). Specifically,

$$SMB = Sn + Ra - Ev - Ru, \tag{5}$$

where *Sn* is snowfall, *Ra* is rainfall, *Ev* is evaporation/sublimation, and *Ru* is runoff. All are in units of m ice-equivalent (i.e.) per year.

### 1.1.2 Firn Air Content

The FAC or depth-integrated porosity represents the integrated volume of air within the entire firn column and is defined as:

$$FAC = \int_0^{z_{\rho_i}} \frac{\rho_i - \rho(z)}{\rho_i} dz, \tag{6}$$

where $\rho_i$ is the density of ice, $z_{\rho_i}$ is the depth in meters at which the density of ice is reached, and $\rho(z)$ is the density at a given depth. The FAC is in units of meters of air.

## 2 Materials and Methods

We simulated firn column processes over both the GrIS and AIS using the Community Firn Model (CFM) framework (Stevens et al., 2020), forced by reanalysis climate variables. These simulations are referred to as GSFC-FDMv1.2.1. First, we provide specifics relating to the CFM as well as our methodology for calibration, spin-up, and implementation. We then describe our selected climate forcing from NASA's Modern-Era Retrospective analysis for Research and Applications, Version 2 (MERRA-2) used in our simulations. Third, we discuss the differences between GSFC-FDMv1.2 and its earlier versions, v1 and v0, the latter of which was used in Smith et al. (2020) and Adusumilli et al. (2020). Finally, we provide details regarding our uncertainty assessment as well as our SMB evaluation approach.

### 2.1 Firn Densification Modeling: GSFC-FDMv1.2.1

#### 2.1.1 The Community Firn Model

The Community Firn Model was built as a resource to the glaciology community, consisting of a modular, open-source framework for Lagrangian modeling of several firn and firn-air related processes (Stevens et al., 2020). The CFM allows the user to select the processes and/or physics of each simulation. The core CFM modules contain physics for firn density and temperature evolution; however, there are several modules for additional processes that the user can implement. For the

GSFC-FDMv1.2.1 simulations, we use modules for grain-size evolution, meltwater percolation and refreezing, and sublimation. Grain-size evolution is simulated for testing purposes and not considered realistic. The user also has several options of firn densification physics from which to choose. Several of the models are calibrated using climate forcing from an RCM, atmospheric reanalysis, or even satellite-derived products, which means that any biases in these climate variables will bias the calibration coefficients in the firn densification model. Thus, it is necessary to have consistent climate forcing between the calibration and actual model runs, so we perform our own densification model calibration (Sect. 2.1.3). Finally, we use a simple bucket scheme for simulating meltwater percolation and refreezing; while the CFM contains a choice of physics of varying complexity, recent work by Verjans et al. (2019) suggests there is currently no evidence that the higher-order models perform better. Here, we use CFM v1.1.6 (Stevens et al., 2020, 2021).

## 2.1.2 Model Spin-up

To ensure that we do not impose any unwanted transients in our simulations, we must have a sufficiently long spin-up interval during which most of the firn column is refreshed. Due to variable snow accumulation rates across the ice sheets, the time required to fully refresh the firn column can vary significantly. Thus, we impose a variable spin-up time that is dependent on the long-term mean climate. Specifically, we use the Herron & Langway (1980) densification model to approximate the depth to the bottom of the firn column (delineated at a density of 910 kg m$^{-3}$) using the long-term reference snow accumulation, temperature, and surface density (see Section 2.1.5). This depth is divided by a burial rate (snowfall – sublimation – melt) to estimate the time needed to refresh the firn column for a given site: this 1$^{st}$ order approximation of the age of the firn near the transition is an overestimate, which ensures that we refresh the entire column. Spin-up intervals typically span 300 to 7,000 years in the Antarctic and 200 to 1,500 years for Greenland. In regions with no net accumulation (snowfall < sublimation + melt), no spin-up is implemented. Rather, the simulations begin with a solid-ice column allowing the model to simulate seasonal snowfall, snowmelt, and runoff.

The CFM has the option to impose a dry-snow spin-up; however, this solution would build a firn column that is in dynamic equilibrium under dry conditions only. If melt were then imposed, meltwater processes would create large negative $dh_f/dt$ and $dh_{FAC}/dt$ that are not realistic. Instead, we only apply a 30-year spin-up to build a dry firn column. We next repeat a baseline reference climate interval (RCI) time series the number of times required to match the estimated spin-up time described above. For example, if a location needs an 800-year spin-up and the RCI is 40 years, the latter is repeated twenty times. If the spin-up time required is not divisible by the RCI interval, we round up to the next integer to exceed the required spin-up time. The CFM is then run using this synthetic time series to generate a firn column that is in dynamic equilibrium with the climate under both dry and wet conditions over the RCI. Because the firn column consists of snow that has fallen years to decades to even centuries ago, its density evolution at present is still responding to atmospheric conditions from the recent and distant past. Thus, an RCI consisting of true atmospheric forcing over these longer timescales would provide the ideal RCI for firn column spin-up; however, these records do not exist over the ice sheets but rather span only a few decades, which means we must make assumptions regarding model spin-up and the RCI. First, we use a baseline RCI for GrIS of

January 1, 1980 – December 31, 1995, which we assume is representative of a longer-term mean climate state. We then assume that the firn column is in equilibrium with the atmospheric conditions over that RCI, which imposes no changes of firn conditions over that interval, and the firn column can evolve freely beginning in 1996. The GrIS underwent a significant increase in temperatures and meltwater production after 1995 (see Sect. 3.2.1). For Antarctica, we define the baseline RCI as January 1, 1980 – December 31, 2019 because there were no appreciable shifts in climate during that time and to remain consistent with prior GSFC-FDM simulations. Like the GrIS, this assumption of RCI allows firn conditions to evolve in time, however, they are constrained by our steady-state assumption in our spin-up which requires no net change in firn conditions over the entire RCI. We discuss the selection of RCI for both ice sheets in the Sect. 3.2 and explore the limitations of the approach in the Sect. 4.

### 2.1.3 Densification Model and Calibration

We use a subset of 256 published firn depth-density profiles from both the GrIS and AIS as the basis of our calibration procedure and perform a single calibration that is representative of both ice sheets. The density-profile dataset is described in Appendix A. The Arthern et al. (2010) dry-snow densification model provides the physical basis for our GSFC-FDMv1.2.1 simulations. Specifically, modelled dry-snow densification rates are separated into two stages during which the parcels experience different compaction processes and that are defined by the density of the parcel:

$$\frac{d\rho}{dt} = c_0(\rho_i - \rho); \ \rho \leq 550 \ \frac{kg}{m^3}, \tag{7}$$

$$\frac{d\rho}{dt} = c_1(\rho_i - \rho); \ \rho > 550 \ \frac{kg}{m^3}, \tag{8}$$

where the densification rate coefficients for stage 1 and stage 2 ($c_0$ and $c_1$) are defined as a function of the cumulative accumulation above a given firn parcel ($\dot{b}$: defined as the mean accumulation rate in ice equivalence, m.i.e. yr$^{-1}$, experienced since that parcel was deposited), the temperature of the parcel in Kelvin, $T$, and the mean annual temperature, $\bar{T}$:

$$c_0 = 0.07 \ \dot{b}^{\alpha_0} \ g \ exp\left(\frac{-E_{c_0}}{RT} + \frac{E_g}{R\bar{T}}\right), \tag{9}$$

$$c_1 = 0.03 \ \dot{b}^{\alpha_1} \ g \ exp\left(\frac{-E_{c_1}}{RT} + \frac{E_g}{R\bar{T}}\right), \tag{10}$$

where $g$ is the gravitational acceleration constant (9.8 m s$^{-2}$), the activation energy for lattice diffusion commonly used for ice is $E_{c_0} = E_{c_1} = 60$ kJ mol$^{-1}$ (Cuffey and Paterson, 2010), $E_g$ is the activation energy for grain growth (42.4 kJ mol$^{-1}$), $R$ is the gas constant (8.314 J K$^{-1}$ mol$^{-1}$), and the exponential dependence of overburden is $\alpha_0 = \alpha_1 = 1$ (Arthern et al., 2010). Thus, the dry densification rate experienced by a given firn parcel varies in time and is based on $\rho$, $\dot{b}$, and $T$ within a single stage of densification.

To begin the calibration procedure, we first run the model in its original form at 226 calibration sites across Greenland and Antarctica (Figure 1). The number of model calibration runs is less than the actual number of observations (256) as some fall within the same grid cell (e.g., several observations from the vicinity of Summit, Greenland). All 256 observations are

used. Unlike other calibration efforts (Kuipers Munneke et al., 2015; Li and Zwally, 2004, 2011; Ligtenberg et al., 2011), the calibration procedure presented here treats dry-firn densification from both ice sheets together, forming a single calibration parameterization, which benefits from a much wider range of climate conditions than if each ice sheet was treated individually.

The logarithm of the firn density profile with depth is approximately linear, largely for stage 2 (Herron and Langway, 1980). More discussion on the use of a logarithmic density profile is in Appendix B. For each calibration site, we compare the slopes of the logarithmic density versus depth for the two stages of densification between observations $(C_0^O, C_1^O)$ and the equivalent model output $(C_0^M, C_1^M)$ using the original Arthern et al. (2010) model configuration (Eqns. 7–10) forced by the RCI. Both the density measurements and model output are binned into half-meter depth increments to obtain similar sampling intervals. After binning, the slopes are estimated. Because density measurements are noisy, we determine the slopes in an iterative fashion, removing individual density measurements with residuals to the linear model larger than 3-sigma, recalculating the linear model, and repeating until all residuals are less than 3-sigma (i.e., an iterative 3-sigma edit). Calibration sites were not used in a given stage if they either (1) did not contain more than 7 data points for that stage prior to the 3-sigma edit, (2) did not span more than 5 meters in depth, (3) the final linear model produced a slope that was not significant ($p > 0.01$), or (4) encountered significant melt (mean annual surface melt exceeds 1% of the mean annual snow accumulation). The latter ensures that we are only calibrating to dry-snow densification. Our final calibration dataset contains 141 depth-density profiles spanning stage 1 and 76 spanning stage 2. Note, there are fewer profiles for stage 2 because not every density profile extends to stage-2 densities. There are a limited number of sites used in the calibration from Greenland because most of them cannot fully reflect dry snow conditions (i.e., they do not meet the aforementioned melt criterion) (Figure 1). The ratios $(R_0, R_1)$ of the observed slopes $(C_0^O, C_1^O)$ to the modeled slopes $(C_0^M, C_1^M)$ provide the necessary correction (or calibration coefficient) for each site as described below.

Rather than develop a new physical form for calibration, we optimize two parameters within the Arthern et al. (2010) model: the exponential dependence on the mean annual accumulation rate since the parcel was deposited and the activation energy for creep. Arthern et al. (2010) found evidence that the activation energy is not well constrained for the sites investigated, suggesting that the physical processes at play under various conditions are not fully understood. Similarly, Ligtenberg et al. (2011) and Kuipers Munneke et al. (2015), found that the Arthern et al. (2010) model required additional dependence on snow accumulation to best fit observations. Thus, we elect to calibrate the parameters relating to variations in snow accumulation $(\alpha_0, \alpha_1)$ and temperature $(E_{c_0}, E_{c_1})$ for each stage of densification. This choice of calibration parameters is also important because the climate forcing contains unknown biases, which can be partially overcome through calibration. Thus, the calibrated model presented here (along with all others) is only relevant when used with the same climate forcing (see Sect. 2.2).

We define our calibration coefficients for the two stages of densification $(R_0, R_1)$ as a function of the mean accumulation rate $(\bar{b})$ and temperature $(\bar{T})$:

$$R_0 = \frac{C_0^O}{C_0^M} = \bar{b}^{\beta_0} \, exp\left(\frac{-E_0}{R\overline{T_0}}\right), \tag{11}$$

$$R_1 = \frac{C_1^O}{C_1^M} = \bar{b}^{\beta_1} \, exp\left(\frac{-E_1}{R\overline{T_1}}\right). \tag{12}$$

We solve for $\beta$ and $E$ using a least-squares fit regression model (intercept = 0) with the climate forcing, $\bar{b}$ and $\bar{T}$, as predictor variables and our calibration coefficient, $R$, as the response variable. We force the intercept to zero to minimize overdetermination and allow the changes in the Arthern et al. (2010) functional form to be linked to a physical control (e.g., overburden, temperature) rather than a bulk bias shift. We first linearize Eq. (11) and (12):

$$ln(R_0) = \beta_0 ln(\bar{b}) - E_0 \left(\frac{1}{R\overline{T_0}}\right), \tag{13}$$

$$ln(R_1) = \beta_1 ln(\bar{b}) - E_1 \left(\frac{1}{R\overline{T_1}}\right). \tag{14}$$

To generate $\bar{b}$, we calculate the mean accumulation rate for each parcel since its deposition and take the average for each stage of densification. For stage 2, the firn column is effectively isothermal, so we substitute in Eq. (14) the mean temperature of all firn parcels with a density greater than 550 kg m⁻³, $\overline{T_1}$. Parcels undergoing stage 1 densification incur much larger fluctuations in temperature, especially near the surface. Prior versions of GSFC-FDM used a mean effective temperature within stage 1 to capture the non-linear relationship between temperature and compaction rates; however, that practice was abandoned after further evaluation against daily simulations suggest more refinement to the CFM is required to use an effective temperature (Appendix C). Thus, as is done for stage 2, we use the mean temperature of all firn parcels with a density less than 550 kg m⁻³, $\overline{T_0}$, in Eq. (13) to calibrate stage 1 densification.

We finally iteratively solve for $\beta_0$, $\beta_1$, $E_0$, and $E_1$; however, only a single iteration was sufficient for both stages. To determine the uncertainties in our parameterization, we use the Monte Carlo method to explore the impact of uncertainties in the predictors. Specifically, we perform $n = 10,000$ least-squares-fit regression models (intercept = 0) using randomly perturbed predictors and predictands. The uncertainty in the modelled predictors ($\overline{T_0}$, $\overline{T_1}$, $\bar{b}$, $C_0^M$, $C_1^M$) are derived from their variability in time (i.e., are randomly sampled in time), and the uncertainty in the observed predictands ($C_0^O$, $C_1^O$) are derived from the uncertainty in the logarithmic linear fit, which represents a Gaussian spread. Our final parameters are the mean of all 10,000 regression models, and their uncertainties are equal to the 2-sigma deviations. We find the optimal parameters for Eq. (11) and (12) are:

$$\beta_0 = -0.09 \pm 0.03, \ \ \beta_1 = -0.356 \pm 0.017, \ \ E_0 = -500 \pm 300 \, J \, mol^{-1}, \ \ E_1 = -3130 \pm 100 \, J \, mol^{-1}. \tag{15}$$

These calibrated parameters, when plugged into Eqns. 11–12, provide the calibration coefficients for the two stages of densification, which scale the densification rate provided by the original Arthern et al. (2010) rate model (Figure 2). The calibration largely finds reduced rates of densification during stage 1, especially at higher accumulation rates. For stage 2, the modelled compaction at the coldest and driest sites will increase, while compaction at sites experiencing moderate to high accumulation rates (> ~100 mm i.e. yr⁻¹) will decrease, largely as a function of the accumulation rate.

Combining Eq. 9–12, 15, we define our densification rate coefficients as:

$$c_0 = R_0 \, 0.07 \, \dot{b} \, g \, exp\left(\frac{-60000}{RT} + \frac{42400}{R\bar{T}}\right), \tag{16}$$

$$c_1 = R_1 0.03 \, \dot{b} \, g \, exp\left(\frac{-60000}{RT} + \frac{42400}{R\bar{T}}\right), \tag{17}$$

which requires certain assumptions. Specifically, we assume that and $\bar{b} \approx \dot{b}$ and $\bar{T}_0 \approx \bar{T}_1 \approx T$. Concerning the former, because the CFM defines $\dot{b}$ as the mean accumulation rate after deposition of a parcel, $\dot{b}$ approaches $\bar{b}$ with depth. Near the surface, $\dot{b}$ of a parcel can differ from $\bar{b}$, however, the assumption is that integrated across all parcels, the deviation is negligible. The same is true for $\bar{T}_0$ and $\bar{T}_1$: the firn pack reaches thermal equilibrium with depth, so the temperature of a parcel will deviate from the mean closer to the surface, but with increasing depth, $T$ approaches $\bar{T}_1$. While these assumptions

are valid for deeper firn, they are practical simplifications within the upper part where deviations in the integrated accumulation rate and temperature from the mean exist. The assumption is that in a column integrated sense, the impact is minimized. Therefore, Eq. 11–12, 15, 16–17 and the aforementioned assumptions produced newly calibrated parameters for use with Eq. 9–10:

$$\alpha_0 = 0.91, \quad \alpha_1 = 0.644, \quad E_{c_0} = 59500 \, J \, mol^{-1}, \quad E_{c_1} = 56870 \, J \, mol^{-1}. \tag{18}$$

The dry compaction model used in the GSFC-FDMv1.2.1 simulations presented here is summarized by Eq. 7–10 and 18. We note that the new parameters in Eq. 18 are similar to those developed by Verjans et al. (2020a) despite substantial differences in the techniques used to complete the calibration. Model performance is discussed in Sect. 2.4.

### 2.1.4 Spatial Domain

For Greenland, we define the ice boundary using the Greenland Mapping Project (GIMP) ice mask posted at 90-meter spatial

resolution (Howat et al., 2014). We identified approximately 13,200 of the 12.5 km GSFC-FDMv1.2.1 pixels as ice if any of the GIMP pixels that fell within were flagged as ice. For integrated SMB determination, we scale each pixel by the area of ice within based on the GIMP ice mask; the total ice sheet area along with the peripheral ice not connected to the main ice sheet is $1.78 \times 10^6 \, km^2$. The grid cells with positive net accumulation (i.e., snowfall – sublimation – meltwater > 0), a condition required to build a firn column, amount to just over 9,000. About 4,100 grid cells did not meet the requirements to

sustain a firn column, and their seasonal snowfall, melt, and runoff are simulated as described in Sect. 2.1.2.

For Antarctica, we use the drainage basins at 1-kilometer resolution defined by Zwally et al. (2012). We identified any of its 12.5 km pixels that contain an ice-flagged pixel from Zwally et al. (2012) as ice, resulting in just over 88,300 ice-covered pixels. We assume all the pixels are 100% ice-covered, which is equivalent in area to $13.6 \times 10^6 \, km^2$ (grounded ice sheet area: $12.1 \times 10^6 \, km^2$). Most meet the positive net accumulation condition to sustain a firn column (87,800). To improve

efficiency, we do not simulate firn column processes for each grid cell. Rather, we investigate the similarities in atmospheric forcing between neighboring pixels to eliminate redundant simulations. If a cell has a neighbor where its: (1) mean annual temperature is consistent within 0.75 K, (2) the root mean square difference (RMSD) in snowfall-minus-sublimation is less

than 10% of the mean annual snowfall-minus-sublimation, (3) the RMSD in skin temperature is less than 0.25 K, and (4) the RMSD in meltwater production is less than 5% of the mean annual meltwater production, then we do not run a simulation for that grid cell. These selection criteria reduce the number of simulations to 38,200. With these criteria, the fine spatial resolution is preserved in coastal regions where climate gradients are strong and is coarsened in the interior where correlation length scales are quite large (Figure 3). Once the subset of simulations was complete, we linearly interpolated the runoff and FAC time series to fill the ~88,300 ice-covered cells.

### 2.1.5 Initial (Surface) Density

Because of the low accumulation rates over the ice sheets and the coarse (5 days) time resolution of our simulations, we anticipate significant reworking of the initial, low-density surface snowpack. Ideally, the imposed initial density would vary in time based on the ambient climate conditions; however, there are few studies that focus on the temporal evolution of freshly fallen snow over the ice sheets (e.g., Groot Zwaaftink et al., 2013). Thus, we focus rather on improving the bulk (or time-invariant) initial density for each grid cell based on the mean annual climate conditions as done by Helsen et al. (2008) and Kuipers Munneke et al. (2015). This approach means that, on average, we will approximate the surface density well, but we accept that there might be significant deviations from this bulk density over shorter timescales.

To build a model of initial density ($\rho_0$), we estimate initial densities from 233 depth-density profiles (stage 1) by finding the surface-intercept of a linear fit to the logarithm of density versus depth (Figure 1). These represent the best-fit of the initial density to the observed density profile and include sites that are both dry and wet, resulting in a larger number of useable stage 1 profiles than for the dry-snow densification calibration (Sect. 2.1.3; $n = 141$). We then trained a Gaussian Process Regression model to predict the observed initial densities using the mean annual MERRA-2 surface climate (snow accumulation, air temperature, total wind speed, and specific humidity) of which air temperature had the largest impact on prediction. The 233 initial densities were split into a training ($n = 187$) and testing ($n = 46$) partition, the latter of which provides an assessment of model performance. The model results are shown in Figure 4: while we capture nearly 50% of the variability within the testing partition ($r^2 = 0.46$), predicted densities remain too high at the lowest densities. Specifically, the *RMSE* of all observations (training and testing; n = 233) is 16.6 kg m$^{-3}$ whereas the *RMSE* for observed initial densities less than 330 kg m$^{-3}$ is 30.9 kg m$^{-3}$ (n = 20). The $\rho_0$ used in GSFC-FDMv1.2.1 are displayed in Figure 1. The upper and lower 5% of the initial densities span 327–387 kg m$^{-3}$ for the GrIS and 350–417 kg m$^{-3}$ for the AIS with respective median values of 369 and 382 kg m$^{-3}$. For Greenland, we find higher densities around the periphery, which is in line with other studies (Machguth et al., 2016a; Fausto et al., 2018; Kuipers Munneke et al., 2015); however, the lower end of our distribution is biased high, which will have implications on the modeled firn air content.

### 2.2 MERRA-2

MERRA-2 is a global atmospheric reanalysis developed at the Global Modelling and Assimilation Office (GMAO) at NASA Goddard Space Flight Center (Gelaro et al., 2017). Atmospheric variables are provided at 0.625° longitude x 0.5° latitude

resolution and span the satellite era (1980–present). Here, we use the MERRA-2 snowfall, total precipitation, evaporation, 2-meter air temperature, and skin temperature at hourly resolution and land ice runoff flux at monthly resolution covering January 1, 1980 to September 30, 2021 (GMAO, 2015a,b,c). At the ±70° latitude bands, the model has a resolution of 24 km x 56 km, which is too coarse to resolve steep coastal topography such as the Antarctic Peninsula or the GrIS ablation zone. Thus, we rely on offline, 12.5 km 'replay' MERRA-2 runs over both the GrIS and AIS to improve representation of regions

of steeply sloping topography.

  MERRA-2 employs the Incremental Analysis Update (IAU) scheme of Bloom et al. (1996). The IAU uses predictor and corrector model forward integrations where differences with observations are first computed in the predictor segment, and then added as an additional forcing term in the corrector run. It may be noted that an entirely different global model may employ the IAU scheme to correct to MERRA-2 innovation variables every 6 hr, a process referred to as "replay" (e.g.,

Mapes and Bacmeister, 2012). The MERRA-2 12-km replay integration (M2R12K) was produced as part of the NASA Downscaling Project (Tian et al., 2017), and covers the period December 1999 to November 2015. A non-hydrostatic version of the Goddard Earth Observing System (GEOS) model was used in the replay integration with an output grid spacing of 1/8 degree by 1/8 degree, but with the same vertical resolution as the original MERRA-2. The atmospheric model was modified to repartition large-scale and convective processes, and the analysis increment was filtered to allow for

features of a higher resolution than resolved in the original MERRA-2 analysis grid.

  The high resolution M2R12K only spans fifteen years, so it cannot be used as direct forcing of the firn densification model. Rather, we retain the seasonal magnitudes in the atmospheric variables from the M2R12K to provide hybridized MERRA-2 output. First, the MERRA-2 output is oversampled to the M2R12K grid. We then determine the 2000–2014 monthly means in MERRA-2 and remove them from the full MERRA-2 record (1980–2021). The 2000–2014 M2R12K monthly means are

330 then added to the MERRA-2 residuals to form the hybridized MERRA-2 atmospheric variables. In such a manner, the magnitude of the gradients in precipitation and temperature from the high resolution M2R12K are transferred to the coarse MERRA-2 output. Figure 5 and Figure 6 show the mean annual net accumulation (snowfall-minus-sublimation) and skin temperature, respectively, for the GrIS and AIS. For simplicity, we hereinafter refer to the hybridized MERRA-2 as MERRA-2.

While the variables are provided at hourly resolution, to maximize computational efficiency, we perform the firn simulations at a resolution of five days. The 5-day MERRA-2 time series are built by averaging the hourly data over 5-day intervals. Although MERRA-2 includes meltwater processes, only net runoff is retained. Thus, we use a degree-day approach to build gridded meltwater time series, which is described in Sect. 2.2.1.

### 2.2.1 Degree-day Model

For both ice sheets, we used a simple model to generate meltwater fluxes for input into the CFM. Specifically, meltwater production ($m$) was estimated using a calibrated degree-day model (e.g., van den Broeke et al., 2010):

$$m = DDF \times \sum_t (T_{2m} - T_0)\Delta t \; ; T_{2m} > T_0. \tag{19}$$

Melt was activated when the 2-meter air temperature ($T_{2m}$) exceeded a calibrated temperature threshold ($T_0$); the exceedance is then scaled by the calibrated degree-day factor ($DDF$: $\frac{kg}{m^2 hrK}$) to generate the magnitude of melt. Here, we used hourly temperatures ($\Delta t = 1 \, hour$) to estimate five-day ($t = 5 \, days$) meltwater production. While degree-day models traditionally use $\Delta t = 1 \, day$, we used a finer temporal resolution to ensure more realistic meltwater production, but ultimately, melt was accumulated over a five-day window.

We calibrated our melt model for Antarctica using a calibration data set of surface meltwater fluxes (Trusel et al., 2013a) that span the 1999 to 2009 melt seasons, which are linearly interpolated to our 12.5 km grid. Rather than calibrate our model to 5-day meltwater fluxes, we optimized correspondence of annual meltwater production between the model and calibration data and set $t = 1 \, year$. For each grid cell, we quantified the $DDF$ that best relates the annual accumulated exceedance of $T_{2m}$ over a predetermined threshold, $T_0$, which does not vary in space. To evaluate which temperature threshold yielded the best model, we calculated $DDF$ under a wide range of $T_0$ (265–273 K) at quarter-degree intervals. To eliminate unrealistic $DDF$, we set all $DDF$ in the upper 1% to the 99[th] percentile factor. We evaluated the performance of these models in reproducing the annual time series of Antarctic-wide meltwater production as compared to our calibration data set (Trusel et al., 2013a). Specifically, we compared on a grid-cell basis their ability to reproduce interannual variability ($r^2$) and to minimize mismatch ($RMSE$) with the observations. Giving equal weight to the aforementioned, we found the ice-sheet-wide mean $r^2$ and $RMSE$ for each threshold and selected the $T_0$ that maximizes the normalized distance between the curves of the two evaluators ($r^2$ and $RMSE$) in Figure 7a. This approach selected a threshold that lies in between the threshold if determined by one evaluator alone. For Antarctica, we used $T_0 = 270.25 \, K$.

We estimated a temperature threshold over the GrIS using a similar approach. While we used an observation-based calibration data set over Antarctica, a similar data set does not exist for Greenland, so we instead used independent model output as the basis of our calibration. Specifically, we used the 1980–2014 annual meltwater rates from the MARv3.5.2 regional climate model (RCM) (Fettweis et al., 2017). Although this product provides sub-annual resolution, we opted to calibrate to annual meltwater production once more. In such a manner, the short timescale meltwater fluxes were driven by MERRA-2, but the calibration to annual RCM output ensured that the simple model remains aligned with realistic annual magnitudes from MAR. For Greenland, we found a threshold, identical to Antarctic, of $T_0 = 270.25 \, K$ (Figure 7b).

For both ice sheets, the temperature threshold is below freezing, which suggests either (1) a cold bias in MERRA-2 or (2) too strong melt within the calibration data sets. The former has been found over Greenland (Hearty III et al., 2018) and Antarctica (Gossart et al., 2019; Huai et al., 2019) for summer months, but we cannot eliminate the latter as a contributor to the sub-freezing threshold as well, which we discuss more in Sect. 4. We assess the realism of the calibrated GrIS $DDF$ by plotting the mean values over 250 m elevation bins. Moving into the interior, we would expect lower $DDF$s as the surface is typically bright snow, whereas lower elevations are more likely to exhibit bare ice and lower albedos, which would yield

higher $DDF$s. For Greenland, the relationship between elevation and $DDF$ exhibits high values at lower elevations, which drop off to a near stable value around 1500 m, above which the values rapidly increase (Figure 8). We assume that the stable values around 0.13 kg m$^{-2}$ hr$^{-1}$ K$^{-1}$ are likely more representative of expected values moving upward into the dry snow zone than the values obtained by allowing the $DDF$s to unrealistically rise. Above 1500 m, $DDF$s are capped at 0.13 kg m$^{-2}$ hr$^{-1}$ K$^{-1}$ while calibrated values below that cap are untouched. The lower and upper 5% $DDF$ bounds over the GrIS are 0.06 and 0.21 kg m$^{-2}$ hr$^{-1}$ K$^{-1}$. For Antarctica, we cannot take a similar approach as its geometry and the presence of large floating ice shelves complicate the interpretation of the relationship between the $DDF$ and elevation. Thus, since majority of melting occurs over the ice shelves, we use typical values from the ice shelves to limit $DDF$ over the higher elevations of the grounded ice sheet. Specifically, we found the 95% bounds of $DDF$s over the ice shelves. If a $DDF$ is less than the lower bound, we set it to zero, and if it is larger than the upper bound, we cap it at that upper bound. The lower and upper bounds for $T_0 = 270.25\ K$ are 0.01 and 0.18 kg m$^{-2}$ hr$^{-1}$ K$^{-1}$ with a mean of 0.06 kg m$^{-2}$ hr$^{-1}$ K$^{-1}$. After capping the lower and upper bounds, the range of $DDF$s remains effectively the same: 0.01–0.18 kg m$^{-2}$ hr$^{-1}$ K$^{-1}$ (95$^{th}$ percentile), but the mean goes up to 0.09 kg m$^{-2}$ hr$^{-1}$ K$^{-1}$. These modified $DDF$s are then used to generate 5-day meltwater production using Eq. (19) with a temperature threshold of $T_0 = 270.25\ K$ (Figure 9). The calibrated $DDF$s for Antarctica are typically smaller than those from Greenland, which is logical given that a significant portion of GrIS has exposed bare ice during the summer months enhancing meltwater production. The lower bound was capped for Antarctica to exclude unrealistically low melt factors; however, follow-on analysis should involve studying the impact of this assumption on the final meltwater fluxes. The meltwater production model implemented is a source of substantial uncertainty within our results; development of a surface energy balance scheme within the CFM is underway and will provide a more robust representation of meltwater fluxes in the future.

## 2.3 Improvement from GSFC-FDMv0 and v1

The results presented here build off prior simulations, GSFC-FDMv0 and v1, detailed in a previous publication (Smith et al., 2020; Medley et al., 2020). We have since incorporated major improvements to the GSFC-FDMv1.2.1, which we outline below. Version 1.2 is obsolete as there was a bug in the CFM that excluded time steps with net sublimation. The CFM bug was fixed for v1.2.1 runs. GSFC-FDMv1.2.1 includes:

1. a spatially variable initial density ($\rho_0$; see Sect. 2.1.5), whereas v0 used a constant 350 kg m$^{-3}$. While v1 also used a spatially varying initial density, the formulation was not physically realistic as it only used northward wind and not eastward winds as predictors. For v1.2.1, we also include more observations, even those where wet firn processes occur, whereas in v0 and v1, the observations used were limited to largely dry-firn conditions, which generated a larger mismatch in modeled initial densities around the periphery of the GrIS with observations.

2.  calibration of the dry snow/firn compaction model that limits the inclusion of observations based on the ratio of mean annual meltwater production to snowfall (see Sect. 2.1.3). The calibration approach for v0 did not discard observations based on their exposure to liquid water processes. This change in v1 and v1.2.1 should lead to an improvement in the representation of dry compaction but we note that this calibrated dry snow/firn compaction model is still used in regions of meltwater percolation;

3.  a more robust approach to handling mass fluxes at the surface. The CFM underwent a significant update between v0 and v1, including allowing the explicit removal of mass via sublimation and also inclusion of rainfall. For v0, sublimation was handled by aggregating the accumulation from neighboring time steps until positive thereby still accounting for sublimation but at the cost of smoothing out the accumulation signal. Rainfall was not included in v0. For v1 and v1.2.1, mass via rainfall can now be added to the total liquid

volume present and become subject to liquid water processes;

4.  an improved meltwater model. The degree-day approach for both v0 and v1 are the same; however, the v0 model was built using skin temperature, which cannot exceed 273.15 K and will not capture the large temperature deviations above freezing, especially in Greenland. For v1, we use 2-meter air temperature (see Sect. 2.2.1), which is a more robust approach; however, extreme $DDF$ values, largely in the interior, resulted in

unrealistic melt rates. Thus, for v1.2.1, we capped $DDF$s based on realistic dry-snow values, which should improve meltwater fluxes in the nearly dry interior of the GrIS;

5.  runoff as an output. The older CFM version used for v0 did allow for melt, percolation, and refreezing, but did not provide runoff as an output. Thus, we are now able to calculate surface mass balance using v1 and v1.2.1;

6.  an uncertainty analysis of the dry snow calibration coefficients, which was not completed in v0 or v1. This

exercise provides part of the basis for estimating total uncertainty in FAC and its evolution in time as well as total height and volume change;

7.  a time resolution of 5 days for both the GrIS and AIS. The prior versions (v0 and v1) ran subsets of the AIS at 5, 10, and 20 days, depending on their mean climate. Within v1.2.1, the entire AIS is run at 5-day resolution.

**2.4 Model Performance**

To evaluate the model improvement through our calibration procedure (Sect. 2.1.3), we evaluate the uncalibrated and calibrated model abilities to capture the slopes of the logarithmic density versus depth for both stages against the calibration data set. We found that the mean absolute error (MAE) in modeled slopes for both stages was reduced by nearly one half after calibration, and the explained variance between observed and modeled was significantly increased in stage 2 (Figure 10). The mean observed slope is 0.067 m$^{-1}$ for stage 1 and 0.030 m$^{-1}$ for stage 2. After calibration, the MAE in modeled

slope reduces from 0.021 m$^{-1}$ to 0.013 m$^{-1}$ for stage 1 and from 0.009 m$^{-1}$ to 0.005 m$^{-1}$ for stage 2 (Figure 10). The calibration relies heavily on modification to the accumulation rate (i.e., overburden) component of densification for both stages. Modification to the temperature dependence is necessary for stage 2 and of very minor importance for stage 1. For

both stages, the sensitivity of densification rates to increasing accumulation is reduced, although the changes are more pronounced for stage 2. Densification due to temperature fluctuations during stage 2 is increased, especially at colder temperatures. Ligtenberg et al. (2011) and Kuipers Munneke et al. (2015) similarly found that the semi-empirical Arthern et al. (2010) model mostly overestimated the rate of densification and found an empirical link with the accumulation rate.

We would ideally prefer to perform an evaluation of modeled firn densification rates, but a substantial number of published observations is lacking. Here, we further evaluate the ability of GSFC-FDMv1.2 to reproduce the observed densities in our full data set of sites that are in both dry and wet conditions. Most of these observations were used in the calibration; however, those with significant melt were excluded (see Sect. 2.1.3). Thus, we break out our evaluation into sites exhibiting zero, moderate, and high melt rates, quantified by their ratio to net snowfall and there are at least two observations within a stage. Specifically, these are respectively defined as 0%, less than 10%, and more than 10% of the mean annual snowfall, and we evaluate the modelled mean absolute error in reproducing depth-density observations (Figure 11). The error increases with larger melt fractions, especially for stage 1 where the impact of melt is stronger. Because most of these observations are included in the calibration, we report them as interquartile ranges and assume the upper bounds are more representative of a realistic error for each group. For stage 1, we expect density errors of 15.2 to 29.9 kg m$^{-3}$ for dry snow/firn, 26.0 to 41.3 kg m$^{-3}$ for moderate melt fractions, and 49.4 to 72.5 kg m$^{-3}$ for high melt fractions. For stage 2, we expect density errors of 10.7 to 25.9 kg m$^{-3}$ for dry snow/firn, 18.1 to 38.2 kg m$^{-3}$ for moderate melt fractions, and 32.5 to 78.5 kg m$^{-3}$ for high melt fractions. We note here that we assume that each observation was taken on January 1, 1980 for comparison with the model, which likely introduces additional error.

If we evaluate the bias in our model-derived density profiles for each stage, we find that with increasing melt, the modeled profiles exhibit a more positive bias ($bias = model - observation$). Specifically, the median stage 1 bias under melt scenarios of 0%, less than 10%, and more than 10% of the mean annual snowfall are 7.4 kg m$^{-3}$, 31.4 kg m$^{-3}$, and 61.5 kg m$^{-3}$, respectively. The respective biases for stage 2 are 11.2 kg m$^{-3}$, 23.1 kg m$^{-3}$, and 40.3 kg m$^{-3}$. These biases suggest that the model likely underestimates the FAC i.e., overestimates the density in regions of strong melt.

## 2.5 Uncertainty Analysis

### 2.5.1 Firn Air Content, Surface Mass Balance, and Height Change

We estimated the uncertainty in the total FAC and its variability through time through ensemble perturbation runs of the CFM at select locations over each ice sheet. Specifically, we completed principal component analysis (PCA) on the 5-day climate time series of variables of critical importance to our simulations: SMB and temperature. We then found the principal components that account for 95% of the variability for both SMB and temperature. This selection yielded 41 PCs for SMB and 4 for temperature for AIS and 14 and 4 for GrIS, respectively. We then correlated each individual PC time series with the equivalent time series at every grid cell over the respective ice sheet. The grid cell with the largest correlation with the PC was selected as a perturbation site. As such, we had 45 sites for AIS and 18 sites for GrIS. We used these locations

because they are the most representative of the forcing time series across the entire ice sheet. PCA analysis of melt was not performed because it is determined by the temperature (Sect. 2.2.1).

For each of the calibration sites, we ran the CFM 100 times, each time applying 11 perturbations to the climate forcing variables, CFM parameters, and the reference climate interval. Each of the perturbations sampled from a Gaussian distribution with a mean of zero and a standard deviation based on observations or model performance with specifics found

in the 2nd column of Table 1. The choice of reference climate interval and the parameterization for the thermal conductivity of ice were exceptions: we assumed a uniform distribution of each of the various scenarios in Table 1, which details each perturbation, their sampling window, and any references. For each of the 100 perturbations, we sampled each of the aforementioned Gaussian distribution of uncertainties for the modelled initial density ($\rho_0$) and the calibration parameters ($\alpha_0, \alpha_1, E_{c_0}, E_{c_1}$). We also sampled approximated Gaussian uncertainties in the mean snow accumulation rate ($Sn - Ev$),

Rainfall ($Ra$), Melt ($Me$), and skin temperature, and then scaled the original MERRA-2 time series to modify the climate forcing used. Finally, we selected our choice of the parameterization of the thermal conductivity of firn from 7 different models within the CFM and our choice of the end of the reference climate interval assuming a uniform distribution. Each calibration-site perturbations were sampled independently of others resulting in 4500 and 1800 unique CFM runs for the AIS and GrIS. We note that this is a simplified approach into uncertainties in firn column evolution considering atmospheric

variables as well as CFM parameters are undoubtedly correlated, which we do not consider but could be implemented in future model versions with some modification.

We assessed uncertainties by taking the standard deviation of the mean FAC for each of the 100 perturbations over the entire time series for a given site. We next used mean annual climate parameters (snow accumulation, rain, melt, and temperature) for each site (the original, non-perturbed MERRA-2 mean values) to predict the standard deviations in FAC. We broke the

490 regression into two groups based on the ratio of the mean annual liquid water content (melt + rain) divided by the mean annual snowfall. This ratio is defined as the liquid-to-solid ratio (LSR). We created two uncertainty models in FAC: one for LSR < 0 or LSR ≥ 1 (n = 55) and another for 0 ≤ LSR < 1 (n = 8). The latter approximates the existence of a firn column where the mass of snowfall received outweighs the combination of meltwater production and rainfall. The former is indicative of conditions that are not suitable for firn development: a negative LSR suggests net sublimation (i.e., no solid

accumulation) and an LSR greater than 1 reflects conditions where liquid processes outweigh the solid limiting formation of a firn column. Therefore, locations with LSR < 0 or LSR ≥ 1 conditions experience only transient snow or firn pack, so we estimated the uncertainty in the mean FAC by simply taking the standard deviation of the FAC time series. Combining results from both AIS and GrIS where 0 ≤ LSR < 1, we developed a linear regression model to approximate uncertainty in the mean annual FAC. We found that mean snow accumulation, $\bar{b}$, and skin temperature, $\bar{T}$, provide a robust prediction

(Figure 12) of the 1-sigma uncertainties in mean FAC, $\sigma_{\overline{FAC}}$:

$$\sigma_{\overline{FAC}} = 15.1 + 0.78\,\bar{b} - 0.055\,\bar{T}, \qquad 0 \le LSR < 1 \tag{20}$$
$$\sigma_{\overline{FAC}} = \sigma_{FAC}, \quad LSR < 0 \; or \; LSR \ge 1 \tag{21}$$

Using Eqs. 20–21, we estimated the 2-sigma uncertainty in the mean FAC ($2\sigma_{\overline{FAC}}$) for the GrIS and AIS, which yielded typical values ranging from 0.2 to 3.9 m for the GrIS and from 2.8 to 6.4 m for the AIS (lower and upper 5% bounds). Colder temperatures and higher accumulation rates produce larger uncertainties in FAC. Melt, rainfall, and the LSR were not significant predictors of $\sigma_{\overline{FAC}}$ where $0 \leq LSR < 1$, so they were excluded from the prediction.

To quantify the uncertainty in FAC variability through time, we used the same set of perturbations and estimated the standard deviation in FAC change for each of the 100 perturbation runs over every 5-day time step, producing a time series of standard deviations. We then scaled the standard deviation in 5-day FAC change by dividing them by absolute value of the mean 5-day FAC change, yielding a time series of standard deviations relative to the absolute value of the mean FAC change. Finally, we calculated the median scaled standard deviation over the entire time series to approximate the typical uncertainties in FAC change, which was done for each of the perturbation sites. We were unable to quantify a relationship between the relative error in FAC change and the mean climate forcing even when separating between sites that experience melt and those that do not. Rather, the relative uncertainty in 5-day FAC change ($\sigma_{dh_{FAC}/dt}$) did not largely change between sites, so we use the mean relative error for all sites:

$$\sigma_{dh_{FAC}/dt} = 0.134|dh_{FAC}/dt|, \tag{21}$$

where $dh_{FAC}/dt$ is the firn thickness change due to changes in FAC in units of meters per unit time. We use the results of our SMB evaluation to assess the uncertainty in total thickness change due to SMB (Sect. 3.4). We found the median absolute bias when comparing our mean annual SMB to a series of observations for each ice sheet (Sect. 2.5.2). Specifically, we found a 1-sigma uncertainty of 14% and 23% for GrIS and AIS, respectively:

$$\sigma_{dh_{SMB}/dt} = 0.14|dh_{SMB}/dt|, \quad GrIS \tag{22}$$

$$\sigma_{dh_{SMB}/dt} = 0.23|dh_{SMB}/dt|, \quad AIS \tag{23}$$

where $dh_{SMB}/dt$ is the SMB-induced height change in units of meters per time. Future work would likely involve developing a more comprehensive assessment of SMB against observations to quantify SMB uncertainties. For instance, SMB over the AIS is largely biased at lower accumulation rates, so uncertainty development in the future could explore more complex relationships under different climate conditions or even explore spatial biases. All uncertainties listed in the publication are expressed as their 2-sigma equivalent.

### 2.5.2 Surface Mass Balance

We evaluated our SMB estimates through comparison with *in situ* measurements from across both ice sheets. For the AIS, we attempted to replicate the analyses as presented by Mottram et al. (2021) to ease comparisons of our performance against a suite of state-of-the-art SMB models. We used a new compilation of SMB observations from Wang et al. (2021), excluding those from Dattler et al. (2019) and Medley et al. (2013). The former study generated SMB using airborne shallow radar; however, because of the lack of age constraint of the observed radar horizons, the layers were dated in a way

to allow the derived SMB estimates match the large-scale MERRA-2 mean. Thus, the Dattler et al. (2019) dataset is dependent on the MERRA-2 SMB and is excluded. The Medley et al. (2013) dataset was not used because it was excluded in the Mottram et al. (2021) evaluation, which cited the challenge in evaluating a coarsely resolved SMB dataset against finely resolved radar-derived measurements. We performed a separate analysis that includes the Medley et al. (2013) dataset.

After filtering the observations as described in Mottram et al. (2021) by limiting observations to the 1950–2018 interval, we arrived at a total number used in the evaluation of 16,427. We used a reference interval of 1987–2015 to match Mottram et al. (2021). For SMB observations that fall entirely within the reference interval, we compared the observation against the model mean SMB over the contemporaneous period. For the observations that cover years outside of the reference interval, we used those that span more than 5 years and compare the mean against the mean SMB over the reference interval. We also used the same aggregation approach by (1) interpolating the modelled SMB values to the location of the SMB observation and (2) averaging all the interpolated model values and observations that fall within the same grid cell. We do not do the comparison on the same common grid as Mottram et al. (2021), but rather use the 12.5 km grid used in this analysis. The final number of aggregated observations for comparison against modeled SMB was 1,037 as many of the observations fall within the same grid cell (1,207 if the Medley et al. (2013) dataset is included).

For the GrIS, we performed a similar analysis as with the AIS using ice core observations of SMB compiled by Fettweis et al. (Fettweis et al., 2020) and PROMICE (v2020) SMB observations compiled by Machguth et al. (2016b), filtering the latter to observations of greater than 3 months with a start date after 1980. We also used an ensemble mean of 13 SMB models (GrSMBMIP) to add context to the evaluation (Fettweis et al., 2020). For each observation, we linearly interpolated the model SMB to the observation location, repeating for both the GSFC and GrSMBMIP models. To minimize bias imparted by poor spatial sampling, we averaged all the observations and their associated model values into the 12.5 km grid used in this study, as done in Mottram et al (2021). We compared the observations to the models in three ways. First, we determined the mean GSFC SMB over the exact observation interval. Second, to ease comparison with GrSMBMIP, we calculated annual GSFC mean SMB and took the mean GSFC annual SMB of the years the observation interval covered (referred to as GSFC/ANN). Third, we perform the same as the latter with the GrSMBMIP ensemble (referred to as GrSMBMIP/ANN). The final number of aggregated observations for comparison against GSFC, GSFC/ANN, and GrSMBMIP/ANN was 312. Results from the SMB evaluation follow in Sect. 3.4 and provide the basis of our SMB uncertainty analysis in Sect. 2.5.1.

## 3. Results

### 3.1 Firn Air Content

During the RCI, the average firn air content over the GrIS was 15.7 meters (the mean 2-sigma FAC uncertainty was 3.0 m), but it varied quite substantially in space (Figure 13a) from 0.1 to 23.9 meters (lower and upper 5% bounds). The 2-sigma

FAC uncertainty varied from 0.2 to 3.9 meters (Figure 13c). The peripheral ice contained less FAC with an average of 1.6 meters (the mean 2-sigma FAC uncertainty was 0.7 m), yet, like the GrIS, there was a substantial range (0.1–12.4 m; 2-sigma FAC uncertainty: 0.1–2.6 m). Between September 1, 1996 and September 1, 2021, the mean loss of FAC over the GrIS was 4.8%, however, local losses up to 100% exist while the majority ranged between a loss of 19.1% to a gain of 1.6%. These change estimates were based on locations where the mean annual RCI SMB was greater than zero (i.e., a firn column exists). We note that our surface density model likely overpredicts the initial density value at the lowest density values (Figure 4), which suggests that the model might underpredict total FAC where the modeled initial densities are the lowest (Figure 1). We attempted to account for this bias within our uncertainty analysis by perturbing the initial density (Sect. 2.5.1).

Because of the much colder conditions, the AIS firn column contains, on average, substantially more air than the GrIS. The average FAC during the RCI for the AIS was 24.0 meters (the mean 2-sigma FAC uncertainty was 4.7 m), which typically ranges in space between 14.2 and 36.6 meters (2-sigma FAC uncertainty: 2.9–6.4 m) (Figure 13b,d). Floating ice has a lower average FAC (17.0 m) than the grounded ice (24.9 m) because of higher temperatures and increased meltwater production.

## 3.2 Surface Mass Balance

The net mass flux at the surface of an ice sheet is referred to as the surface mass balance (SMB; Eq. 5) and is typically presented in units of mass per unit time. Here, we use gigatons per year (Gt yr$^{-1}$) to refer to area-integrated values and meters of ice equivalence per year (m i.e. yr$^{-1}$) for local values (i.e., grid cell). We also present total meltwater production, $Me$. The excess $Ra + Me$ over $Ru$ is retained within the firn column in either a solid or liquid state. $Ru$ is taken directly from the CFM output and not from MERRA-2.

### 3.2.1 Greenland Ice Sheet and Peripheral Ice

Over the RCI (1980–1995), the mean annual SMB of the GrIS was $406 \pm 103$ Gt yr$^{-1}$ ($\pm$ 1 standard deviation), which was comprised of $617 \pm 62$ Gt yr$^{-1}$ in net accumulation ($Sn + Ev$), $25 \pm 5$ Gt yr$^{-1}$ in rainfall, and $237 \pm 59$ Gt yr$^{-1}$ in runoff (Figure 14). Total meltwater production averaged $361 \pm 68$ Gt yr$^{-1}$, suggesting that the firn column accommodated 39% of all liquid water at the surface ($Ra + Me$). The average local SMB was 0.25 m i.e. yr$^{-1}$; however, it typically ranged from -0.68 to +0.93 m i.e. yr$^{-1}$ (lower and upper 5% bounds) where approximately 10% of the ice sheet by area experienced $SMB < 0$. The largest positive SMB (+3.9 m i.e. yr$^{-1}$) was found in the snowfall-rich Southeastern GrIS, while the largest negative SMB (-5.6 m i.e. yr$^{-1}$) was found along the most coastal portion of the Southwestern GrIS. Such large magnitudes, however, are extremely atypical. Our choice of RCI remains an assumption and we chose to select the MERRA-2 interval that is likely similar in state to conditions of prior decades over the GrIS. We find that for the GrIS neither SMB nor any of its components nor skin temperatures experienced a significant trend over our chosen RCI (p-values > 0.3; Figure 14b). We also used a two-sample t-test to evaluate whether the variables from the RCI are sampled from a population with different

means than after the RCI (1996–2021). We found no significant difference in annual means for $SMB$ and $Sn + Ev$ between the intervals during and after the RCI; however, rainfall, meltwater production, and skin temperatures are significantly elevated post-RCI (p-values $< 0.05$). Because our spin-up involves repeating the RCI until the entire column is refreshed, our choice of RCI (1980–1995) should not generate transients associated with the initialization process in out simulation, and the firn column at the beginning of the transient simulation is in steady-state with atmospheric conditions over the RCI (1980–1995) after which the firn will evolve freely in response to post-RCI conditions.

After 2003, the mean annual SMB for the GrIS was $347 \pm 118$ Gt yr$^{-1}$, a reduction of 59 Gt yr$^{-1}$ as compared to the RCI. Insignificant increases in solid and liquid precipitation (21 Gt yr$^{-1}$) were outweighed by a strong increase in meltwater production (107 Gt yr$^{-1}$) and ultimately runoff (79 Gt yr$^{-1}$). The firn column only accommodated 37% of liquid water present at the surface, suggesting decreased firn-air storage. The ablation zone grew in area by 36%, covering 13% of the entire GrIS. After the major melt event of 2012 when GrIS experienced its second lowest SMB (+133 Gt yr$^{-1}$) over the 40-year interval, sharp reductions in runoff coupled with above normal net precipitation allowed the SMB to recover between 2013 and 2018. In 2019, however, the GrIS incurred its lowest annual SMB (+131 Gt yr$^{-1}$) due to a combination of well-below average precipitation and well-above average melt.

The SMB of Greenland peripheral ice was never positive over the entire 1980–2021 period with a mean of $-55 \pm 22$ Gt yr$^{-1}$ and $-74 \pm 25$ Gt yr$^{-1}$ during the RCI and after 2003, respectively. After 2003, like the GrIS, the peripheral ice bodies experienced minimal precipitation gains (5 Gt yr$^{-1}$) in conjunction with moderate increases in melt and runoff (both 20 Gt yr$^{-1}$). Over the entire 40-year record, the firn only accommodated 18% of all liquid water, indicating that the majority of Greenland's peripheral ice is bare ice. Local SMB over the RCI ranges from -3.6 to +1.0 m i.e. yr$^{-1}$ with a mean of -0.9 m i.e. yr$^{-1}$. As expected, 73% of the peripheral ice experienced $SMB < 0$.

### 3.2.2 Antarctic Ice Sheet

The SMB of the AIS is nearly entirely controlled by snowfall (Figure 15). Of the $2605 \pm 145$ Gt yr$^{-1}$ annual mass gain over the RCI (1980–2019), net accumulation ($Sn + Ev$) accounts for $2605 \pm 146$ Gt yr$^{-1}$ whereas rainfall contributed a mere $6 \pm 3$ Gt yr$^{-1}$ and runoff removed only $6 \pm 4$ Gt yr$^{-1}$. Meltwater production does exist ($96 \pm 30$ Gt yr$^{-1}$), however, majority (94%) is retained within the firn column. Local SMB is predominantly positive with a mean of +0.21 m i.e. yr$^{-1}$, and values commonly span +0.04 to +0.71 m i.e. yr$^{-1}$ (lower and upper 5% bounds). Approximately 0.4% of the ice sheet by area exhibited mean annual $SMB < 0$. The maximum SMB of +6.11 m i.e. yr$^{-1}$ was found along the spine of the western Antarctic Peninsula, whereas the minimum of -0.38 m i.e. yr$^{-1}$ was found at the Northwestern corner of the Ross Ice Shelf. Net snow accumulation, rainfall, runoff, and skin temperatures did not experience significant trends (p-values $> 0.4$) over the RCI; thus, we assume the full 40-year record (1980–2019) is a realistic guess regarding atmospheric conditions before 1980 in the absence of longer-term atmospheric models. Meltwater production exhibited a significant negative trend (-1.1 Gt yr$^{-2}$; p-value $= 0.01$); however, because of the extremely small contribution relative to net accumulation (Figure 15b) and its

highly localized spatial distribution, the firn column initialized over the RCI spin-up should be in equilibrium with steady state climate conditions.

Most mass gains over the AIS occur in the form of net accumulation over the grounded ice sheet ($2148 \pm 127$ Gt yr$^{-1}$), whereas floating ice accumulates $457 \pm 27$ Gt yr$^{-1}$. Although not substantial, meltwater production over floating ice ($66 \pm 19$ Gt yr$^{-1}$) was on average double that over grounded ice ($31 \pm 11$ Gt yr$^{-1}$). Nearly all this meltwater is retained within the firn
column as runoff averages 1 Gt yr$^{-1}$ and 5 Gt yr$^{-1}$ for grounded and floating ice, respectively. We note that the area of grounded ($12.1 \times 10^6$ km$^2$) ice is an order of magnitude larger than floating ($1.5 \times 10^6$ km$^2$) ice.

### 3.3 Height and Volume Change

The combined fluctuations in SMB and FAC drive the total ice-sheet volume changes due to surface processes, yet only the former constitutes an actual mass change. We evaluate the relative contributions of mass (SMB) and air (FAC) at seasonal
and multi-annual timescales. When propagating errors, we account for the variable correlation in time and space.

### 3.3.1 Greenland Ice Sheet

The seasonal amplitudes of the SMB and FAC components of ice-sheet-wide volume change averaged over the RCI are 143 km$^3$ and 236 km$^3$, respectively (Figure 16a), indicating that changes in the FAC are more than 1.5x larger than SMB at sub-annual timescales. When combined, this volume change translates into ice-sheet-wide average height change of 23 cm due
to seasonal variability of surface processes. During the RCI, volume increases until May when it typically reaches its maximum and rapidly decreases to its minimum in August, bringing the ice sheet effectively back in balance (i.e., net zero change) as by design. After the RCI, the GrIS seasonal amplitudes of the SMB and FAC components increased, respectively, to 218 km$^3$ and 308 km$^3$; however, that increase is driven largely by two extreme years in 2012 and 2019 when the GrIS lost in total 585 km$^3$ and 594 km$^3$, respectively (Figure 16c). Although FAC exhibits a larger seasonal cycle, the
contribution to ice-sheet-wide volume change over longer timescales (i.e., several years) is smaller for the FAC than SMB (Figure 16b). Between September 1, 2003 and September 1, 2021, SMB anomalies and FAC changes contributed to a decrease in GrIS volume of $1591 \pm 245$ km$^3$: $368 \pm 139$ km$^3$ due to FAC (Figure 17) and $1224 \pm 119$ km$^3$ due to SMB.

### 3.3.2 Antarctic Ice Sheet

The seasonal amplitude of height change due to surface processes alone averages to 6 cm over the entire AIS (Figure 18c),
which is one-fourth that of the GrIS (23 cm). Due to its large area, however, the seasonal volume change amounts to 808 km$^3$. The change in FAC is 2.5 times larger than SMB and dominates the seasonal signal, amounting to 576 km$^3$ in seasonal change (Figure 18a), which is larger than the seasonal signal of 340 km$^3$ from Ligtenberg et al. (2012). While the maximum and minimum volume changes due to SMB occurs in October and February, respectively, those due to FAC variability occur one month later (November and March). Between March 31, 2003 and March 31, 2021, the AIS has grown in volume by
$1526 \pm 505$ km$^3$ from surface processes alone of which $1050 \pm 252$ km$^3$ resulted from FAC changes (Figure 19) and $477 \pm$

248 km$^3$ from SMB. In sum, surface processes contributed +85 ± 28 km$^3$ yr$^{-1}$ to the volume of the AIS since 2003, a number that is vastly overshadowed by the seasonal cycle. Because the RCI encompassed the entire 1980–2019 interval, the height and volume changes in our model experiments begin and return to zero at the end of 2019 (i.e., no height change over the entire RCI).

## 3.4 Surface Mass Balance Evaluation

To contextualize the SMB values derived here from MERRA-2 and the CFM, we perform SMB evaluation against observations inspired by two recent SMB model intercomparison exercises for the AIS (Mottram et al., 2021) and GrIS (Fettweis et al., 2020).

### 3.4.1 Greenland Ice Sheet

The comparison between observations and modeled SMB for the GrIS indicates that our model performs similar to several of the models within the GrSMBMIP exercise. Figure 20 shows the performance of the GSFC/ANN comparison against the GrSMBMIP/ANN, and Table 2 provides the statistical comparison with the observations. We note here that the GrSMBMIP ensemble mean resolved SMB better than any individual model within the ensemble, so we expect the GSFC model to have lower performance metrics than the ensemble mean. The GSFC model reproduces observed SMB under near equal performance as GrSMBMIP for observations with SMB > ~ -2 kg m$^{-2}$ yr$^{-1}$, but experiences more spread from the observations at higher melt rates. Table 2 indicates that while the net bias of the GSFC/ANN model is comparable to the GrSMBMIP/ANN, the GSFC model experiences higher spread from the observations (RMSE = 0.35 kg m$^{-2}$ yr$^{-1}$), which indicates partly diminished performance in capturing the spatial variability.

Using the $n = 312$ observation-model comparison pairs (Figure 20 and Table 2), we approximate the uncertainty in the GSFC modeled SMB in a relative sense. Specifically, we found the absolute bias for each pair, $bias = |(model - observation)/model|$, and assigned an uncertainty in modeled SMB equal to the median absolute bias, which is less sensitive to outliers than the mean. The typical relative bias for GrIS is 14%, which we employ as the 1-sigma uncertainty in SMB (Sect. 2.5.1; Eq. 22).

We also directly compare the GrSMBMIP ensemble mean annual SMB with our GSFC results in Figure 21 over the common 1980–2012 interval, interpolating our model results onto the GrSMBMIP grid. The GSFC model exhibits elevated SMB over the interior relative to the GrSMBMIP ensemble mean with variable differences in sign around the periphery (i.e., exhibits positive and negative differences). The statistical summary in Table 2 suggests that integrated over the entire ice sheet, the GSFC model has a slightly higher SMB. The annual mean SMB from the GrSMBMIP of 347 Gt yr$^{-1}$ is smaller than the GSFC mean of 383 Gt yr$^{-1}$. The GrSMBMIP ensemble mean SMB trend is -7.2 Gt yr$^{-2}$ whereas our GSFC results have a slightly less negative trend of -4.5 Gt yr$^{-2}$, which falls within the entire ensemble spread (-3.1 to -12.9 Gt yr$^{-2}$). We also compare runoff values between the GrSMBMIP ensemble mean (328 Gt yr$^{-1}$) and GSFC (304 Gt yr$^{-1}$), which suggests our runoff estimates are more muted than some models, and the GSFC trend (5.4 Gt yr$^{-1}$) is less positive than GrSMBMIP

ensemble (8.0 Gt yr$^{-1}$), but still falls within the ensemble spread (4.0 to 13.4 Gt yr$^{-1}$). These findings suggest that GSFC SMB is on average larger than the ensemble mean because of larger snow accumulation and less runoff, which is also evidenced by the differences across the interior in Figure 21. The difference in SMB trend from the ensemble average is largely sourced from a difference in runoff trends as snowfall exhibits no trend in both.

Finally, we compare our degree-day model annual melt rates with those used to train our model (i.e., MARv3.5.2; Figure 22). The time series have a high correlation ($r^2 = 0.94$); however, there agreement in magnitude differs between the RCI and post-RCI. The MARv3.5.2 produces a stronger increase in melt than our degree-day model. This difference could stem from multiple sources including (1) a weaker increase in temperature within the MERRA-2 model, (2) our capping of melt factors above 1500 m (see Sect. 2.2.1), and (3) our final selection of the temperature threshold. Over the contemporaneous interval (1980–2014), we find that MARv3.5.2, MERRA-2, and our GSFC GrIS runoff values average 258, 279, and 274 Gt yr$^{-1}$. While the GSFC melt values derived in this study are lower than the training data set, runoff values are on average larger. We note that the older GSFC-FDMv1 model, along with a newer version of MAR, showed poor performance when compared with ICESat-2 derived surface height changes in the low-melt, high-elevation portions of the ice sheet over the summer melt seasons of 2019 and 2020 (Smith et al., 2022). The same study found that our new degree-day model parameterization with reduced runoff presented here performed better than v1, which averaged 307 Gt yr$^{-1}$ of runoff from 1980 through 2014. Thus, recent data suggest our modifications to the degree-day model better replicated observations; however, meltwater flux and its ultimate fate is at present the largest discrepancy between the SMB and FAC models and is the largest source of uncertainty in our results.

### 3.4.2 Antarctic Ice Sheet

Replicating the analysis within Mottram et al. (2021) was more straightforward, so we present analysis that allows for direct comparison with their results. Figure 23 compares all SMB observations with the GSFC modelled SMB, and statistics of the evaluation are presented in Table 3, broken down into different categories as done by Mottram et al. (2021). Considering the AIS as a whole, GSFC SMB has a very small positive mean bias (6 kg m$^{-2}$ yr$^{-1}$) as compared to larger, negative biases from the ensemble of models in Mottram et al. (2021). Otherwise, the performance is very similar. Table 3 suggests that the GSFC SMB over ice shelves is remarkably good as compared to the Mottram et al. (2021) ensemble that suggests most models underestimate SMB. Notable differences between the GSFC SMB and the ensemble from Mottram et al. (2021) include: (1) smaller SMB bias at lower elevations than the ensemble, (2) similar performances over mid-elevations, and (3) larger, positive bias in GSFC SMB at the highest elevations (> 2800 m) where snowfall is the lowest. We observe this bias in Figure 23 as well where the GSFC SMB values fall above the 1:1 line for the lowest observed SMB values. Thus, we find that the GSFC SMB performs well over the ice shelves and coastal grounded ice sheet, but likely overestimates SMB in the dry interior (Figure 24). As done with the GrIS, we assigned an uncertainty in modeled SMB equal to the median absolute bias between the observation pairs ($n = 1201$), which yielded a relative uncertainty of 23% for the AIS, providing the 1-sigma uncertainty in SMB within our uncertainty analysis (Sect. 2.5.1; Eq. 23).

The 1980–2010 mean annual GSFC SMB is 2,620 Gt yr$^{-1}$, which is larger than the Mottram et al. (2021) ensemble mean of 2,483 Gt yr$^{-1}$ but remains within the ensemble spread (2,023–2,752 Gt yr$^{-1}$). For the grounded ice sheet, the mean GSFC SMB is 2,170 Gt yr$^{-1}$, which is also within the model spread (1,743–2,323 Gt yr$^{-1}$) and similar to the ensemble mean of 2,073 Gt yr$^{-1}$. The fact that both the grounded and total AIS SMB are larger in the GSFC model than the ensemble average is not surprising given that: (1) most of the ensemble models have a negative bias over ice shelves and (2) the GSFC model has a positive bias over the interior of the ice sheet. In fact, the GSFC SMB over the ice shelves is 450 Gt yr$^{-1}$ a value that is only exceeded by two models within the ensemble of 9 models. The integrated GSFC SMB did not exhibit any trends through time, which is also evident in the ensemble of models from Mottram et al. (2021). We note that we use a different grid than in Mottram et al. (2021), which could have a large impact on integrated SMB (Hansen et al., 2022).

Finally, we compare our degree-day model annual melt rates with those used to train our model (i.e., Trusel et al. (2013b); Figure 25). We also compare our annual melt fluxes against two regional climate models (Van Wessem et al., 2018; Agosta et al., 2019) to provide a longer context because the QSCAT observations cover only a decade. By design, our degree-day model best matched the magnitude of the observations from Trusel et al. (2013b). The contemporaneous (1981–2016) mean annual melt rates from our degree-day model, RACMO2.3p2, and MARv3.6.4 are 99, 107, and 83 Gt yr$^{-1}$. We note that the annual means are accumulated over each melt season, so the degree-day model begins in 1981, which spans July 1, 1980 to June 30, 1981. All melt fluxes are calculated in the same fashion. The degree-day model annual melt corresponds closest with RACMO2.3p2 ($r^2 = 0.72$), followed by MARv3.6.4 ($r^2 = 0.56$). The time series from the two RCM's show a similar correspondence ($r^2 = 0.73$). These three models similarly agree on very low runoff amounts (6, 1, and 2 Gt yr$^{-1}$, respectively); however, the MERRA-2 land ice runoff is nearly an order of magnitude larger (68 Gt yr$^{-1}$). The annual runoff from our degree day model and MERRA-2 significantly correlate in time ($r^2 = 0.65$). Thus, there is a discrepancy between the firn and regional climate modeling runoff and the reanalysis-derived runoff over the Antarctic Ice Sheet. Without meltwater fluxes directly from MERRA-2, we cannot determine whether this is related to the snow model within the MERRA-2 framework or whether MERRA-2 predicts larger melt fluxes than our degree-day model leading to more runoff.

## 4 Discussion and conclusion

We present simulations of GrIS and AIS firn processes using the CFM forced by MERRA-2 atmospheric reanalysis data spanning more than 40 years. Specifically, we calibrate the Arthern et al. (2010) firn densification model through modification of its dependence on overburden and temperature. The resulting model reduces the rates of densification, largely due to reduced sensitivity to increasing overburden, which is approximated by the mean accumulation rate. Modification to the temperature dependence was necessary for the second stage of densification, which is in line with other studies that found the accumulation rate as a key parameter in model calibration (Kuipers Munneke et al., 2015; Ligtenberg et al., 2011). Our calibration differs, and is comparable to the approach by Verjans et al. (2020b), as we derive the form of our calibration using the original form of the Arthern et al. (2010) densification equation, which provides adjusted model

parameters that best fit observed depth-density profiles and the MERRA-2 climate conditions. Additionally, we calibrate the model using observations from both ice sheets, resulting in one set of adjusted parameters. It is important to note that the

adjustments to the densification model parameters reflect missing physical processes as well as persistent biases within the climate forcing (e.g., if the forcing exhibited a cold bias). Thus, application of these adjustments when using a different climate forcing is not recommended. Future work will investigate use of alternative calibration equations to assess its impact on the resulting volume changes.

The surface density parameterization is also dependent on the mean annual climate conditions derived from MERRA-2, so

any biases will manifest in the derived coefficients. We note that while the model does a satisfactory job of reproducing moderate to high surface densities (325–415 kg m$^{-3}$), it appears insufficient at capturing the lowest observed densities. Thus, our model potentially overestimates the initial density, predominantly over GrIS, which leads to an underestimation of the FAC. We do perturb the initial density in our uncertainty CFM runs, so we expect our FAC errors to reflect this lack of constraint. More exploration into the density of new snow accumulations and their subsequent evolution over short time

scales (hours to days) and across several locations is necessary to improve this simple density model. While new snow accumulation is often very low density, these values cannot be directly applied to the firn densification model, which models density evolution over coarse time steps (5 days) during which the snow can undergo rapid densification. Thus, the GSFC-FDMv1.2.1 does not account for sub-time-step surface density evolution and requires a bulk density representative of snowfall that has been exposed at the surface for several days. We note that the Arthern et al. (2010) densification model

was not developed for densification at very low densities, so even with a more realistic fresh snow density, the model as presented here and in Arthern et al. (2010) would not adequately reproduce densification of freshly fallen snow. Future improvements in the time resolution of the simulations as well as observations of the rapid evolution of new snow accumulation should provide important future improvements to the model presented. Furthermore, the modeled surface density does not evolve in time, which is likely an oversimplification, but future work will evaluate the potential to capture

seasonal initial density in future versions of the GSFC-FDM.

We next review other limitations of the work we have presented, which will be the focus of future work. The choice of running the model at 5-day time steps was a subjective choice, based on the need for computational efficiency. The firn is subject to diurnal changes in temperature and melt that our model is not capable of resolving; however, we attempt to capture much of the signal at 5-day windows through accumulating fluxes at hourly resolution such as melt and snow

accumulation. In the prior simulations, we used an effective mean temperature to try to capture the non-linear impact of the large diurnal fluctuations in temperature and their resulting impact on the densification rate. We abandoned that effort (see Sect. 2.1.3 and Appendix C) given its degraded performance when compared against simulations performed at 1-day time steps. Future work preserving both the physical and effective temperature means through time will help us better understand if we can adequately capture the sub-time-step temperature impact on densification moving forward.

While we indicate that our choice of RCI was our best attempt at capturing the long-term conditions, it remains a partly subjective choice that does have an ultimate impact on our results and interpretation. The challenge for all firn modeling

efforts is that the firn column was built of 10s to 1000s of years of snow accumulation, yet we only have a spatiotemporally complete understanding of polar climate conditions arguably since the beginning of the satellite era (1979 and onwards). Thus, we make assumptions regarding how that firn column will respond to modern conditions without knowledge of the

past prevailing conditions. Studies suggest variable spatial trends in both snow accumulation rates (Medley and Thomas, 2019; Thomas et al., 2017) and air temperatures (Steig et al., 2009; Nicolas and Bromwich, 2014; Bromwich et al., 2013) over the AIS, which are not considered in this work. Similarly, reconstructed SMB from a twentieth century found significant trends over GrIS since 1870, which this work does not capture (Hanna et al., 2011). Thus, any deviation of the RCI atmospheric conditions from reality will bias the trends in firn column evolution. Future work investigating the impact

of these reconstructed trends would help to quantify the resulting uncertainty in height changes due to long-term climate change. Deviations from observed height changes thus reflect both errors in firn modeling efforts as well as unknown trends due to a lack of constraint on recent climate, impacting results over both ice sheets.

Meltwater fluxes as well as their ultimate fate remain the largest source of uncertainty in our firn modeling effort. Our simple degree-day model of melt was employed due to the absence of MERRA-2 meltwater flux output. At present, the

CFM does not have an energy balance model subroutine, although it is in preparation, so future versions of GSFC-FDM will use a physically based melt model. Comparisons against the degree-day model training data, as well as other RCM results, suggest that we are capturing a significant portion of the annual signal (Figure 22 and Figure 25). The total magnitude of melt is less than the training dataset for the GrIS, which might be due to (1) an overestimation of melt within the RCM used to train our model, (2) a cold bias in the MERRA-2 air temperatures, (3) capping $DDFs$ above 1500 m, or (4) a combination

of the aforementioned. Thus, the runoff produced by GSFC-FDMv1.2.1 is on the lower end of several existing SMB models for the GrIS and exhibits a smaller increase in runoff through time. We note that a recent study by Smith et al. (2022) found that the older melt model used for GSFC-FDMv1.1, as well as a more recent version of MAR than used in this study (i.e., MARv3.11.5; Amory et al. (2021)), systematically overpredicted the height changes within the high-elevation pats of the ice sheet, particularly in association with melt events. After capping the unrealistic melt factors above 1500 m, the melt model

in v1.2.1 yields a better match of the firn height changes with satellite altimetry. Because this comparison only covers two melt seasons, the evaluation suggests improvement, but comparison against more melt event/seasons is necessary to fully evaluate this improvement, rule out possible compensating errors, and highlight other potential future improvements.

While not the focus of the work, one important output from the GSFC-FDMv1.2.1 simulation is surface runoff, which allows us to estimate ice sheet SMB. The mean annual GrIS SMB is comparable to SMB estimates from an ensemble of models of

varying complexity (Fettweis et al., 2020). Our estimates of the 1980–2012 GrIS mean annual SMB (383 ± 111 Gt yr$^{-1}$) and runoff (304 ± 86 Gt yr$^{-1}$) are similar to the ensemble averages (347 ± 111 Gt yr$^{-1}$ and 328 ± 101 Gt yr$^{-1}$, respectively). The lower runoff derived in this study along with slightly larger snow accumulation rates account for the larger SMB. Comparison with an accompanying Antarctic model ensemble suggests that our AIS SMB estimate for grounded and floating ice is larger than most models: Mottram et al. (2021) found the ensemble mean of AIS SMB of 2483 Gt yr$^{-1}$ (range:

2023–2752 Gt yr$^{-1}$), which is less than our estimate of 2620 Gt yr$^{-1}$. We note, however, that the evaluation in Mottram et al.

(2021) of each model against observations suggests they each contain a negative bias (i.e., the modeled SMB is typically less than the observed).

Deviations in SMB from its mean over the RCI result in ice-sheet height and volume fluctuations; however, these SMB deviations along with changes in temperature also modulate the total air content within the firn column, amplifying the mass-related height and volume fluctuations. Thus, the SMB impact on height change is twofold: both imposing a change in mass as well as a change in air (i.e., fresh snowfall is a matrix of ice and air), which means that the fluctuations in SMB and FAC change are strongly correlated. We keep height changes due to mass separated from those due to air because of the relevance to interpretation of satellite derived height changes. While the SMB and FAC contributions to total firn volume change over multiannual time scales are somewhat comparable, the seasonal signal is dominated by FAC for both ice sheets. This difference suggests that 62% for the GrIS and 71% for the AIS of sub annual volume fluctuations are in response to a change in the air content rather than actual mass change. Thus, determination of seasonal mass change using satellite altimetry requires a substantial FAC correction, highlighting the importance of firn densification and the atmospheric models that force the FDMs, especially when investigating shorter intervals of change as not being mindful of the seasonal cycles of SMB and FAC can generate large biases.

Finally, we briefly note the differences between the GrIS GSFC results for v1.1 and v1.2.1 (differences were negligible over the AIS). The largest difference is the muted FAC change through time integrated over the GrIS (Figure 16b). This change is partly due to the improved surface density model that yields lower densities over the interior and higher densities around the periphery, which led to a larger increase in firn air over the interior in response to additional snowfall and a smaller decrease in firn air in the percolation zone (Figure 17). Other factors include the modification of the melt regime at high elevations, which acted to reduce total meltwater fluxes in the interior, reducing the FAC losses due to melt. Finally, the overestimation of density (or underestimation of FAC) at sites with high melt would potentially generate FAC change biased low as there is less air to lose when melt occurs. Thus, substantial FAC loss occurs along the periphery of GrIS, but those losses are partly balanced by gains in the interior. Small changes in the surface density and liquid water processes yield measurable changes in FAC and SMB, and their uncertainty limits our ability to constrain mass balance estimates from satellite altimetry. Thus, future work constraining melt, its routing, and the initial density and their spatiotemporal evolution is necessary and should be a priority.

The time series of firn height and volume change, split into its respective SMB (ice) and FAC (air) components, provide the data necessary to isolate the ice-dynamical change from the changes observed using airborne and satellite altimeters. Future work improving the representation of the near surface climate, initial density, and especially liquid water processes within the firn column should improve future iterations of GSFC-FDM modeled firn volume changes. Because of the challenges in measuring firn processes *in situ*, future evaluations of firn densification model representation will likely rely on direct comparisons with altimetry-derived volume changes.

### Appendix A. Density Data

The calibration depth-density data were compiled through combination of the SUMup datasets (Koenig and Montgomery, 2018; Montgomery et al., 2018) and other compiled sources that are listed in Table A1.

Table A1. Locations and sources of the depth-density profiles used in model calibration.

| ID | Name | Latitude | Longitude | Elevation | Source |
|----|------|----------|-----------|-----------|--------|
| 1 | US-ITASE-99-1 | -80.62 | -122.63 | 1350 | Mayewski and Dixon (2013) |
| 2 | US-ITASE-99-2 | -81.2 | -126.17 | 1040 | Mayewski and Dixon (2013) |
| 3 | US_ITASE-00-1 A | -79.3831 | -111.239 | 1791 | Mayewski and Dixon (2013) |
| 4 | US_ITASE-00-2 C | -78.733 | -111.4966 | 1675 | Mayewski and Dixon (2013) |
| 5 | US_ITASE-00-3 D | -78.433 | -115.9172 | 1742 | Mayewski and Dixon (2013) |
| 6 | US_ITASE-00-4 E | -78.0829 | -120.0764 | 1697 | Mayewski and Dixon (2013) |
| 7 | US_ITASE-00-5 F | -77.683 | -123.995 | 1828 | Mayewski and Dixon (2013) |
| 8 | US_ITASE-00-6 H | -78.3325 | -124.484 | 1639 | Mayewski and Dixon (2013) |
| 9 | US_ITASE-00-7 I | -79.133 | -122.267 | 1495 | Mayewski and Dixon (2013) |
| 10 | US_ITASE-01-1 | -79.1597 | -104.9672 | 1842 | Mayewski and Dixon (2013) |
| 11 | US_ITASE-01-2 | -77.8436 | -102.9103 | 1336 | Mayewski and Dixon (2013) |
| 12 | US_ITASE-01-3 | -78.1202 | -95.6463 | 1620 | Mayewski and Dixon (2013) |
| 13 | US_ITASE-01-4 | -77.6116 | -92.2483 | 1483 | Mayewski and Dixon (2013) |
| 14 | US_ITASE-01-5 | -77.0593 | -89.1376 | 1239 | Mayewski and Dixon (2013) |
| 15 | US_ITASE-01-6 | -76.0973 | -89.0177 | 1228 | Mayewski and Dixon (2013) |
| 16 | US_ITASE-02-1 | -82.00099 | -110.00816 | 1746 | Mayewski and Dixon (2013) |
| 17 | US_ITASE-02-2 | -83.500781 | -104.98681 | 1957 | Mayewski and Dixon (2013) |
| 18 | US_ITASE-02-3 | -85.000451 | -104.99531 | 2396 | Mayewski and Dixon (2013) |
| 19 | US_ITASE-02-4 | -86.5025 | -107.9903 | 2586 | Mayewski and Dixon (2013) |
| 20 | US_ITASE-02-5 | -88.002153 | -107.98333 | 2747 | Mayewski and Dixon (2013) |
| 21 | US_ITASE-02-6 (SPRESSO) | -89.93325 | 144.39383 | 2808 | Mayewski and Dixon (2013) |
| 22 | US_ITASE-03-1 | -86.84 | 95.31 | 3124.2 | Mayewski and Dixon (2013) |
| 23 | US_ITASE-03-3 | -82.08 | 101.96 | 3444.24 | Mayewski and Dixon (2013) |
| 24 | US_ITASE-03-4 | -81.65 | 122.6 | 2965.704 | Mayewski and Dixon (2013) |
| 25 | US_ITASE-03-6 | -80.39 | 138.92 | 2392.68 | Mayewski and Dixon (2013) |
| 26 | US_ITASE-03-7 | -77.88 | 158.66 | 2264.616 | Mayewski and Dixon (2013) |
| 27 | US_ITASE-06-1 | -77.880222 | 158.45822 | 2365 | Mayewski and Dixon (2013) |
| 28 | US_ITASE-06-2 | -77.761944 | 153.38139 | 2277 | Mayewski and Dixon (2013) |
| 29 | US_ITASE-06-3 | -79.0362 | 149.6803 | 2241 | Mayewski and Dixon (2013) |
| 30 | US_ITASE-07-1 | -81.658 | 136.084 | 2450 | Mayewski and Dixon (2013) |
| 31 | US_ITASE-07-2 | -84.39507 | 140.6308 | 2645 | Mayewski and Dixon (2013) |
| 32 | US_ITASE-07-3 | -85.781889 | 145.71948 | 2817 | Mayewski and Dixon (2013) |
| 33 | US_ITASE-07-4 | -88.50953 | 178.53079 | 3090 | Mayewski and Dixon (2013) |
| 34 | PARCA-NASA EAST A | 75.0 | -30 | 2631 | Mosley-Thompson et al. (2001) |
| 35 | PARCA-NASA EAST B | 75.0 | -30 | 2631 | Mosley-Thompson et al. (2001) |
| 36 | PARCA-S DOME B | 63.149 | -44.817 | 2850 | Mosley-Thompson et al. (2001) |
| 37 | PARCA-S DOME A | 63.149 | -44.817 | 2850 | Mosley-Thompson et al. (2001) |
| 38 | PARCA-S DOME A (2) | 63.149 | -44.817 | 2850 | Mosley-Thompson et al. (2001) |
| 39 | PARCA-S TUNU C | 69.5 | -34.5 | 2650 | Mosley-Thompson et al. (2001) |
| 40 | PARCA-S TUNU B | 69.5 | -34.5 | 2650 | Mosley-Thompson et al. (2001) |
| 41 | PARCA-S TUNU A | 69.5 | -34.5 | 2650 | Mosley-Thompson et al. (2001) |
| 42 | PARCA-S TUNU A (2) | 69.5 | -34.5 | 2650 | Mosley-Thompson et al. (2001) |

| 43 | PARCA-N DYE 3 B (Saddle) | 66 | -44.501 | 2640 | Mosley-Thompson et al. (2001) |
|----|---------------------------|-----|---------|------|-------------------------------|
| 44 | PARCA-N DYE 3 A (Saddle) | 66 | -44.501 | 2640 | Mosley-Thompson et al. (2001) |
| 45 | PARCA-7653 B | 76 | -53 | 2200 | Mosley-Thompson et al. (2001) |
| 46 | PARCA-7653 A | 76 | -53 | 2200 | Mosley-Thompson et al. (2001) |
| 47 | PARCA-7551 | 69.5 | -34.5 | 2650 | Mosley-Thompson et al. (2001) |
| 48 | PARCA-7247 | 71.926 | -47.487 | 2277 | Mosley-Thompson et al. (2001) |
| 49 | PARCA-7147 | 71.05 | -47.23 | 2134 | Mosley-Thompson et al. (2001) |
| 50 | NUS08-7 | -74.11996 | 1.60049 | 2679.67 | Pers. comm. J.R. McConnell (2017) |
| 51 | NUS08-5 | -82.62929 | 17.87432 | 2544.26 | Pers. comm. J.R. McConnell (2017) |
| 52 | NUS08-4 | -82.8111 | 18.9 | 2551.59 | Pers. comm. J.R. McConnell (2017) |
| 53 | NUS07-2 | -76.06524 | 22.46301 | 3587.71 | Pers. comm. J.R. McConnell (2017) |
| 54 | NUS07-5 | -78.64639 | 35.64142 | 3620.05 | Pers. comm. J.R. McConnell (2017) |
| 55 | NUS07-7 | -82.06607 | 54.89009 | 3716.09 | Pers. comm. J.R. McConnell (2017) |
| 56 | BER01C09_01 | -78.3 | -46.283 | 730 | Wagenbach et al. (1994a) |
| 57 | BER02C09_02 | -79.658 | -45.617 | 940 | Wagenbach et al. (1994b) |
| 58 | DML01C97_00 | -78.855 | -2.55 | 2831 | Oerter et al. (1999a) |
| 59 | DML03C97_00 | -74.4995 | 1.961167 | 2843-2855 | Oerter et al. (1999b) |
| 60 | DML03C98_09 | -74.499167 | 1.960833 | 2843-2855 | Oerter et al. (2000a) |
| 61 | DML04C97_00 | -74.399 | 7.2175 | 3161-3179 | Oerter et al. (1999c) |
| 62 | DML05C98_06 | -75.002667 | 0.022667 | 2880 | Oerter et al. (2000b) |
| 63 | DML05C98_07 | -74.997 | 0.036167 | 2880 | Oerter et al. (2000c) |
| 64 | DML07C97_00 | -75.5815 | -3.430333 | 2669, 2680 | Oerter et al. (1999e) |
| 65 | DML09C97_00 | -75.933 | 7.213 | 3145-3156 | Oerter et al. (1999g) |
| 66 | DML10C97_00 | -75.216667 | 11.35 | 3349-3364 | Oerter et al. (1999h) |
| 67 | DML11C98_03 | -74.854667 | -8.497 | 2600 | Oerter et al. (2000e) |
| 68 | DML12C98_17 | -75.000667 | -6.498333 | 2680 | Oerter et al. (2000f) |
| 69 | DML13C98_16 | -75 | -4.496333 | 2740 | Oerter et al. (2000g) |
| 70 | DML14C98_15 | -74.949167 | -1.4945 | 2840 | Oerter et al. (2000h) |
| 71 | DML15C98_14 | -75.083667 | 2.501 | 2970 | Oerter et al. (2000i) |
| 72 | DML07C98_31 | -75.5815 | -3.430333 | 2669-2680 | Oerter et al. (2004) |
| 73 | DML08C97_00 | -75.752833 | 3.282833 | 2962-2971 | Oerter et al. (1999f) |
| 74 | DML16C98_13 | -75.16733 | 5.003333 | 3100 | Oerter et al. (2000j) |
| 75 | DML17C98_33 | -75.167 | 6.4985 | 3160 | Oerter et al. (2000k) |
| 76 | DML18C98_04 | -75.250333 | -6 | 2630 | Oerter et al. (2000l) |
| 77 | DML19C98_05 | -75.167333 | -0.0995 | 2840 | Oerter et al. (2000m) |
| 78 | DML20C98_08 | -74.750667 | 0.999833 | 2830 | Oerter et al. (2000n) |
| 79 | DML21C98_10 | -74.667167 | 4.001667 | 2980 | Oerter et al. (2000o) |
| 80 | DML22C98_11 | -75.084 | 6.5 | 3160 | Oerter et al. (2000p) |
| 81 | DML23C98_12 | -75.250833 | 6.501667 | 3160 | Oerter et al. (2000q) |
| 82 | DML24C98_18 | -74.449 | -9.18067 | 2169 | Oerter et al. (2000r) |
| 83 | DML25C00_01 | -75.006 | 0.081867 | 2882 | Graf et al. (2002a) |
| 84 | DML26C00_03 | -74.839367 | 0.00995 | 2874 | Graf et al. (2002b) |
| 85 | DML27C00_04 | -75.056 | 0.704017 | 2899 | Graf et al. (2002c) |
| 86 | DML28C01_00 | -75.0017 | 0.0678 | 2882 | Oerter (2002) |
| 87 | DML60C98_02 | -74.205 | -9.741667 | 1439-1451 | Oerter et al. (2000s) |
| 88 | BER11C95_25 | -79.6146 | -45.72433 | 886 | Gerland and Wilhelms (1999) |
| 89 | DML05C98_32 | -75.002333 | 0.007 | 2882-2892 | Oerter et al. (2000d) |
| 90 | DML06C97_00 | -75.000667 | 8.005333 | 2880-3246 | Oerter et al. (1999d) |
| 91 | DML66C03_01 | -71.110709 | 1.646268 | 1013 | Anschutz and Oerter (2007) |
| 92 | DML96C07_39 | -71.4083 | -9.9167 | 655 | Wilhelms (2007) |
| 93 | DML641C02_01 | -71.214361 | -6.79861 | 600 | Fernandoy et al. (2010a) |

| 94 | DML651C02_03 | -71.457222 | -9.860722 | 630 | Fernandoy et al. (2010b) |
|---|---|---|---|---|---|
| 95 | FRI0C92_246 | -78.42778 | -52.50639 | 68 | Graf and Oerter (2006d) |
| 96 | FRI07C84_340 | -78.60611 | -55.43167 | 92 | Graf et al. (1988) |
| 97 | FRI09C90_13 | -76.98111 | -52.26778 | 41 | Graf and Oerter (2006m) |
| 98 | FRI09C90_90 | -76.98111 | -52.26778 | 41 | Graf and Oerter (2006l) |
| 99 | FRI10C90_136 | -77.19389 | -53.14083 | --- | Graf and Oerter (2006f) |
| 100 | FRI11C90_235 | -77.51306 | -54.54667 | --- | Graf and Oerter (2006a) |
| 101 | FRI12C90_236 | -77.9375 | -55.97833 | 61 | Graf and Oerter (2006g) |
| 102 | FRI12C92_15 | -77.935 | -55.936 | 61 | Graf and Oerter (2006n) |
| 103 | FRI13C90_335 | -78.30194 | -56.98 | 64 | Graf and Oerter (2006) |
| 104 | FRI14C90_336 | -78.72167 | -57.8475 | --- | Graf and Oerter (2006h) |
| 105 | FRI15C90_131 | -76.95889 | -54.69222 | --- | Graf and Oerter (2006b) |
| 106 | FRI16C90_230 | -77.35783 | -56.05933 | --- | Graf and Oerter (2006) |
| 107 | FRI17C90_231 | -77.68222 | -57.32639 | --- | Graf and Oerter (2006c) |
| 108 | FRI18C90_330 | -78.03361 | -58.69056 | --- | Graf and Oerter (2006i) |
| 109 | FRI19C90_05 | -81.46306 | -0.61472 | --- | Graf and Oerter (2006j) |
| 110 | FRI20C90_06 | -81.617 | -57.917 | 133 | Graf and Oerter (2006k) |
| 111 | FRI21C90_HWF | -78.31722 | -39.43472 | --- | Graf and Oerter (2006e) |
| 112 | FRI23C95_16 | -77.99167 | -51.53333 | 65 | Graf et al. (1999p) |
| 113 | FRI24C95_15 | -78.41333 | -52.4733 | 68 | Graf et al. (1999a) |
| 114 | FRI25C95_14 | -78.84 | -53.47333 | 71 | Graf et al. (1999b) |
| 115 | FRI26C95_13 | -79.26833 | -54.20167 | 75 | Graf et al. (1999k) |
| 116 | FRI27C95_12 | -79.97 | -54.89167 | 85 | Graf et al. (1999c) |
| 117 | FRI28C95_11 | -80 | -55.5 | 93 | Graf et al. (1999d) |
| 118 | FRI29C95_10 | -80.43 | -55.98 | 104 | Graf et al. (1999e) |
| 119 | FRI30C95_09 | -80.833333 | -56.58833 | 107 | Graf et al. (1999l) |
| 120 | FRI31C95_08 | -81.21833 | -57.20333 | 125 | Graf et al. (1999m) |
| 121 | FRI32C95_07 | -81.605 | -57.88833 | 132 | Graf et al. (1999h) |
| 122 | FRI33C95_06 | -82.335 | -57.82667 | 143 | Graf et al. (1999f) |
| 123 | FRI34C95_03 | -82.75 | -58.69167 | 145 | Graf et al. (1999g) |
| 124 | FRI35C95_01 | -83.016667 | -59.575 | 163 | Graf et al. (1999i) |
| 125 | FRI36C95_02 | -83.385 | -60.06333 | 185 | Graf et al. (1999n) |
| 126 | FRI37C95_05 | -83.97833 | -60.36 | 482 | Graf et al. (1999o) |
| 127 | FRI38C95_04 | -84.81833 | -59.635 | 1191 | Graf et al. (1999j) |
| 128 | NM033C98_01 | -70.706667 | -8.426667 | 35 | Oerter et al. (2000t) |
| 129 | ngt03C93.2 | 73.9402 | -37.6299 | 3040 | Wilhelms (2000a) |
| 130 | ngt06C93.2 | 75.2504 | -37.6248 | 2820 | Wilhelms (2000b) |
| 131 | ngt14C93.2 | 76.617 | -36.4033 | 2508 | Wilhelms (2000c) |
| 132 | ngt27C94.2 | 80 | -41.1374 | 2185 | Wilhelms (2000d) |
| 133 | ngt37C95.2 | 77.2533 | -49.2167 | 2598 | Miller and Schwager (2000a) |
| 134 | ngt42C95.2 | 76.0039 | -43.492 | 2874 | Miller and Schwager (2000b) |
| 135 | NM01C82_04 | -70.6167 | -8.3667 | 28 | Schlosser et al. (2002) |
| 136 | NM02C02_02 | -70.655692 | -8.253632 | 28 | Fernandoy et al. (2010c) |
| 137 | SUFA 2007 Core | 72.5961 | -38.421972 | 3200 | Adolph and Albert (2014) |
| 138 | PARCA-6345 | 63.8 | -45 | 2730 | Mosley-Thompson et al. (2001) |
| 139 | PARCA-6348 | 63 | -48 | 1960 | Mosley-Thompson et al. (2001) |
| 140 | PARCA-6642B | 66.5 | -42.5 | 2380 | Mosley-Thompson et al. (2001) |
| 141 | PARCA-6745 | 67.5 | -45 | 2250 | Mosley-Thompson et al. (2001) |
| 142 | PARCA-6839 | 68.5 | -39.5 | 2790 | Mosley-Thompson et al. (2001) |
| 143 | PARCA-6841 | 68 | -41 | 2640 | Mosley-Thompson et al. (2001) |
| 144 | PARCA-6938 | 69 | -38 | 2920 | Mosley-Thompson et al. (2001) |

| | | | | | |
|---|---|---|---|---|---|
| 145 | PARCA-6939 | 69.6 | -39 | 2955 | Mosley-Thompson et al. (2001) |
| 146 | PARCA-6941 | 69.4 | -41 | 2765 | Mosley-Thompson et al. (2001) |
| 147 | PARCA-6943 | 69.2 | -43 | 2500 | Mosley-Thompson et al. (2001) |
| 148 | PARCA-6945 | 69 | -45 | 2150 | Mosley-Thompson et al. (2001) |
| 149 | PARCA-7145 | 71.5 | -45 | 2615 | Mosley-Thompson et al. (2001) |
| 150 | PARCA-7245 | 72.25 | -45 | 2770 | Mosley-Thompson et al. (2001) |
| 151 | PARCA-7249 | 72.2 | -49.4 | 2170 | Mosley-Thompson et al. (2001) |
| 152 | PARCA-7345 | 73 | -45 | 2815 | Mosley-Thompson et al. (2001) |
| 153 | PARCA-7347 | 73.6 | -47.2 | 2600 | Mosley-Thompson et al. (2001) |
| 154 | IC12 | -70.2458 | 26.3349 | --- | Philippe et al. (2016) |
| 155 | WDC06A | -79.4828 | -112.008 | --- | Kreutz et al. (2011) |
| 156 | FA13 | 66.1812 | -39.0345 | 1563 | Koenig et al. (2014) |
| 157 | GrIT (1) | 73.344 | -39.7235 | --- | Hawley et al. (2014) |
| 158 | GrIT (2) | 74.01818 | -40.6216 | --- | Hawley et al. (2014) |
| 159 | GrIT (3) | 76.499883 | -43.732217 | 2803 | Hawley et al. (2014) |
| 160 | GrIT (4) | 76.50235 | -44.8438 | --- | Hawley et al. (2014) |
| 161 | GrIT (5) | 77.6248 | -58.5284 | --- | Hawley et al. (2014) |
| 162 | GrIT (6) | 77.37073 | -55.927 | --- | Hawley et al. (2014) |
| 163 | GrIT (7) | 77.4492 | -50.5395 | --- | Hawley et al. (2014) |
| 164 | NEEM2009S2 | 77.45 | -51.06 | --- | Baker (2012) |
| 165 | ACT10-A | 65.9671 | -41.4807 | 1825 | Miege et al. (2013) |
| 166 | ACT10-B | 65.7751 | -41.8672 | 1999 | Miege et al. (2013) |
| 167 | ACT10-C | 65.9997 | -42.7831 | 2354 | Miege et al. (2013) |
| 168 | DIV2010 | -76.77 | -101.738 | 1329 | Medley et al. (2014) |
| 169 | PIG2010 | -77.957 | -95.962 | 1593 | Medley et al. (2014) |
| 170 | THW2010 | -76.952 | -121.22 | 2020 | Medley et al. (2014) |
| 171 | BYRD | -80 | -120 | 1500 | Gow (1968) |
| 172 | Camp Century | 77.18333 | -61.16667 | 1886 | Kovacs et al. (1969) |
| 173 | DE08 DE08-2 | -66.721944 | 113.19944 | 1250 | Etheridge and Wookey (1989) |
| 174 | Dome C | -74.5 | 123.6667 | 3240 | Alley (1980) |
| 175 | Dome GRIP | 72.56667 | -37.616667 | 3230 | Spencer et al. (2001) |
| 176 | DSS | -66.769722 | 112.80694 | 1370 | Spencer et al. (2001) |
| 177 | Dye3-11B-1984 | 65.18333 | -43.8333 | 2479 | Spencer et al. (2001) |
| 178 | Dye3-15B-1984 | 65.18333 | -43.8333 | 2479 | Spencer et al. (2001) |
| 179 | Dye3-16C-1984 | 65.18333 | -43.8333 | 2479 | Spencer et al. (2001) |
| 180 | Dye3-4B-1983 | 65.18333 | -43.8333 | 2479 | Spencer et al. (2001) |
| 181 | Dye3-5B-1984 | 65.18333 | -43.8333 | 2479 | Spencer et al. (2001) |
| 182 | Dye3-9B-1984 | 65.18333 | -43.8333 | 2479 | Spencer et al. (2001) |
| 183 | Dye3-station1-1983 | 65.18333 | -43.8333 | 2479 | Spencer et al. (2001) |
| 184 | Eismitte | 71.75 | -40.75 | 3000 | Spencer et al. (2001) |
| 185 | Inge Lehmann | 77.95 | -39.18333 | 2407 | Gow (1975) |
| 186 | Isaksson A | -72.654167 | -16.645556 | 30 | Spencer et al. (2001) |
| 187 | Isaksson C | -72.761944 | -14.589722 | 70 | Spencer et al. (2001) |
| 188 | Isasksson D | -73.456667 | -12.5575 | 300 | Spencer et al. (2001) |
| 189 | Isaksson E | -73.593889 | -12.426667 | 700 | Spencer et al. (2001) |
| 190 | Isaksson E30m | -73.6 | -12.4333 | 700 | Spencer et al. (2001) |
| 191 | Isaksson F | -73.815833 | -12.210278 | 800 | Spencer et al. (2001) |
| 192 | Isaksson G | -74.013889 | -12.016389 | 1200 | Spencer et al. (2001) |
| 193 | Isaksson G26m | -74.016667 | -12.016667 | 1200 | Spencer et al. (2001) |
| 194 | Isaksson H | -74.351889 | -11.7225 | 1200 | Spencer et al. (2001) |
| 195 | Isaksson I | -74.76667 | -10.78333 | 2300 | Spencer et al. (2001) |

| 196 | Isaksson J | -75.1 | -9.5 | 3000 | Spencer et al. (2001) |
|---|---|---|---|---|---|
| 197 | Isaksson 75 S2 E | -75 | 2 | 2900 | Spencer et al. (2001) |
| 198 | Isaksson 74 16S0 37E shallow | -74.26667 | 0.616667 | 2700 | Spencer et al. (2001) |
| 199 | Isaksson 76 32S6 08E | -76.5333 | 6.1333 | 2300 | Spencer et al. (2001) |
| 200 | JARE | -70.698333 | 44.331667 | 2230 | Kusunoki and Suzuki (1978) |
| 201 | JARE11 | -70.698333 | 44.331667 | 2230 | Kusunoki and Suzuki (1978) |
| 202 | Marie Byrd Land Traverse | -79.495 | -120.0333 | 1544 | Pirrit and Doumani (1961) |
| 203 | Mile 60 | -79.00333 | -119.56667 | 1592 | Pirrit and Doumani (1961) |
| 204 | Mile 90 | -78.505 | -119.71667 | 1616 | Pirrit and Doumani (1961) |
| 205 | Mile 120 | -77.996667 | 120.01667 | 1690 | Pirrit and Doumani (1961) |
| 206 | Mile 150 | -77.496667 | -120.01667 | 1775 | Pirrit and Doumani (1961) |
| 207 | Mile 167 | -77.225 | -119.85 | 1819 | Pirrit and Doumani (1961) |
| 208 | Mile 198 | -76.85 | -118.2333 | 1899 | Pirrit and Doumani (1961) |
| 209 | Mile 222 | -76.626667 | -117.61667 | 1530 | Pirrit and Doumani (1961) |
| 210 | Mile 258 | -76.061667 | -116.95 | 1575 | Pirrit and Doumani (1961) |
| 211 | Mile 288 | -75.588333 | -116.45 | 1117 | Pirrit and Doumani (1961) |
| 212 | Mile 360 | -75.416667 | -116.3 | 83 | Pirrit and Doumani (1961) |
| 213 | Mile 457 | -74.99333 | -116.11667 | 849 | Pirrit and Doumani (1961) |
| 214 | Mile 529 | -75.786667 | -118.75 | 1644 | Pirrit and Doumani (1961) |
| 215 | Mile 565 | -75.986667 | -121.08333 | 1864 | Pirrit and Doumani (1961) |
| 216 | Mile 603 | -76.016667 | -123.68333 | 2108 | Pirrit and Doumani (1961) |
| 217 | Mile 639 | -75.711667 | -125.6333 | 1687 | Pirrit and Doumani (1961) |
| 218 | Mile 676 | -75.796667 | -128.06667 | 2002 | Pirrit and Doumani (1961) |
| 219 | Mile 711.5 | -76.038333 | -130.16667 | 1904 | Pirrit and Doumani (1961) |
| 220 | Mile 747 | -76.338333 | -132.3 | 2138 | Pirrit and Doumani (1961) |
| 221 | Mile 783 | -76.638333 | -134.5 | 2157 | Pirrit and Doumani (1961) |
| 222 | Mile 819 | -76.9 | -136.86667 | 1844 | Pirrit and Doumani (1961) |
| 223 | Mile 855 | -77.15 | -139.3 | 1498 | Pirrit and Doumani (1961) |
| 224 | Mile 890 | -77.358333 | -141.76667 | 1102 | Pirrit and Doumani (1961) |
| 225 | Mile 927 | -77.838333 | -139.95 | 1134 | Pirrit and Doumani (1961) |
| 226 | Mile 963 | -78.311667 | -138.16667 | 1053 | Pirrit and Doumani (1961) |
| 227 | Mizuho G6 | -73.112778 | 39.758333 | 3005 | Watanabe et al. (1997) |
| 228 | Mizuho G15 | -71.19444 | 45.979167 | 2571 | Watanabe et al. (1997) |
| 229 | Mizuho H15 | -69.079444 | 40.781667 | 1050 | Watanabe et al. (1997) |
| 230 | Mizuho S25 | -69.031667 | 40.45556 | 896 | Watanabe et al. (1997) |
| 231 | Ridge B-C | -82.8919 | -136.6603 | 509 | Alley (1987) |
| 232 | Site A | 70.75 | -35.958333 | 3145 | Alley (1987) |
| 233 | Site A (Crete) | 70.634911 | -35.8200 | 3092 | Clausen et al. (1988) |
| 234 | Site B | 70.659011 | -35.4788 | 3138 | Clausen et al. (1988) |
| 235 | Site C | 70.677 | -35.7870 | 3072 | Clausen et al. (1988) |
| 236 | Site D | 70.639828 | -35.6178 | 3018 | Clausen et al. (1988) |
| 237 | Site E | 71.759261 | -35.8505 | 3087 | Clausen et al. (1988) |
| 238 | Site F | 71.492 | -35.8812 | 3092 | Clausen et al. (1988) |
| 239 | Site G | 71.15495 | -35.8377 | 3098 | Clausen et al. (1988) |
| 240 | Site H | 70.8651 | -35.8381 | 3102 | Clausen et al. (1988) |
| 241 | Site 2 | 76.98333 | -56.06667 | 2000 | Langway (1970) |
| 242 | South Pole | -90 | 0 | 2850 | Spencer et al. (2001) |
| 243 | Victoria Land Traverse | -75 | 147 | 2520 | Stuart and Heine (1961) |
| 244 | Station 519 | -74 | 143 | 2541 | Stuart and Heine (1961) |
| 245 | Station 521 | -73 | 142 | 2516 | Stuart and Heine (1961) |
| 246 | Station 524 | -73 | 141 | 2498 | Stuart and Heine (1961) |

| 247 | Station 527 | -72 | 140 | 2467 | Stuart and Heine (1961) |
|-----|-------------|-----|-----|------|--------------------------|
| 248 | Station 531 | -71 | 139 | 2513 | Stuart and Heine (1961) |
| 249 | Station 536 | -72 | 143 | 2356 | Stuart and Heine (1961) |
| 250 | Station 540 | -72 | 146 | 2287 | Stuart and Heine (1961) |
| 251 | Station 544 | -72 | 148 | 2216 | Stuart and Heine (1961) |
| 252 | Station 548 | -72 | 151 | 2205 | Stuart and Heine (1961) |
| 253 | Station 550 | -72 | 154 | 2220 | Stuart and Heine (1961) |
| 254 | Station 553 | -72 | 156 | 2262 | Stuart and Heine (1961) |
| 255 | Station 556 | -72 | 159 | 231 | Stuart and Heine (1961) |
| 256 | Taylor Dome | -83.47778 | -138.09694 | 2437 | Spencer et al. (2001) |
| 257 | Upstream B | -83.47778 | -138.09694 | 664 | Alley (1987) |
| 258 | Vostok (BH-3, BH-5) | -78.46667 | 106.8 | 3502 | Spencer et al. (2001) |

865

**Appendix B. Linear Fit to the Logarithmic Density Profile**

We compared the fit statistics when making a linear fit to the logarithmic density profile versus a linear fit to the actual density profile for each stage of densification. Table B1 summarizes the results taken from the $n = 141$ stage 1 observations and $n = 76$ stage 2 observations (Sect. 2.1.3). For each observation, we use all the depth-density measurements in depth to calculate the corresponding *RMSE* and $r^2$ of the $n = 141$ stage 1 fits and $n = 76$ stage 2 fits. The performances are nearly identical for stage 1; however, the fit to logarithmic density profile is significantly better than using the actual density data based on a two-sample t-test ($p < 0.01$).

Table B1. Fit statistics

| | Logarithmic Density Profile | | | Linear Density Profile | | |
|---|---|---|---|---|---|---|
| | Lower Quartile | Median | Upper Quartile | Lower Quartile | Median | Upper Quartile |
| **Stage 1** | | | | | | |
| RMSE (kg m$^{-3}$) | 12.05 | 14.94 | 19.35 | 12.11 | 14.95 | 19.42 |
| $r^2$ | 0.94 | 0.96 | 0.97 | 0.93 | 0.96 | 0.97 |
| **Stage 2** | | | | | | |
| RMSE (kg m$^{-3}$) | *4.14* | *5.75* | *9.31* | 5.83 | 8.09 | 11.39 |
| $r^2$ | 0.98 | 0.99 | 1.00 | 0.97 | 0.99 | 0.99 |

875

## Appendix C. Discontinued Use of the Effective Mean

We tested how the model results are affected by the surface-temperature averaging scheme, which is needed to upscale the forcing data from its native 1-hour resolution to the desired 5-day resolution for the CFM runs.

To do so, we performed three types of model runs. In the first, we ran the CFM with 1-day time steps, using the 1-hour MERRA-2 fields (labeled in Figure C1 as '1 day'). In the second, we ran the CFM with 5-day time steps, and the surface temperature was calculated by taking the mean temperature for each 5-day period (labeled as '5 day, mean T'). In the third, we also ran the CFM with 5-day time steps, but we calculated the 5-day 'effective' mean temperature, given by:

$$T_{eff} = \frac{-Q}{R * \ln(K)}, \tag{C1}$$

with

$$K = \frac{1}{n}\sum_{j=1}^{n} e^{-\frac{Q}{RT_j}}, \tag{C2}$$

where $n$ is the number of days to average over (here, $n = 5$), $Q = 59.5$ kJ/mol is the activation energy, $R = 8.314$ J/mol/K, and $T_j$ are the temperatures (K) of each day of the resampling interval. The value for $Q$ used was based on the calibrated activation energy for the prior GSFC model v1.

We ran the CFM with the 3 types of runs for two different sites (South Pole and Summit, Greenland). Figure C1 shows the Firn Air Content (FAC) change from 1980 to 2021 predicted for the two sites for each of the three model run types. Table C1 shows, for each site, the mean FAC for the entirety of each model run (Mean FAC row), the change in FAC from the start of the model run to the final time step (FAC change), and the mean modeled FAC in 2020 minus the mean modeled FAC in 1980.

In both cases, the effective mean runs produce a lower total FAC than the 1-day and 5-day mean runs. The FAC change using the 5-day mean setting gives a FAC change that is closer to the 1-day value, whereas the effective mean runs predict a smaller FAC change than the 1-day runs. Thus, the use of an effective mean was abandoned; however, future work on the CFM might allow for tracking of both effective mean and physical mean of the firn parcels, which might resolve these discrepancies.

Table C1: The mean FAC, change in FAC, and 2020 mean FAC minus 1980 mean FAC predicted for each of the three model run types, for each site. The 5-day mean T results are closer to the 1-day results than the 5-day effective T method.

**Summit**

| | 5 day, mean T | 5 day, effective T | 1 day |
|---|---|---|---|
| Mean FAC (m) | 27.6 | 26.3 | 27.9 |
| FAC change (m) | 0.206 | 0.176 | 0.218 |
| mean 2020 FAC-mean 1980 FAC (m) | 0.185 | 0.155 | 0.193 |

**South Pole**

| | 5 day, mean T | 5 day, effective T | 1 day |
|---|---|---|---|
| Mean FAC (m) | 46.8 | 45.5 | 47.3 |
| FAC change (m) | 0.066 | 0.063 | 0.073 |
| mean 2020 FAC-mean 1980 FAC (m) | 0.065 | 0.061 | 0.071 |

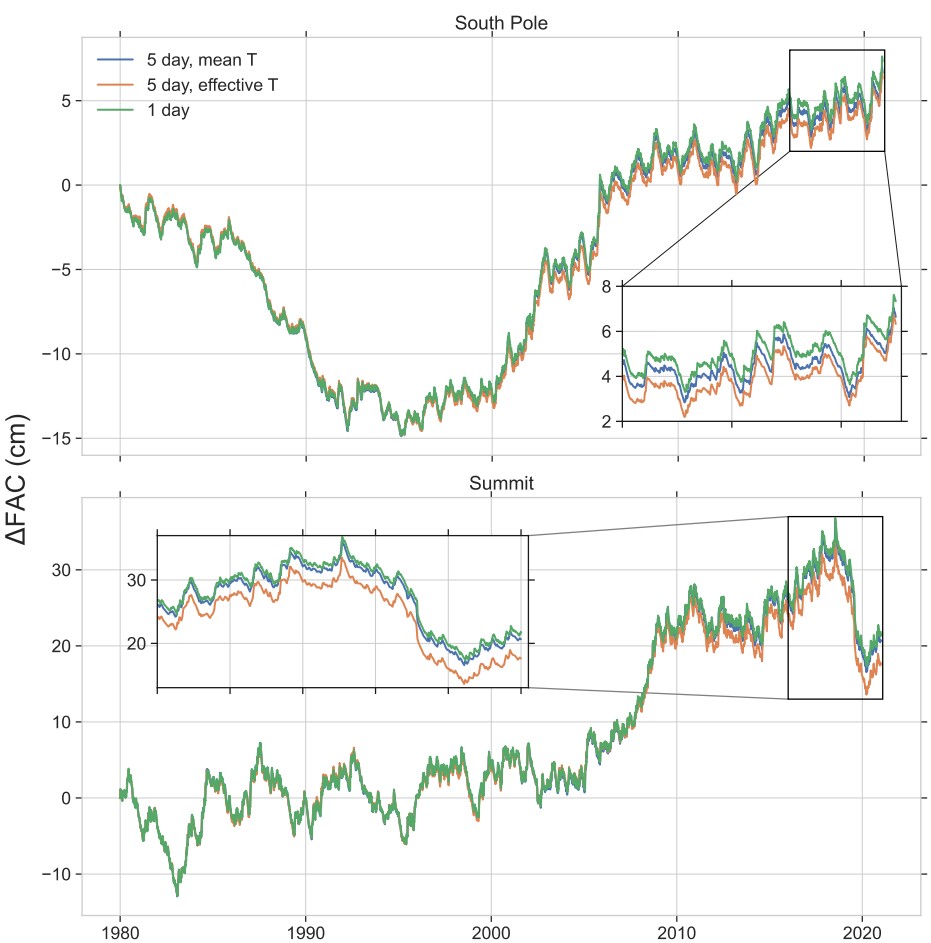

Figure C1: The change in FAC for the duration of the model run for South Pole (top) and Summit (bottom) for each of the 3 model run types.


*Code and data availability.* The NASA GSFC MERRA-2 data are available at https://disc.gsfc.nasa.gov/. The Community Firn Model code is available at https://github.com/UWGlaciology/CommunityFirnModel. The GSFC-FDMv1.2.1 simulations (including firn air content, surface mass balance, and its components) for both ice sheets are available on Zenodo (https://doi.org/10.5281/zenodo.7054574).

*Author contributions.* B.M. led the GSFC-FDM model development including calibration, processing the MERRA-2 climate forcing, and analyzing the output. B.M., T.A.N., H.J.Z., and B.E.S. designed the study and contributed to the manuscript. C.M.S. wrote code for the CFM, ran model simulations, and contributed to the manuscript.

*Competing interests.* We declare no competing interests.

*Acknowledgements.* The GSFC-FDM effort was supported by the NASA ICESat-2 Project Science Office. The authors 920 would like to acknowledge all who have contributed to the Community Firn Model effort and made it a useful resource for the community. We would also like to acknowledge Richard Cullather and Lauren Andrews who provided significant insight into MERRA-2 and the NASA GMAO for the M2R12K data. Finally, we would like to thank Tyler Sutterley and Susheel Adusumilli for providing feedback on the GSFC-FDMv0 and v1 output.

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

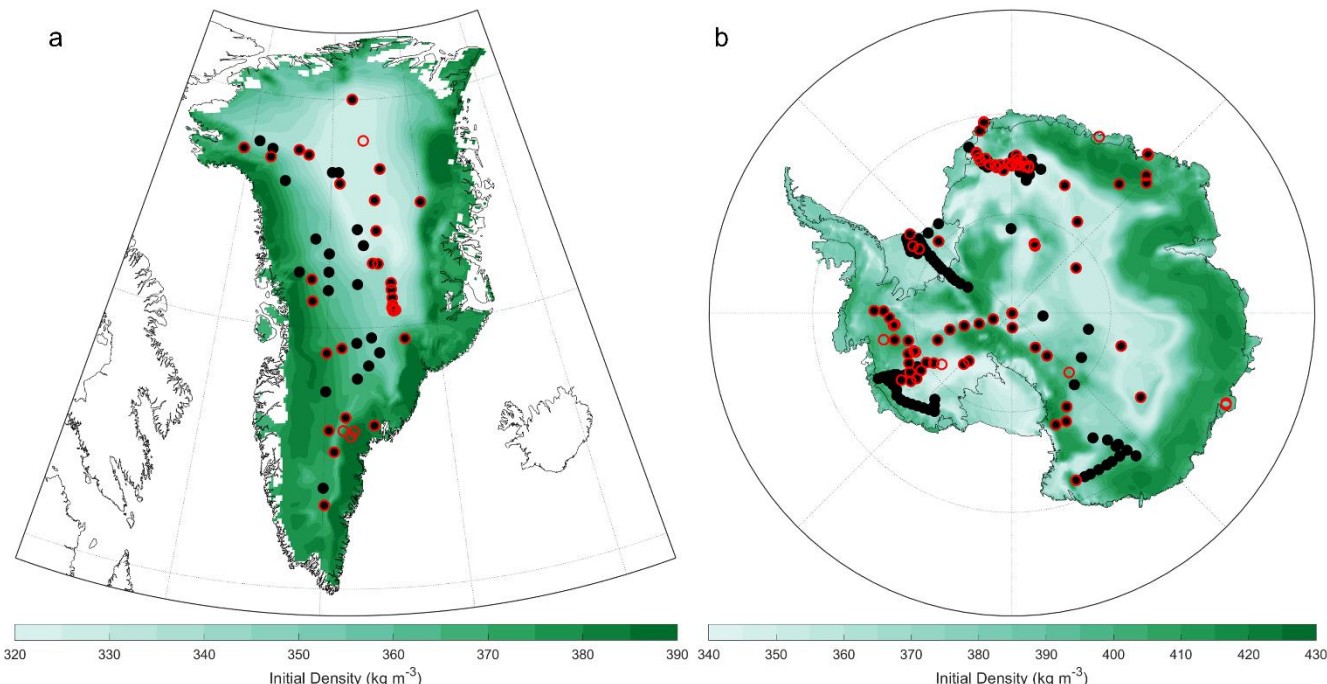

Figure 1. Modeled time-invariant initial density for the (a) Greenland and (b) Antarctic Ice Sheets. These results are based
on MERRA-2 mean surface climate conditions. The solid black circles indicate locations that were used to train and test the
initial density model as well as used in stage 1 calibration (Section 2.1.5), and the red open circles indicate sites used in stage
2 calibration. Note the differences in color scale for Greenland and Antarctica.

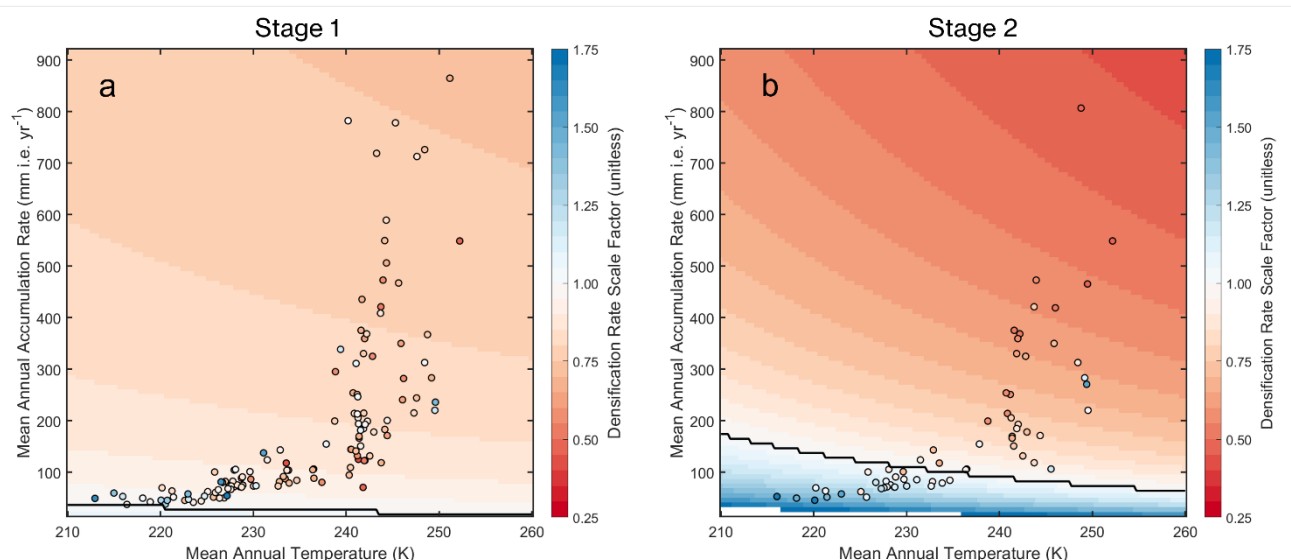

Figure 2. The dry-snow densification calibration coefficients for (a) stage 1, $R_0$, and (b) stage 2, $R_1$ (Sect. 2.1.3; Eqns. 11–12, 15) over a range of mean annual temperatures and snow accumulation rates provide the background color contours. The coefficients derived for each of the calibration sites are plotted as closed circles, colored by their scale factor (i.e., calibration coefficients). The background color contours are derived directly from the calibrated $R_0$, $R_1$ equations. The circles each represent the calibration coefficient derived directly from the depth-density profile at each calibration site. The black contour separates the region of enhanced densification (blue) from the region of reduced densification (red).

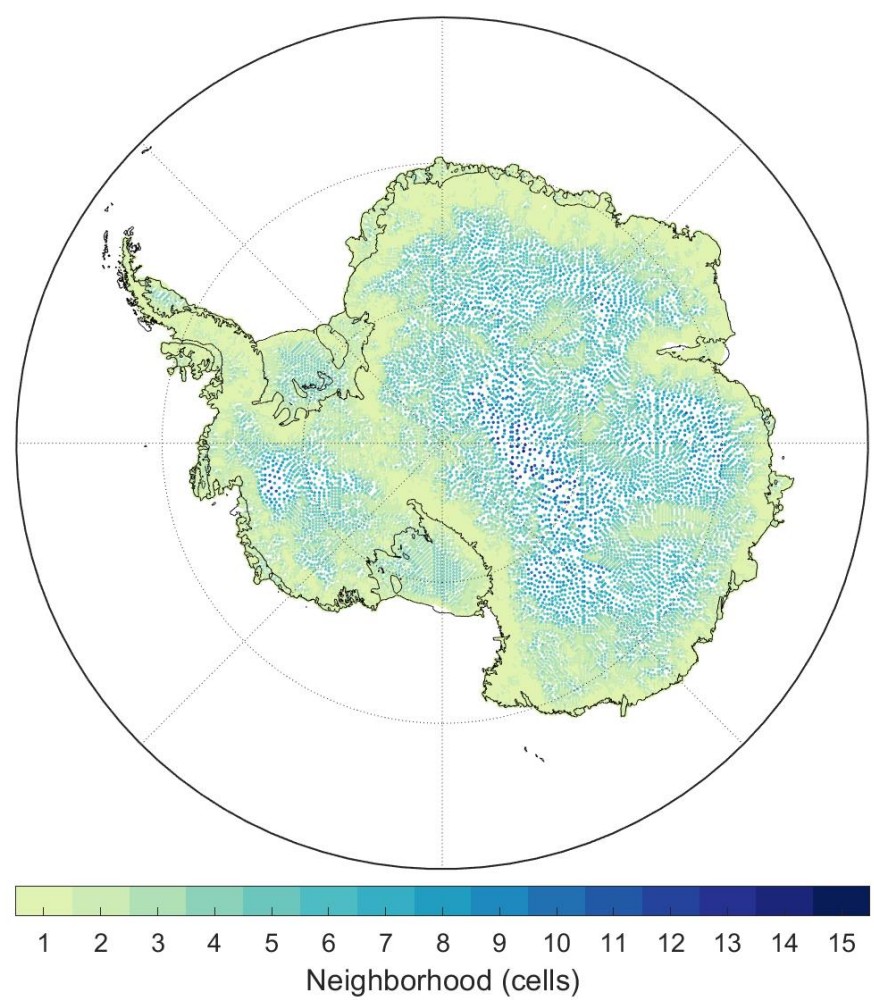

Figure 3. The Antarctic GSFC-FDM simulation locations colored by the representative size of their neighborhood. Darker colors (larger neighborhoods) with more white space (redundant simulations) indicate that the gradients in mean annual climate variables do not vary significantly over short length scales. Paler colors suggest stronger gradients with fewer redundant simulations.

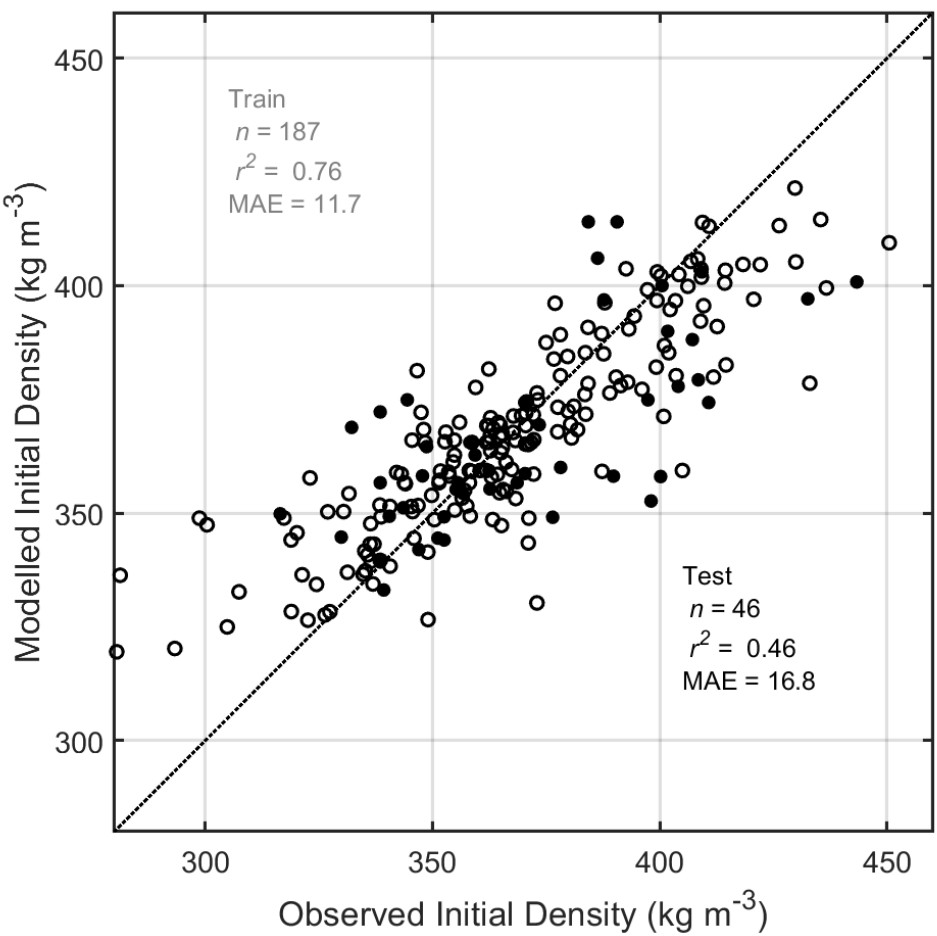

1310

Figure 4. Comparison of observed and modeled initial densities. Solid circles indicate that they were not used in the final model development and represent an independent Testing dataset, whereas open circles represent the training partition used to build the Gaussian Process Regression Model (see Sect. 2.1.5). The statistics in black are in reference to the solid circles only (Testing partition), and those in grey are in reference to the open circles (Training partition).

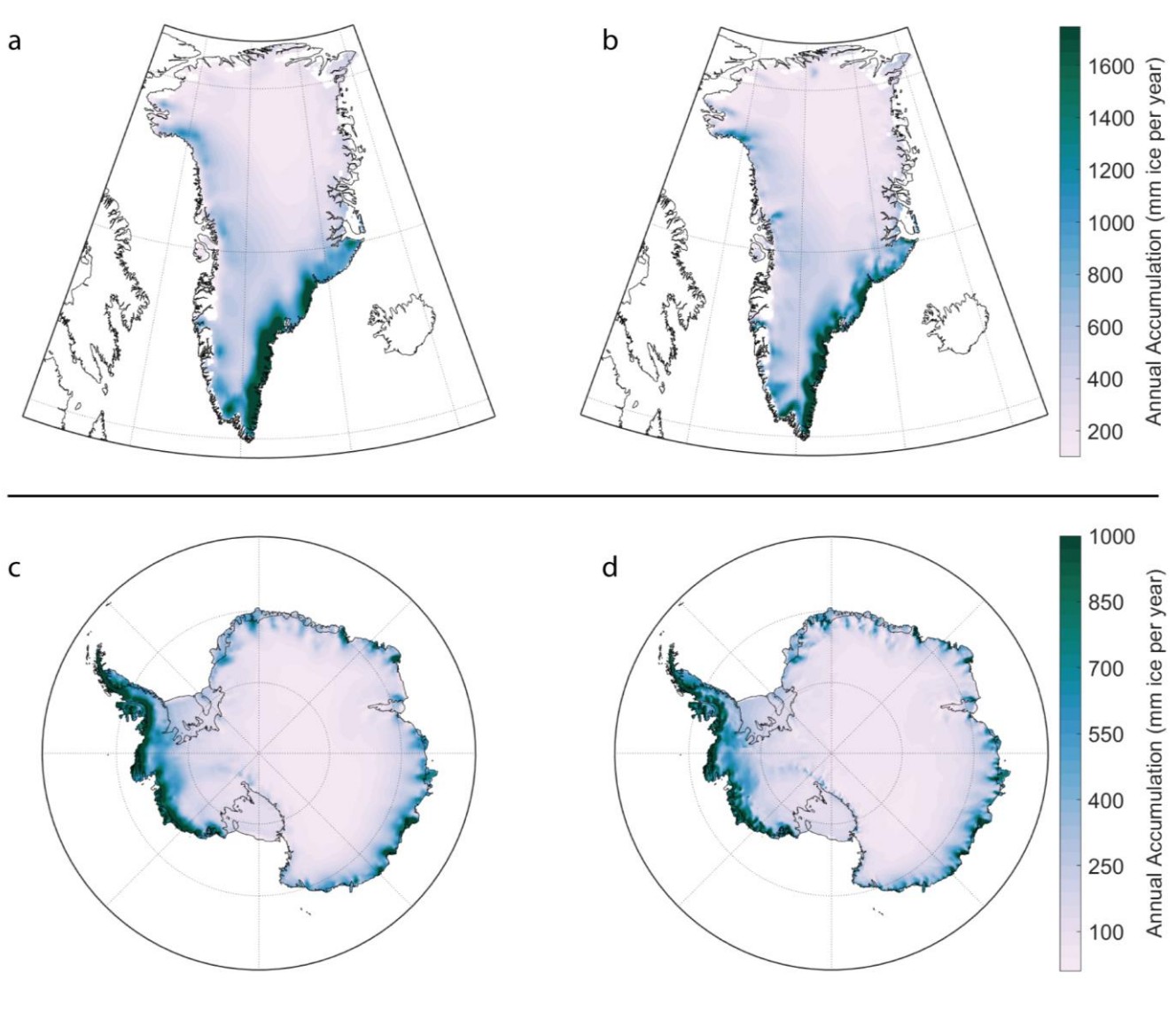

Figure 5. Mean annual net accumulation (snowfall-minus-sublimation) for the Greenland (upper) and Antarctic (lower) ice sheets from MERRA-2 (a,c) and M2R12K (b,d) over their contemporaneous time span (2000–2014). Note the differences in color scale for Greenland and Antarctica.

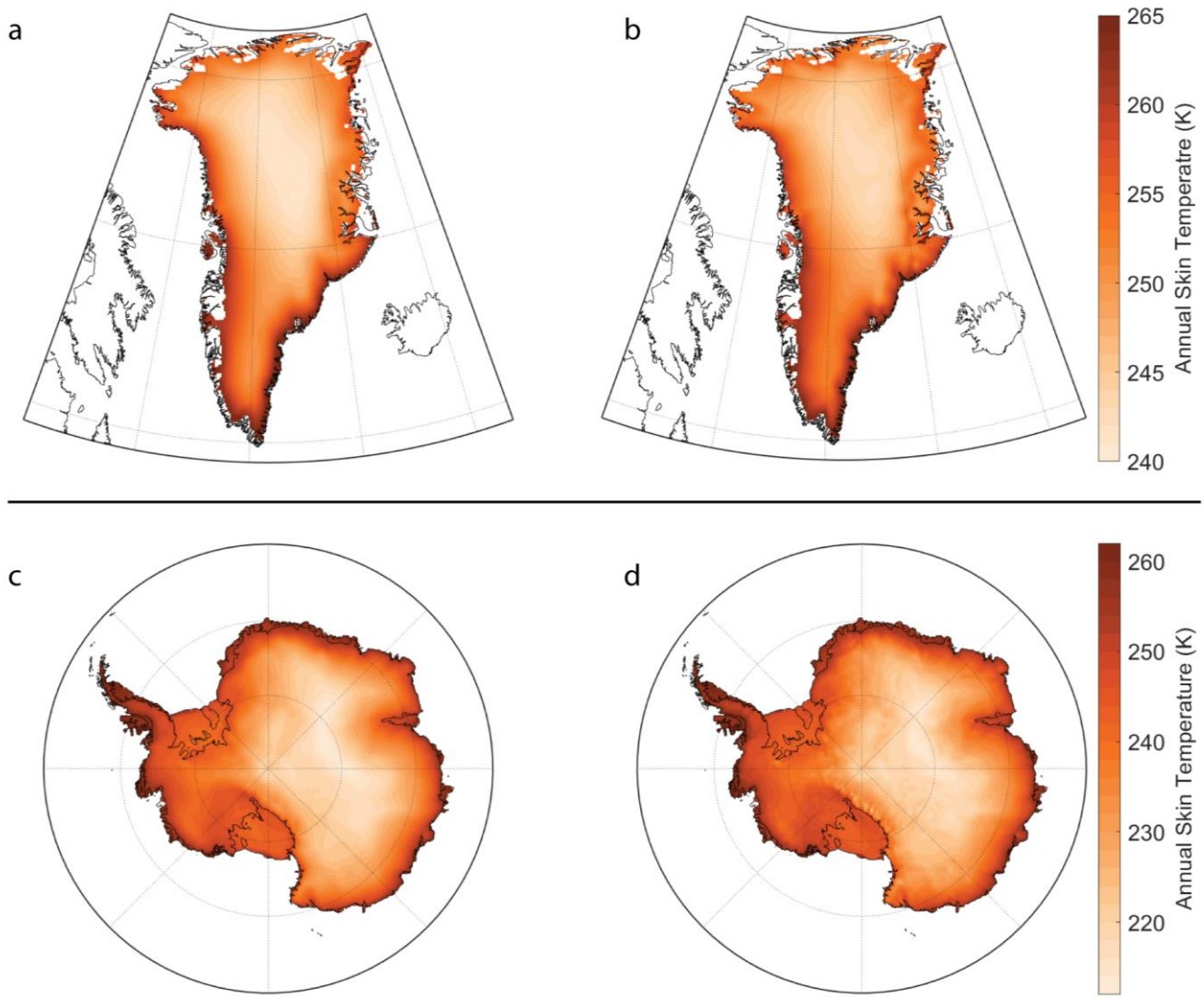

Figure 6. Mean annual skin temperature for the Greenland (upper) and Antarctic (lower) ice sheets from MERRA-2 (a,c) and M2R12K (b,d) over their contemporaneous time span (2000–2014). Note the differences in color scale for Greenland and Antarctica.

1325

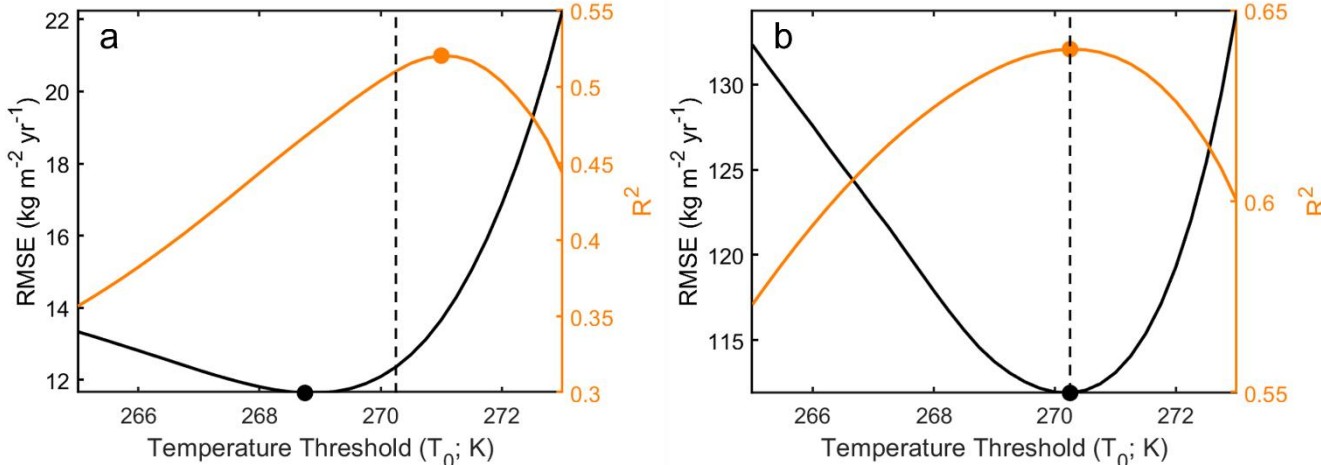

Figure 7. Degree-day model evaluation for (a) Antarctica and (b) Greenland for different temperature thresholds, $T_0$. The orange and black lines represent the mean $r^2$ and $RMSE$ of every grid cell for a given temperature threshold. The orange and black dots indicate the threshold that maximize $r^2$ and minimize $RMSE$, respectively. The dashed line represents the threshold selected for each ice sheet as it maximizes the normalized distance between the two curves and yields the best model according to these evaluators. Note the differences in vertical scales.

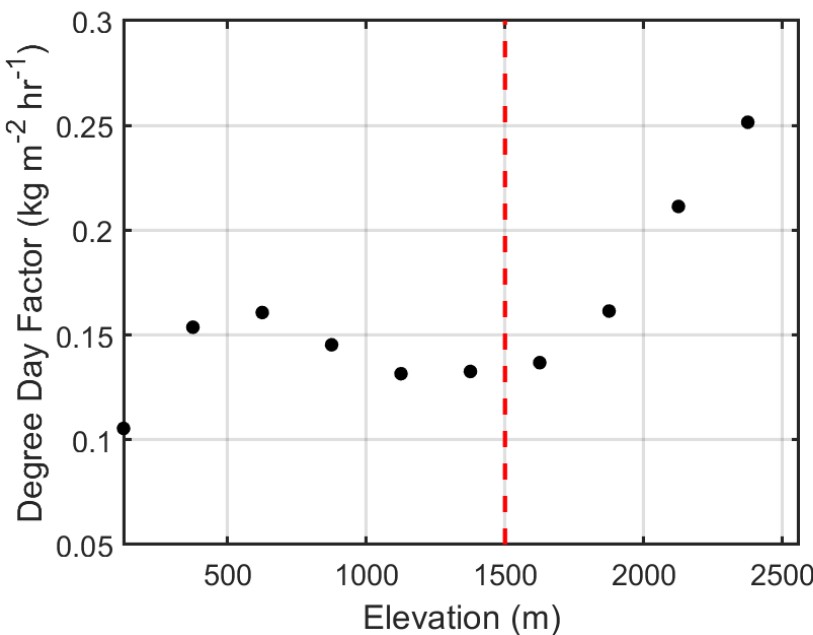

Figure 8.  Mean degree-day factors over the Greenland Ice Sheet binned at 250 m elevation intervals for a temperature threshold, $T_0$, of 270.25 K.  We assume that degree-day factors above the vertical dashed red line at 1500 m are non-physical (i.e., too large).  For all elevations greater than 1500 m, we apply a maximum degree-day factor equal to the mean within the 1250-1500 m elevation bin (0.13 kg m$^{-2}$ hr$^{-1}$ K$^{-1}$).

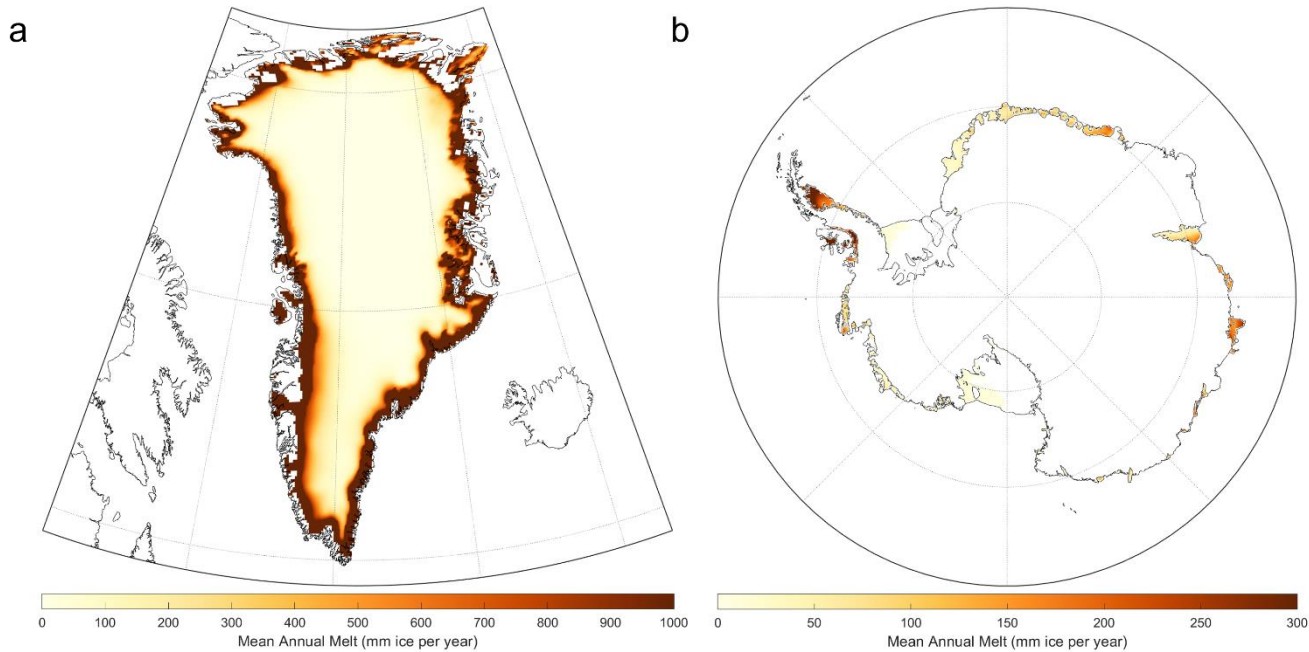

1340

Figure 9. Mean annual meltwater fluxes for the (a) Greenland and (b) Antarctic Ice Sheets based on a degree-day approach.
Note the differences in color scale for Greenland and Antarctica.

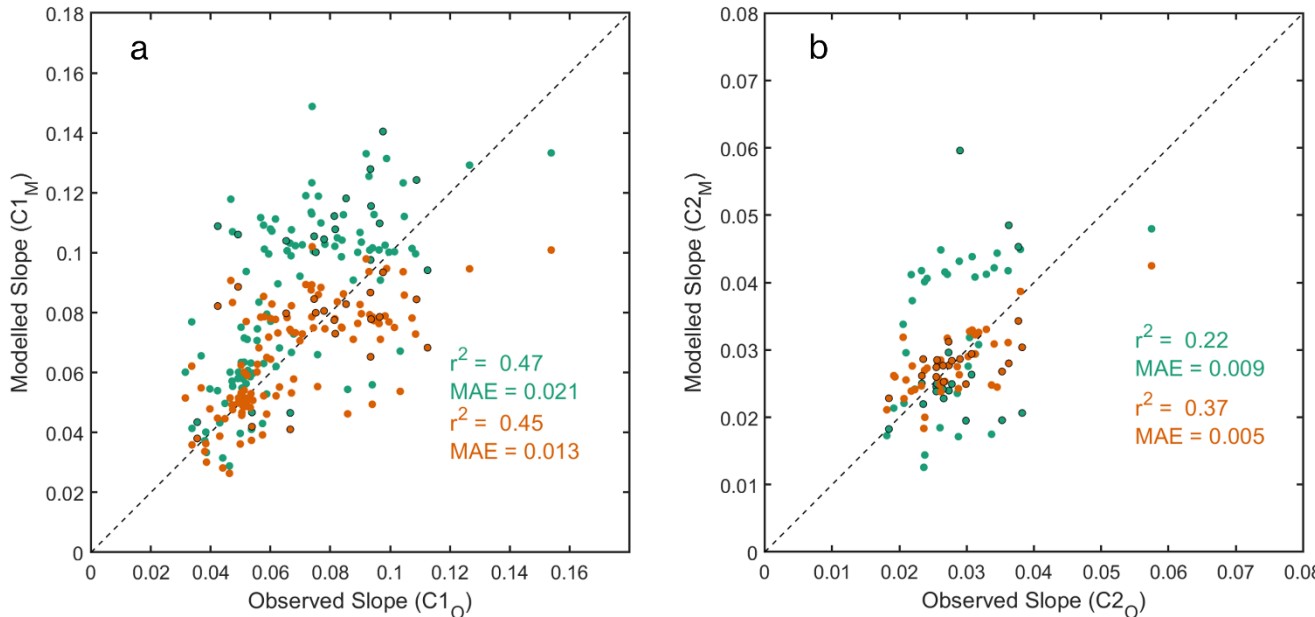

Figure 10. Comparison of observed and modeled slopes of the logarithmic density with depth for (a) stage 1 and (b) stage 2. Green circles reflect comparisons of observed slopes with those from the original densification model (Arthern et al., 2010), and the orange circles compare the same observations after calibration. Circles without an edge color are sites from Antarctica and those with a black edge are from Greenland. Summary statistics are also color coded, where $r^2$ is the coefficient of determination and *MAE* is the mean absolute error. The dashed black line is the 1:1 line. Note the differences in scale.

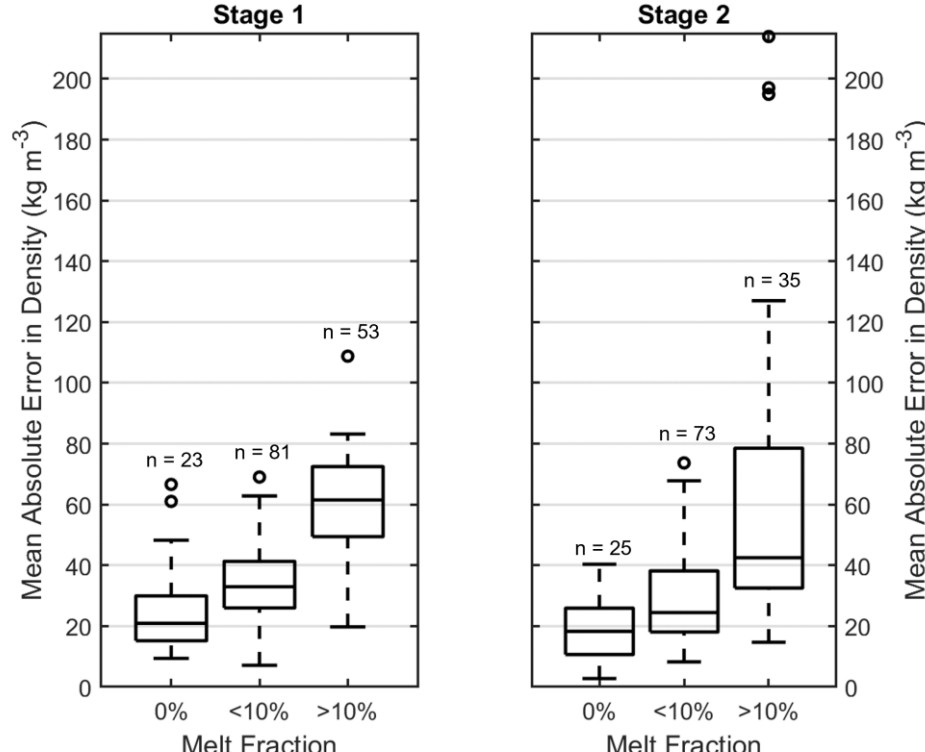

Figure 11. Boxplot of the mean absolute errors in density for stages (a) 1 and (b) 2 of densification in relation to the total melt experienced. The number of observations that make up each distribution is listed above each bar. The black rectangle represents the interquartile range of the mean absolute error while the horizontal line within represents the median. The whiskers show the maximum and minimum with outliers posted as open circles. Melt fraction is defined as the ratio of mean annual melt to mean annual snow accumulation expressed as a percentage.

Table 1. The 1-sigma (standard deviation) uncertainties in various CFM parameters and atmospheric forcing, provide the basis of our uncertainty analysis (Sect. 2.5). We also uniformly sample two assumptions regarding model set up: the thermal conductivity and RCI end year. The perturbation developed sample randomly from a Gaussian distribution with a mean of zero and a standard deviation provided within the table. The values used are based either on analysis within this work or based on other references provided.

| Gaussian Sampling | 1-sigma Uncertainty | Reference (if applicable) |
|---|---|---|
| *CFM parameters* | | |
| Initial Density ($\rho_0$) | 16.8 kg m$^{-3}$ | Mean absolute error of the Test data in Fig. 3 (Sect. 2.1.5) |
| Ec1 | 150 J mol$^{-1}$ | 1-sigma calibration uncertainty (Sect. 2.1.3; Eq. 15) |
| Ec2 | 50 J mol$^{-1}$ | 1-sigma calibration uncertainty (Sect. 2.1.3; Eq. 15) |
| alpha1 | 0.015 | 1-sigma calibration uncertainty (Sect. 2.1.3; Eq. 15) |
| alpha2 | 0.0085 | 1-sigma calibration uncertainty (Sect. 2.1.3; Eq. 15) |
| *Atmospheric variables* | | |
| Snow Accumulation (Sn+Ev) | 10% | Approximate based on SMB analysis (Sect. 3.4) |
| Rain (Ra) | 10% | |
| Melt (Me) | 10% | |
| Skin Temperature | 2 K | Based on Huai et al. (2019); Hearty et al. (2018) |
| **Uniform Sampling** | | |
| Thermal Conductivity | 7 parameterizations[a] | GSFC-FDM v1.2 uses Calonne et al. (2019) |
| RCI End Year | AIS: 2010-2020; GrIS: 1991-2000 | GSFC-FDMv1.2 RCI ends inclusive of 2019 (AIS) and 1995 (GrIS) |
| [a]Choice of 7 parameterizations within the Community Firn Model | | |

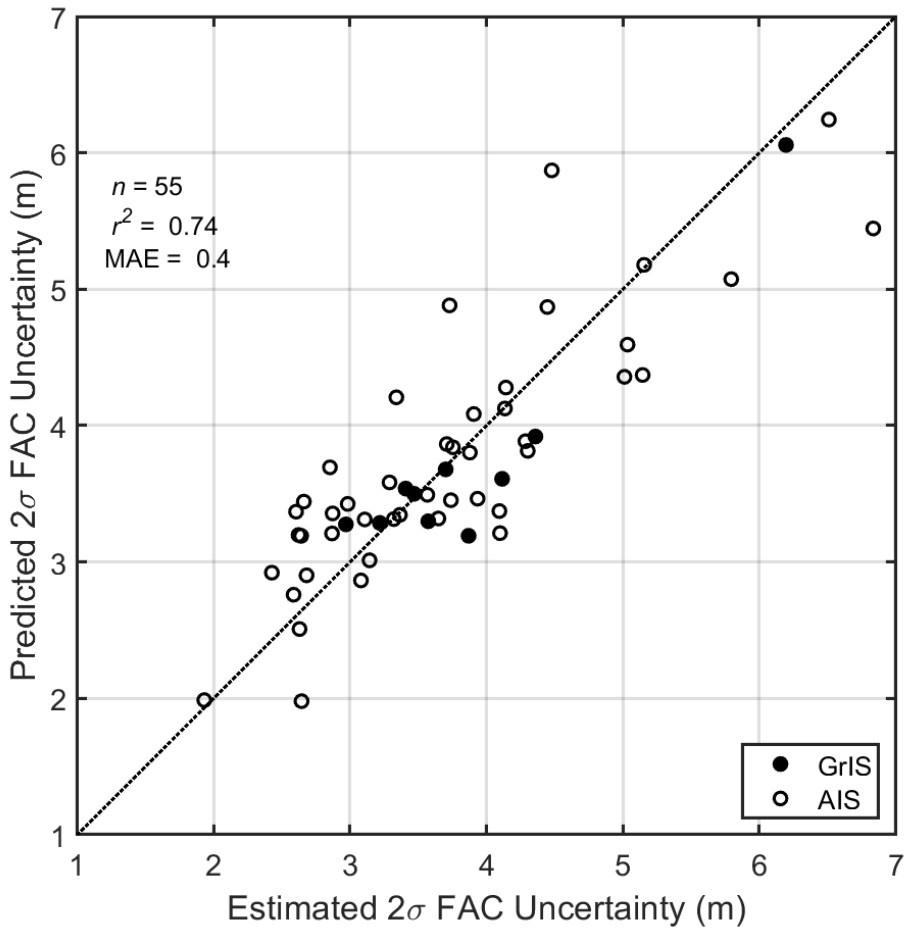


Figure 12. Comparisons of the estimated 2-sigma uncertainty in FAC for both GrIS (solid circles) and AIS (open circles) and the model prediction of the 2-sigma uncertainty derived from Eqn. 20, which only applies to sites with a liquid-to-solid ratio (LSR) between 0 and 1 (Sect. 2.5.1). The uncertainty for those sites that fall outside of those LSR bounds were defined as their standard deviation through time (Eqn. 21). Performance statistics provided apply to the entire population (i.e., GrIS

and AIS combined).

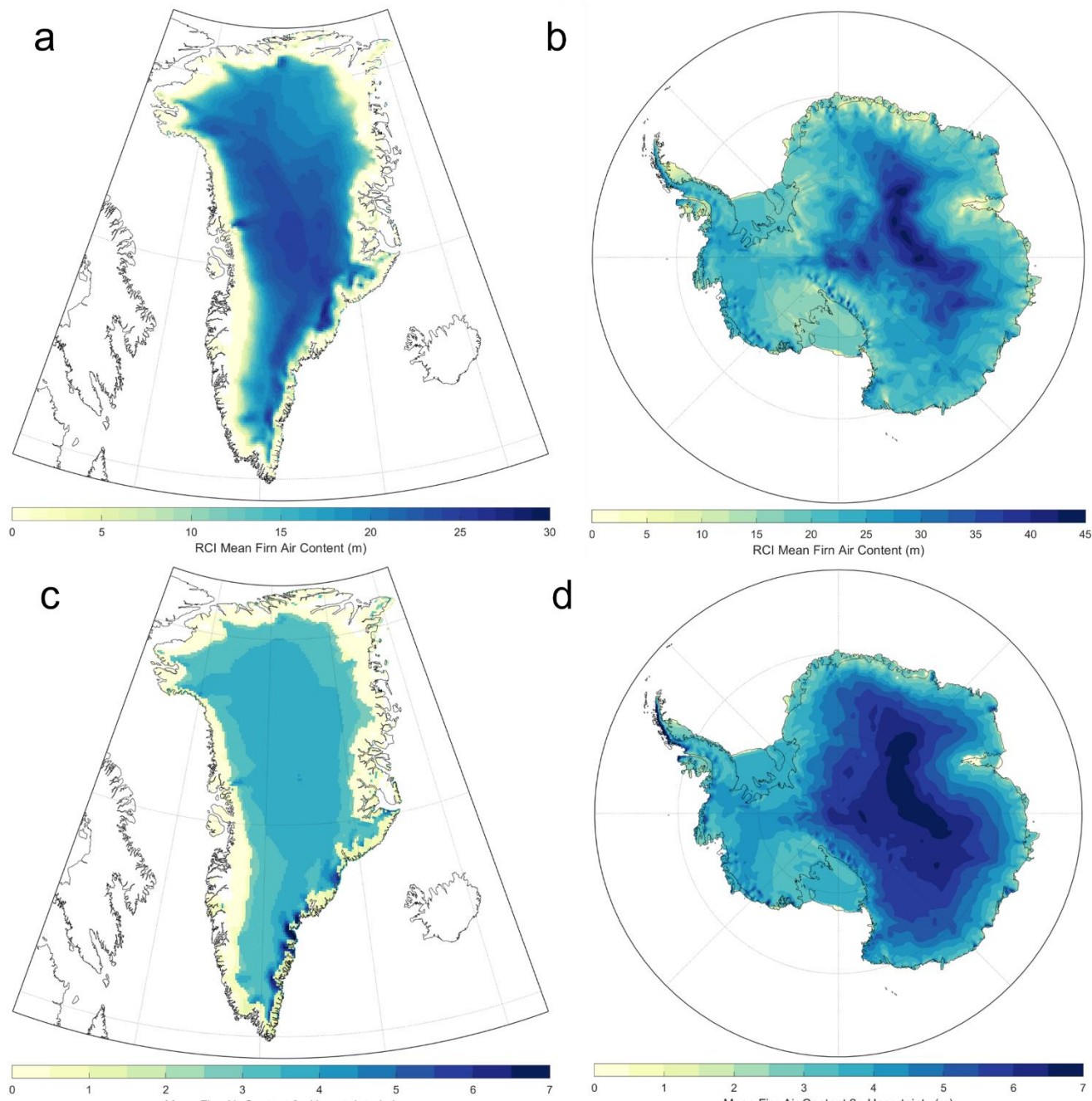

Figure 13. The mean firn air content (FAC) for the (a) Greenland Ice Sheet and (b) the Antarctic Ice Sheet over their respective reference climate intervals and (c,d) the respective 2-sigma uncertainty. Note the differences in color scale for
Greenland and Antarctica and their uncertainties.

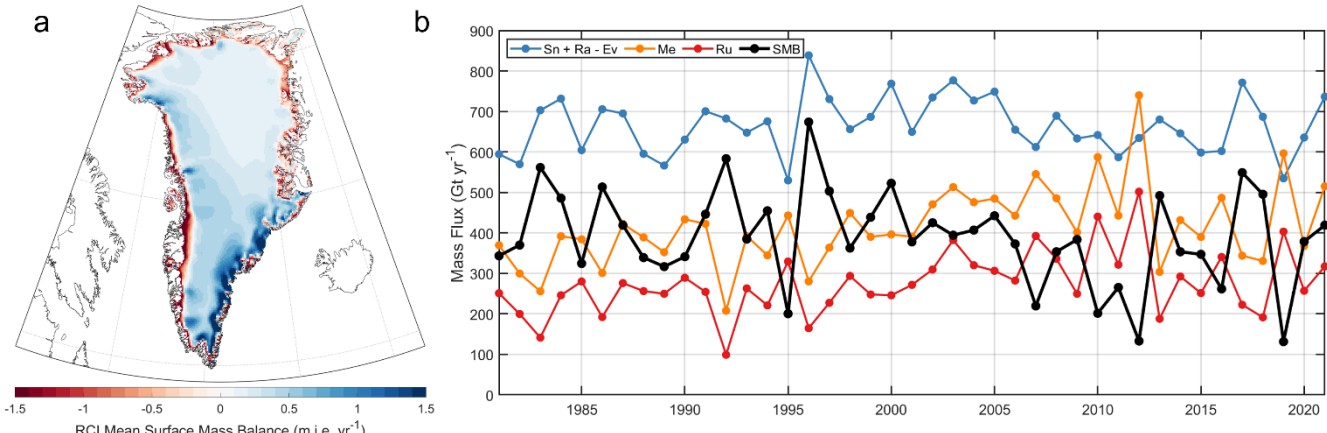

Figure 14 (a) The mean annual surface mass balance of the Greenland Ice Sheet and its peripheral ice over the Reference Climate Interval (RCI: 1980–1995). (b) Time series of annual SMB (black) and its components (see Eqn. 5) for the Greenland Ice Sheet only. Net accumulation (blue) and runoff (red) are combined to make SMB. Meltwater production is in yellow. The annual values are calculated from October through September and are defined by the year of the latter.

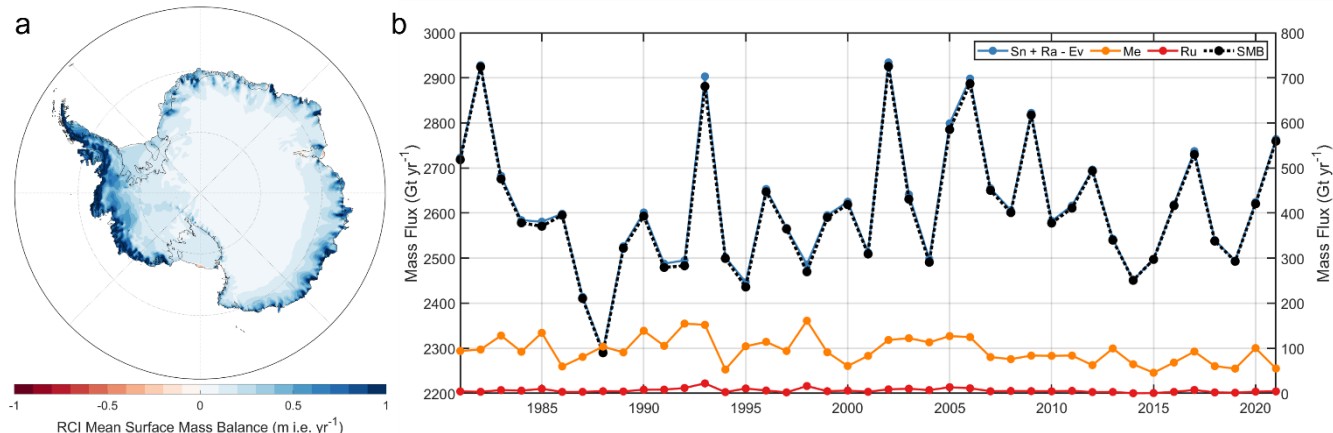

Figure 15. (a) The mean annual surface mass balance of the Antarctic Ice Sheet (AIS; floating and grounded ice) over the Reference Climate Interval (RCI: 1980–2019). (b) Time series of annual SMB and its components (see Eqn. 5) for grounded and floating AIS. SMB (black) and net accumulation (blue) use the left axis and runoff (red) and meltwater production (yellow) use the right axis. Note, the two axes span the same range (800 Gt yr$^{-1}$) but are shifted in magnitude. SMB is presented as a dashed line because of overlap with net accumulation. The annual values are calculated from April through March and are defined by the year of the latter.

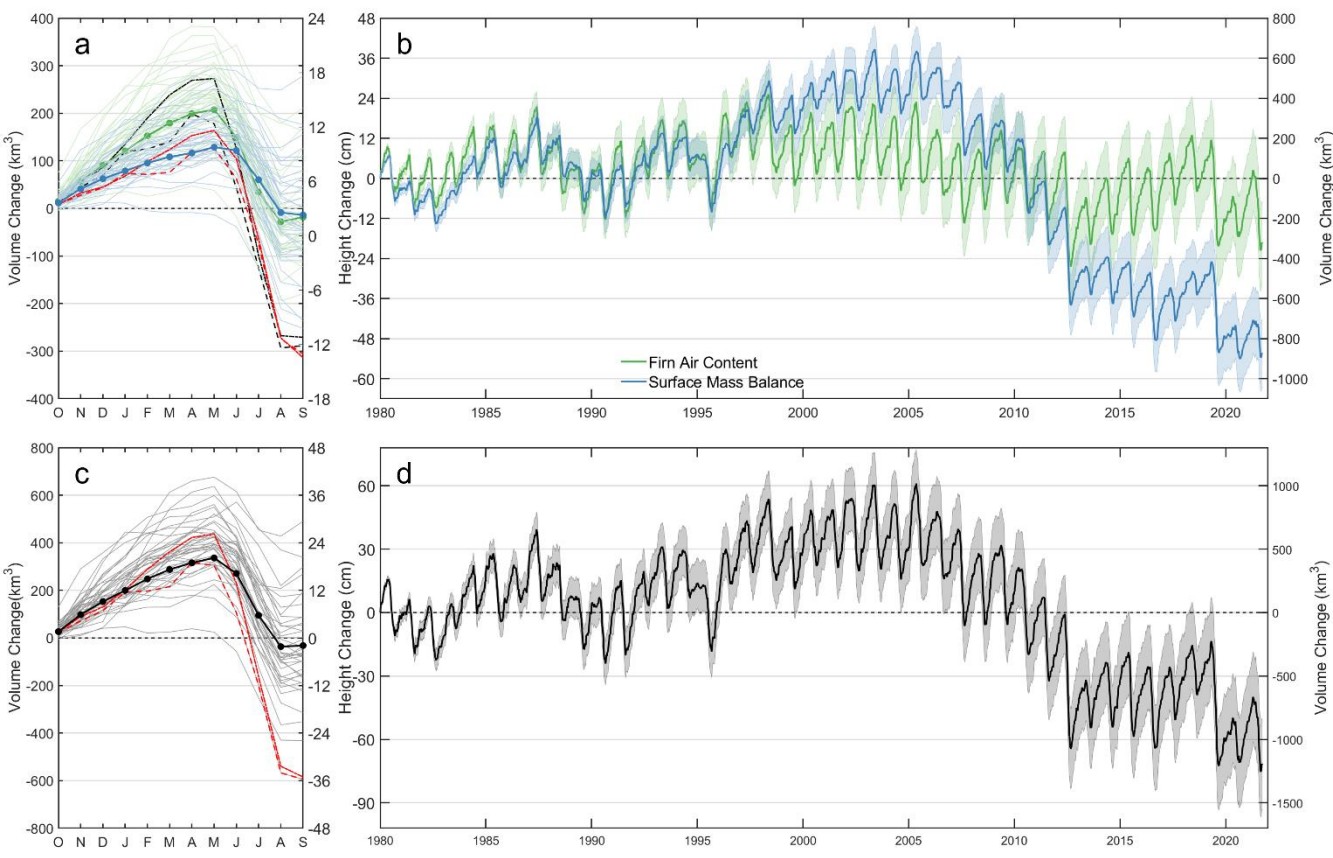


Figure 16. Height and volume change of the Greenland Ice Sheet. (a) Seasonal changes separated into surface mass balance (SMB; blue) and firn air content (FAC; green) components. The thin lines represent each year, and the thick, dotted line represents the mean over the RCI (1980–1996). The solid and dashed lines represent the 2012 and 2019 mass balance years, respectively, where the black lines represent FAC and the red lines represent SMB. (b) A forty-year time series of volume

and height change due to SMB and FAC. (c,d) The FAC and SMB combined height and volume change. The shaded regions represent the cumulative 2-sigma uncertainty in the volume changes, which accounts for spatial and temporal correlation in the time series. Note the differences in scale.

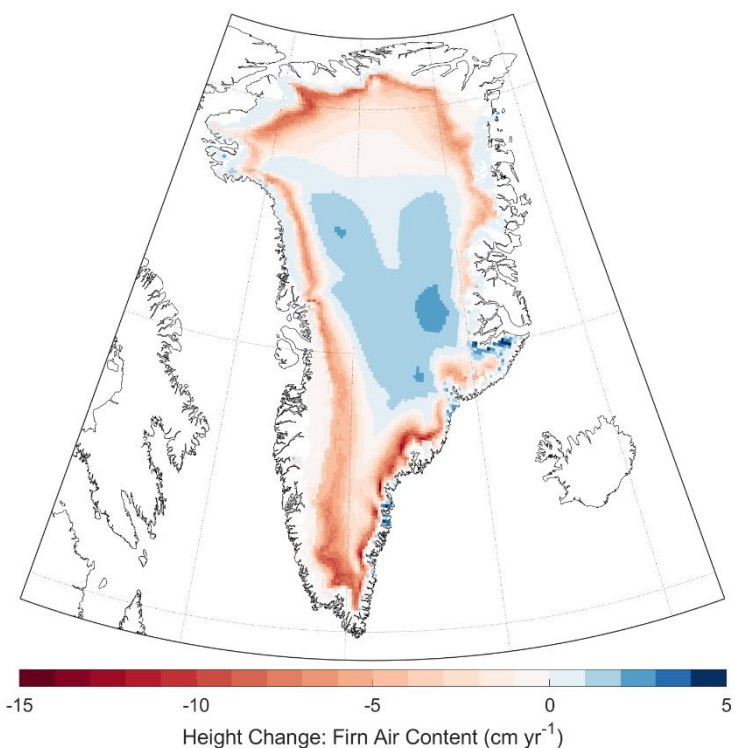

Figure 17. Rate of height change resulting from changes in firn air content over the Greenland Ice Sheet between September
1, 2003 and September 1, 2021.  Note the asymmetric color bar.

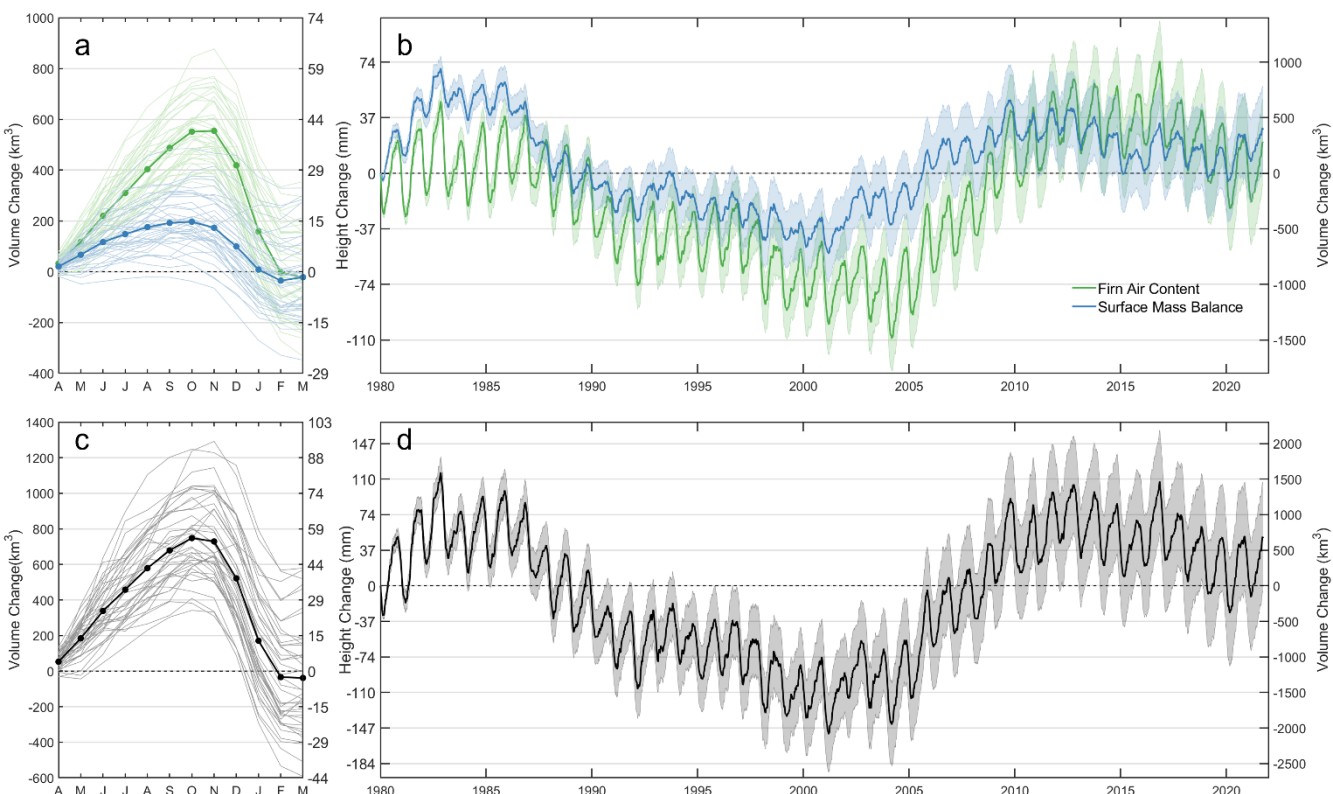

Figure 18. Height and volume change of the Antarctic Ice Sheet (grounded and floating ice). (a) Seasonal changes separated
into surface mass balance (SMB; blue) and firn air content (FAC; green) components. The thin lines represent each year,
and the thick, dotted line represents the mean over the RCI (1980–2019). (b) A forty-year time series of volume and height
change due to SMB and FAC. (c,d) The FAC and SMB combined height and volume change. The shaded regions represent
the cumulative 2-sigma uncertainty in the volume changes, which accounts for spatial and temporal correlation in the time
series. Note the differences in scale.

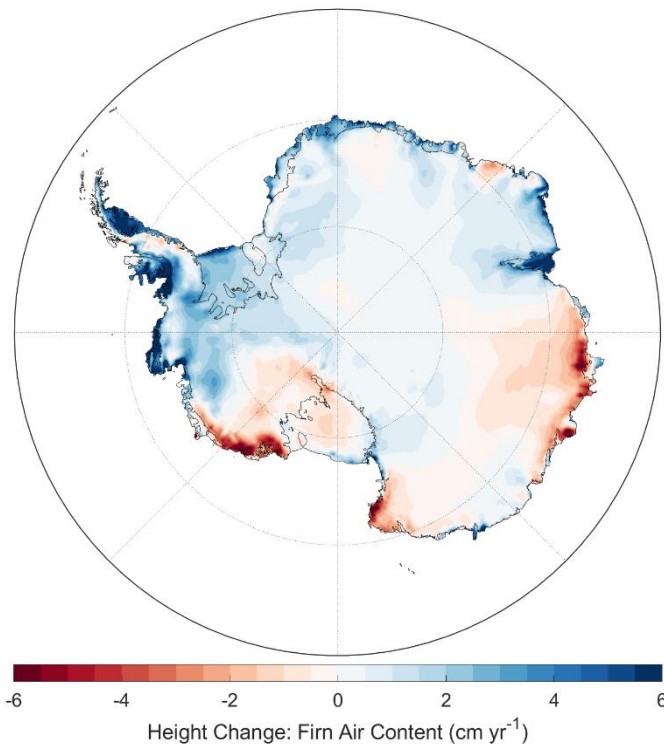

Figure 19. Rate of height change resulting from changes in firn air content over the Antarctic Ice Sheet between March 31, 2003 and March 31, 2021.

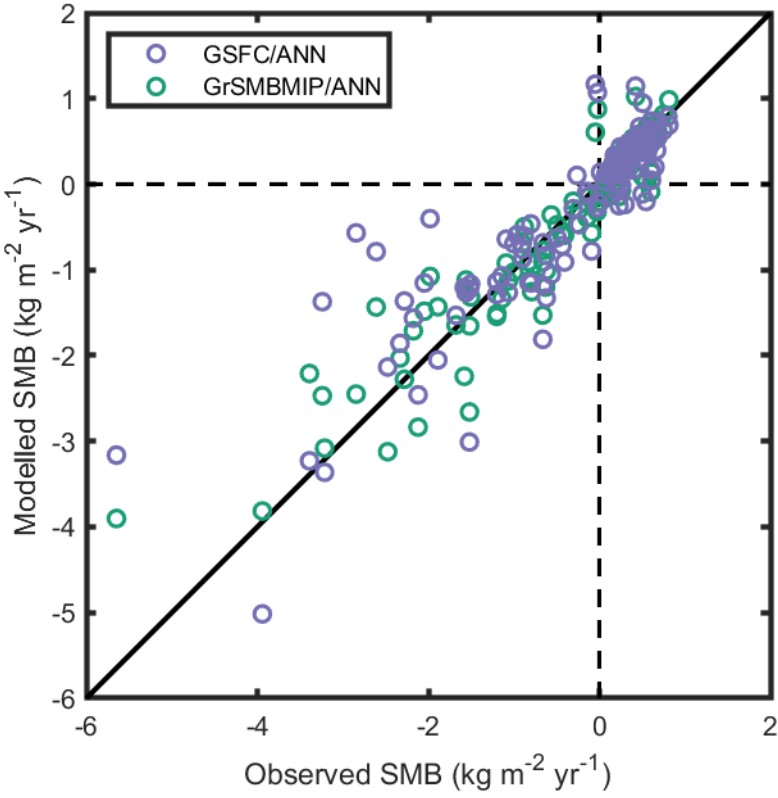


Figure 20. Comparison of SMB observations and the GSFC/ANN (purple) and GrSMBMIP/ANN (green) data sets for the GrIS (see Sect. 2.5.2 for the method).

Table 2. Modelled GrIS SMB performance statistics against N = 312 observations (see Sect. 2.5.2 for description), which are plotted in

Figure 20. The final column presents a comparison (N = 1,688,416) between the ensemble mean annual SMB from GrSMBMIP and the GSFC mean annual SMB interpolated onto the GrSMBMIP grid that is mapped in Figure 21. $N$ is the total number of observations used in the model comparison, $\mu$ is the mean of the observations, $\sigma$ is the standard deviation,

$MB$ is the mean bias (GSFC minus Observation), $RMSE$ is the root mean square error, $r$ is the correlation coefficient.

| | N = 312; μ = 0.032 kg m$^{-2}$ yr$^{-1}$; σ = 0.790 kg m$^{-2}$ yr$^{-1}$ | | | | N = 1,688,416 |
|---|---|---|---|---|---|
| | GSFC | GSFC/ANN | GrSMBMIP/ANN | | GSFC – GrSMBMIP |
| $MB$ (kg m$^{-2}$ yr$^{-1}$) | 0.01 | 0.02 | -0.01 | | 0.0212 |
| $RMSE$ (kg m$^{-2}$ yr$^{-1}$) | 0.45 | 0.35 | 0.24 | | 0.261 |
| $r$ | 0.84 | 0.90 | 0.95 | | 0.87 |

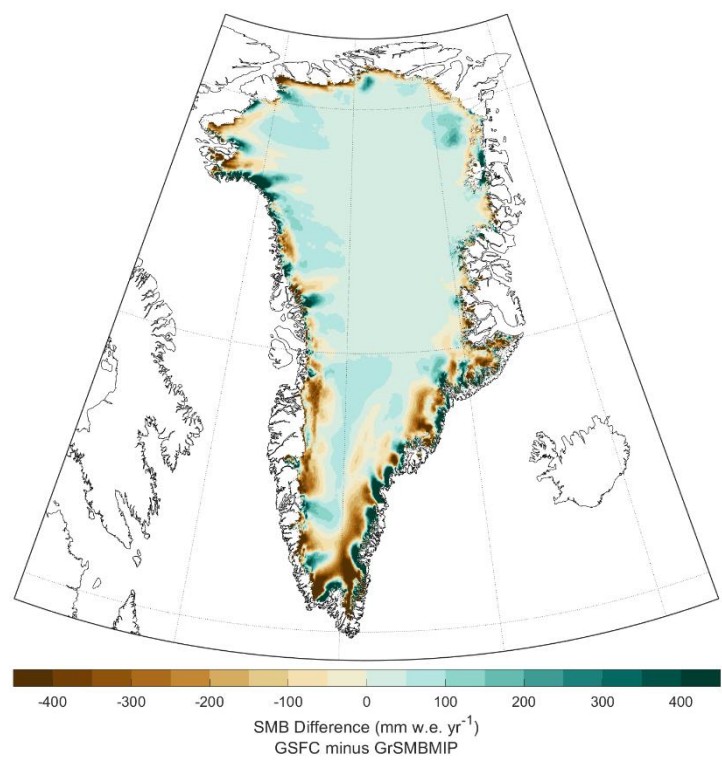

Figure 21. The difference in mean annual SMB between GSFC and the GrSMBMIP ensemble mean.


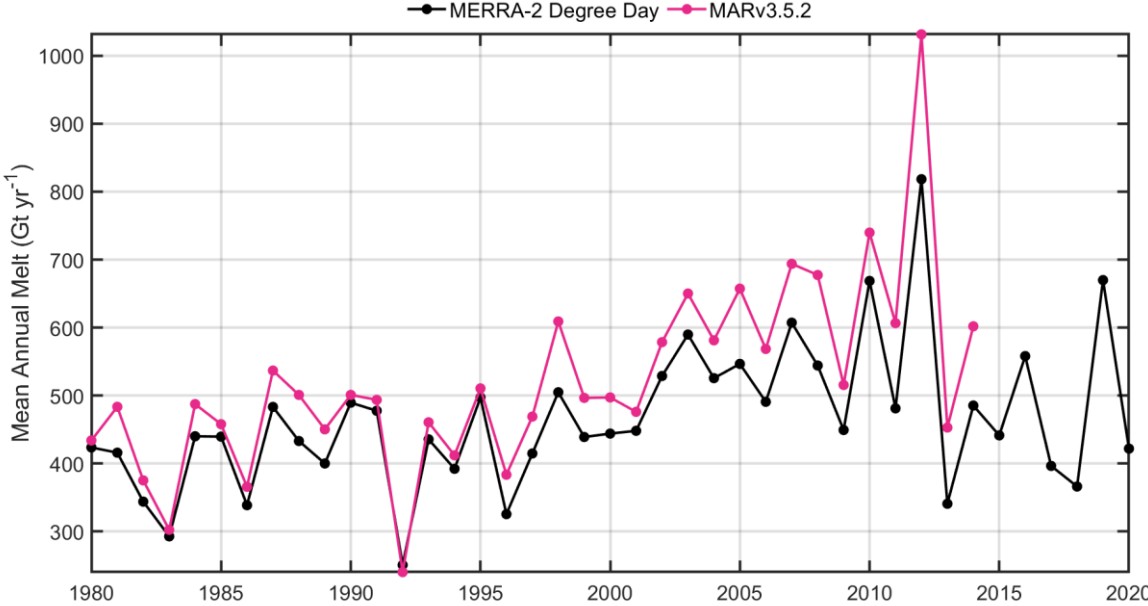

Figure 22. Comparison of annual melt from the MERRA-2 degree-day model (Sect. 2.2.1) and MARv3.5.2, the model used to train the degree-day model (Fettweis et al., 2017).

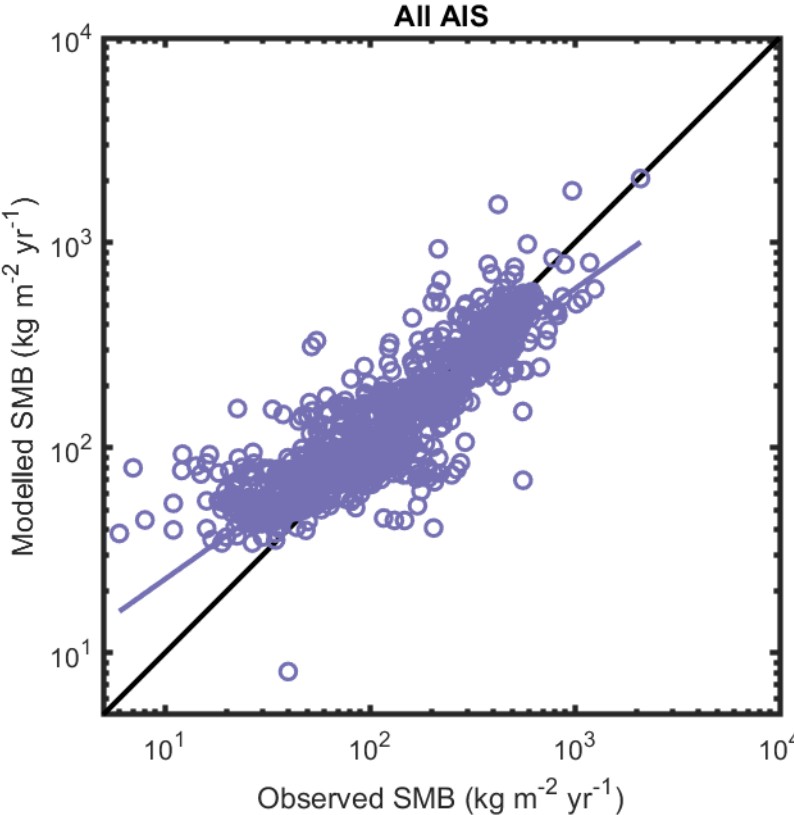

Figure 23. Comparison of N = 1201 observations of SMB with GSFC modelled SMB over the AIS. Note the logarithmic scale.

Table 3. A breakdown of GSFC SMB performance against observations separated into different elevation bands to match Mottram et al. (2021) analysis. $N$ is the total number of observations, $L$ is the number used for the logarithmic analysis, $\mu$ is the mean of the observations, $\sigma$ is the standard deviation, $MB$ is the mean bias (Model minus Observation), $RMSE$ is the root mean square error, $r$ is the correlation coefficient, and $rlog$ is the correlation coefficient of the logarithmic values. Values not in parentheses are our best attempt to match the Mottram et al. (2021) model-observation comparison method. Values

within the parentheses also include SMB values from Medley et al. (2013), which were excluded from the Mottram et al. (2021) analysis.

|  | Shelves | 0–1200m | 1200–2200m | 2200–2800m | 2800–3400m | > 3400m | AIS |
|---|---|---|---|---|---|---|---|
| $N$ | 134 | 187 (211) | 193 (333) | 241 | 179 | 100 | 1037 (1201) |
| $L$ | 132 | 183 (207) | 188 (328) | 241 | 179 | 100 | 1026 (1190) |
| $\mu$ | 200 | 212 (249) | 216 (312) | 90 | 57 | 37 | 140 (183) |
| $\sigma$ | 124 | 238 (247) | 210 (199) | 55 | 28 | 13 | 164 (189) |
| $MB$ | 0 | -5 (-6) | 15 (-14) | 3 | 11 | 17 | 6 (-2) |
| $RMSE$ | 105 | 166 (156) | 131 (112) | 44 | 26 | 19 | 102 (98) |
| $r$ | 0.68 | 0.74 (0.79) | 0.80 (0.83) | 0.61 | 0.52 | 0.68 | 0.80 (0.86) |
| $rlog$ | 0.85 | 0.86 (0.89) | 0.68 (0.80) | 0.59 | 0.51 | 0.62 | 0.84 (0.89) |

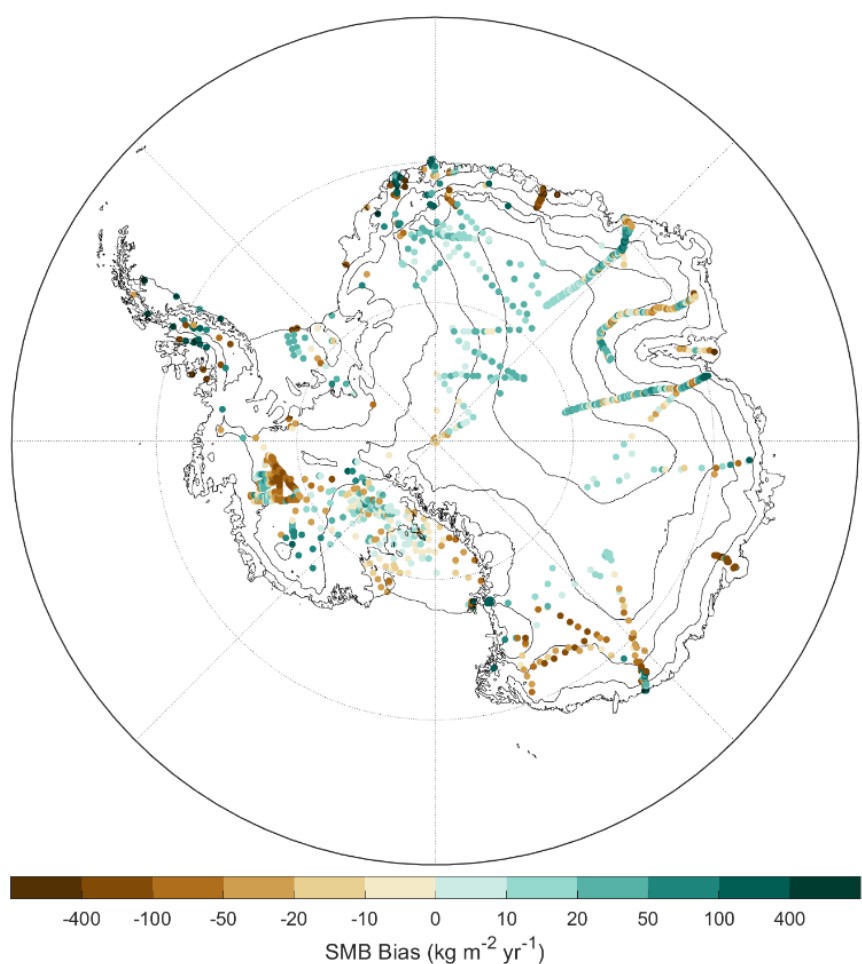

Figure 24. Difference in GSFC SMB and observations over the AIS, including the results from Medley et al. (2013) (see
        Section 2.5.2 for method).

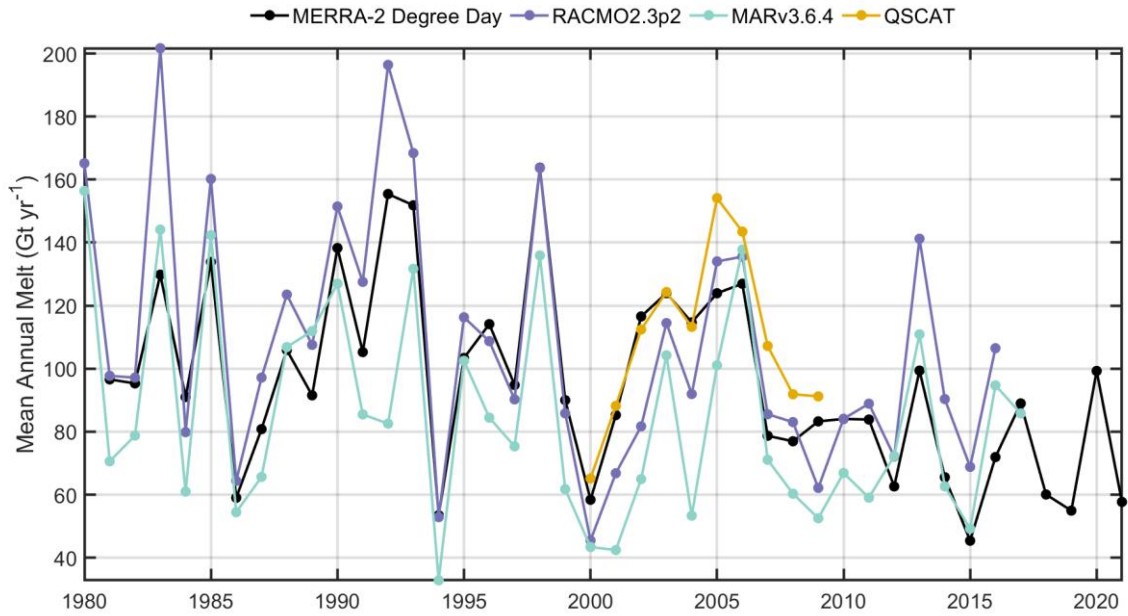

Figure 25. Comparison of annual melt from the MERRA-2 degree day model (Sect. 2.2.1) and Trusel et al. (2013b) QSCAT-
derived surface meltwater fluxes used to calibrate the degree day model, as well as two regional climate models,
RACMO2.3p2 (Van Wessem et al., 2018) and MARv3.6.4 (Agosta et al., 2019).