# Peer review of "Simulations of Firn Processes over the Greenland and Antarctic Ice Sheets: 1980–2021"

_The Cryosphere, 2020_

## Referee Comment (RC1) · Vincent Verjans (Referee) · 6 Nov 2020

This study investigates a modelling approach to estimate ice sheet wide time series of Surface Mass Balance (SMB) and Firn Air Content (FAC) evolution on both the Greenland and Antarctic ice sheets (GrIS and AIS). Using a set of firn cores, the authors recalibrate a firn densification model and establish a new formulation for surface snow density. The MERRA-2 atmospheric reanalysis product is used to compute climatic conditions and to force the firn densification model. The importance of different SMB components as well as their evolution is analysed for both ice sheets. The authors partition ice sheet wide volume changes associated with surface processes between mass and FAC changes, showing that the latter dominates seasonal variability, while the former dominates multi-annual trends.

I believe that this study demonstrates a comprehensive approach to estimate ice sheet SMB and FAC evolution. The modelling framework is robust, well-explained and is undoubtedly a great contribution to the firn modelling community. The use of MERRA-2 is also noteworthy, because this is the first assessment of ice sheet SMB using this product. The authors propose innovative ways to deal with challenges associated with decades-long, large scale simulations, and their results demonstrate an extensive work to perform these simulations. The objective of the study has direct implications for ice sheet mass balance assessments performed via satellite-based altimetry techniques or via the input-output method. As for any model-based study, assumptions and simplifications had to be made. I list my reservations concerning some aspects of the approach in this review. Given the quality of the study, I am confident that a slightly revised version of the manuscript will be accepted for future publication. I realise that modifying the modelling approach to account for any suggestion from the reviewers would subsequently require to re-run the simulations. I do not consider this necessary to address the reservations I raise. However, I expect the authors to provide strong justifications or to better acknowledge limitations with respect to my reservations in the revised manuscript.

I have separated my review in Specific comments requiring more clarity in the manuscript and/or strong justifications from the authors in their response, and Technical comments related to the structure of the text. Despite my numerous comments, I strongly encourage the authors to re-submit the manuscript after the revisions have been made.

Specific Comments
1) The surface density formulation ($\rho_0$)
Constraining surface snow density is both important and difficult. The authors took the approach of using a very large range of possible predictor variables and a single formulation for both ice sheets. My first concern relates to the neglect of melt sites. The formulation is calibrated only to the set of firn cores previously selected for the dry firn compaction calibration. However, surface snow density values are also used in the percolation and ablation areas of the GrIS, which are rejected from the calibration selection but account for a predominant part of the GrIS total area (see Figure 1). The estimated $\rho_0$ values are particularly low in the GrIS southwest, which is the highest melt area. This disagrees with observational studies (e.g. Machguth et al., 2016; Fausto et al., 2018). Why did the authors decide to use the same selection criteria for the surface snow density calibration as for the dry firn compaction calibration? Surface snow density should be accurately represented, including for higher melt areas. Underestimating surface snow density there can lead to overestimation of the firn retention capacity, and thus lower runoff values. I believe that this largely explains why the GrIS-wide runoff estimate of this study is on the lower end of the GrIS SMB intercomparison (Fettweis et al., 2020). A similar effect can be perceived on the Antarctic Peninsula, which has $\rho_0$ values in the lowest range of the AIS. My second concern relates to the use of the northward wind speed in the formulation. This climatic variable has very different physical meanings in various areas of the AIS (i.e. wind from the inner continent or from the shore), and even more between AIS and GrIS. I am thus skeptical about the physical sense to include it in the parameterisation. Furthermore, given the few GrIS cores used in the calibration, I believe that including the northward wind speed may undermine the validity of the parameterisation on the GrIS.
Finally, the model performs worse for the lowest $\rho_0$ values. It should be mentioned that these correspond to surface densities most critical for FAC calculations because firn layers of low density have high FAC values.

2) The use of the effective temperature ($T_E$)
It is well-explained in Section 2.1.3 that stage-1 firn compaction rates cannot be assumed to depend simply on the mean annual temperature. Computing an effective temperature is a novel approach that accounts for the impact of high temperatures on compaction rates, but that leads to other potential problems. Firstly, I think that more details should be given about Equations (15) and (16). In Eq. (16), is $T_E$ meant to be $\overline{T_E}$ ? It is stated that Eqs. (15) and (16) are only used for stage-1 densification, thus should $E_c$ be $E_{c_0}$ ? Is $\bar{k}$ the average of all hourly $k$ values of the climatic forcing? Secondly, several shortcomings related to the use of $T_E$ should be mentioned. As far as I understand, $k$ represents the compaction rate of a firn layer at the skin temperature. The temperature signal is dampened in depth and $k$ thus overestimates the compaction rate of the whole stage 1 firn. In turn, this leads to an overestimation of the effective

temperature. Also, the use of $T_E$ as temperature forcing in the temporal coarsening of the climate input can have a significant impact on the firn temperature profile in the simulations. In the CFM, a newly deposited layer has its temperature set at the temperature of the time step (thus $T_E$ in this approach). Subsequently, the layer is buried and carries this temperature signal, causing advective heating. I guess that $T_E$ is significantly higher than the mean skin temperature for a large majority of the grid cells and coarsened time steps. As such, advective heating is significantly greater, which in turn leads to overestimated firn compaction rates. I do not know how much the 5-, 10- and 20-days $T_E$ values differ from their mean temperature and thus how strong this bias in compaction rates is.

3) Densification Model Calibration (Section 2.1.3)
3a) The assumption that the "*logarithm of the firn density profile with depth is approximately linear*" was stated by Herron and Langway (1980) but never mathematically verified. I would appreciate if the authors could validate their use of this assumption for their calibration process. I suggest that summary statistics are provided to evaluate this validity. It should be straightforward to compare point density measurements of the firn core dataset to corresponding density values taken from the estimated log-profile (and re-converted to kg m$^{-3}$ units). I ask the authors to provide RMSE and R$^2$ values of the fit for both stage-1 measurements and stage-2 measurements. These statistics should be computed before removal of the measurements via the iterative 3-sigma edit. I would welcome any valid alternative way to validate this assumption put forward by the authors.
3b) I am not sure to understand how "*the firn density measurements and model output are binned into half-meter depth increments to obtain similar sampling intervals before slopes are estimated*". Are all measurements (resp. model outputs) averaged in intervals of 0.5 m and the slopes computed on these 0.5 m averaged density points?
3c) There are mathematical inconsistencies when substituting Eq. (11) in Eq. (18). The final formulation of the firn model assumes:
$$\bar{b}^{\beta_0} \times \dot{b} = \dot{b}^{1+\beta_0}$$
$$exp\left(\frac{-60000}{RT}\right) \times exp\left(\frac{-E_{C_0}}{R\bar{T}_E}\right) = exp\left(\frac{-60000-E_{C_0}}{RT}\right)$$
Both these assumptions are mathematically wrong. I understand that these are made for practical purposes, but they should at least be mentioned in the manuscript. Similar concerns hold for the stage-2 formulation (substituting Eq. (12) in Eq. (19)), even though they are less critical because $\bar{b} \approx \dot{b}$ and $\bar{T} \approx T$ in deeper firn.
3d) Why do the authors reject sites falling in a same grid cell? And how do they choose which depth-density profile to exclude? They could very well compute two different pairs $(R_0, R_1)$ within a single grid cell. This would illustrate the natural small-scale heterogeneity of firn structure.
3e) Why is the intercept forced to 0 in the regression? Is it to make the estimation of the parameters well-determined?
3f) Is $E_0$ exactly 0? Or was it sufficiently close to 0 to set it equal to 0?

4) The degree-day model (Section 2.2.1)
4a) In their study, van den Broeke et al. (2010) used $T_0<273.15$ K with the justification that: "*on days with a negative average $T_{2m}$, the method predicts zero melt if $T_0 = 273.15$ K is used, while melt may have occurred during a short period. This problem may be avoided by applying the method to hourly $T_{2m}$ values or by applying a lower value for $T_0$*". Because hourly $T_{2m}$ values are used in this study, citing van den Broeke et al. (2010) to justify the choice of $T_0<273.15$ K is inappropriate. Also, this raises the question of the physical sense of using $T_0<273.15$ K. Should the calibration not rather fix $T_0=273.15$ K and tune $DDF$ only?
4b) The selection of the best $T_0$ threshold depends on maximizing $r^2$ and minimizing $RMSE$. However, it is not explained which particular $r^2$ and $RMSE$ are considered since many grid cells are used. It is only in the caption of Figure 7 that the authors mention that "*the median $r^2$ and RMSE of every grid cell*" are used. This should be stated in the main text. Also, I wonder about the relevance of the choice of the median values. Most of the grid cells have very low melt rates. It is not important to capture the low melt rates with great accuracy. It is much more important to capture melt rates of the grid cells in high melt areas. Thus, why choosing the median?
4c) I ask the authors to provide the final ranges of $DDF$ values used for the GrIS and for the AIS.

5) Wet firn compaction
Simulating wet firn compaction and liquid water processes is a major weakness of firn models. I certainly do not blame the authors for this and addressing this shortcoming is not the subject of this study. I appreciate that the results of the compaction model are also evaluated at high melt sites (Figure 8). I think it is important to also provide the bias of the compaction model at the zero-, moderate- and high-melt sites to know if the model tends to overestimate/underestimate densities in such melt conditions. Also, I believe that the text should remind the reader in the Discussion section about the wet compaction shortcoming and that it can have a significant impact on FAC results

for the GrIS and ice shelves. As stated by the authors themselves, only a limited area of the GrIS satisfies the criteria used for the dry firn compaction calibration. This implies that firn compaction can only be expected to be well represented in that limited area.

6) The Reference Climate Interval (RCI)
In Section 3.2, the authors are perfectly right: "*The RCI is ideally representative of long-term steady-state conditions*". However, when they evaluate the assumption of their RCI choice, they only assess the "*steady-state*" aspect and neglect the "*long-term*" aspect. Indeed, the RCI should show no trend in any climatic field and this is thoroughly investigated for both the GrIS and the AIS RCIs. But the RCI should also be representative of the climate under which the firn column was established (i.e. of the past centuries in AIS). Some studies contradict the assumption that 1980-2019 is representative of the long-term AIS climate and that there are very likely some pronounced regional trends (e.g. Medley and Thomas, 2019). Similarly, in Greenland, regional long-term trends may exist (e.g. Hanna et al., 2011). This impacts the spin-up process because the initial firn column should be in equilibrium with the past climate. Again, such difficulties are inherent to firn simulations because reliable climate forcing covers only the recent decades. Thus, one cannot expect the authors to have a solution to this particular problem. But I would like this limitation to be mentioned in the manuscript, as well as its potential impacts on the findings.

7) Comparing SMB and FAC components
Seasonal variability in height is shown to be driven more by FAC than by snow mass. However, FAC gain/loss is essentially governed by snowfall amounts. For example, if we assume 1 m i.e. accumulation over a given month and a surface snow density of 300 kg m$^{-3}$, the corresponding FAC gain is ~2 m. In other words, without considering compaction, one should expect FAC variability to be around 2 times larger than SMB variability. The values found in this study are around 3 and show the additional effect of seasonally varying compaction rates. But the reader should be explicitly informed about the direct dependence of FAC variability on the SMB variability and, as a consequence, about its expected larger magnitude. Most of the change in FAC is not simulated by firn densification models but is determined by the climatic forcing. In regards to this aspect, I find the statement in the Conclusion line 505 misleading ("*Thus, determination of seasonal mass change using satellite altimetry requires a substantial FAC correction, highlighting the importance of firn densification models, especially when investigating shorter intervals of change as not being mindful of the seasonal cycles of SMB and FAC can generate large biases.*").

Technical comments
p.2 l.39: "*few hundred meters*", I am not sure that the firn column can be that deep (e.g. Ligtenberg et al., 2011), please provide a reference.
p.2 l.53: Make sure to consistently use either "solid earth" or "solid-earth" throughout the manuscript.
p.3 l.71: I suggest adding a statement underlining the sensitivity of Eq. (3), such as "Mass balance estimates are highly sensitive to small errors in the height change measurements and in the modelled firn signal."
p.3 l.72: I think that "*Variable rates of the height change due to compaction*" should be replaced by "Height changes due to variable rates of compaction".
p.3 l.73-76: I suggest not introducing the variables $dh_c/dt$, $dh_m/dt$ and $dh_a/dt$ because these are not used in the remainder of the manuscript.
p.3 l.80: SMB and FAC appear in the wrong order: "*air thickness and the thickness of ice: surface mass balance (SMB) and firn air content (FAC), respectively*"
p.3 l.86: Add a comma: "(…) mass fluxes at the surface, including (…)"
p.4 Eq. (6): There is a typo in the equation, which should have $\rho_i$ in the denominator: $FAC = \int_0^{z_{\rho_i}} \frac{\rho_i - \rho(z)}{\rho_i}\, dz$
p.4 l.110: Why do the authors simulate grain-size evolution?
p.5 l.141: "*subset of 256 published firn depth-density profiles*" The authors should provide a little more detail about the dataset of firn cores used in this study. I suppose that the SUMup dataset is used. If this is the case, the authors should cite the work of Koenig and Montgomery (2019) (https://doi.org/10.18739/A26D5PB2S). If other datasets are used, they should also be cited. All the references can be provided in the section *Code and data availability* or in the section *Acknowledgements*.
p.6 l.156-157: I do not understand the point of this sentence. The authors introduce a model in which grain growth is only a function of mean annual temperature, which is also the case for the model presented above. Do they mean that Arthern et al. (2010) actually developed two different models? However, the model in which grain growth does not depend on the mean annual temperature is not the one calibrated in this study. Please clarify the purpose of this sentence.

p.6 l.160: Typo, change "*form*" to "from".

p.6 l.171: "*did not contain more than 7 data points for that stage*" before or after the 3-sigma edit?

p.6 l.174: Note that stage 1 and stage 2 were not previously defined, which might be confusing for readers less familiar with firn densification models.

p.7 Eq. (14): Typo, there should be no space in $ln$.

p.8 l.215: Change "*equations*" to "Eq.".

p.8 l.216: I think it is worth mentioning the good agreement of the calibrated coefficients with the calibration of Verjans et al. (2020), despite using very different statistical techniques. This reinforces the reliability of the calibrated dry densification model.

p.8 l.217: Change "*equations*" to "Eq.".

p.8 l.221: Remove the italic from "*any*".

p.8 l.223: Define "*peripheral ice*".

p.8 l.233: What is meant by "*interpolate between these neighbors*"? I believe that the same SMB and FAC time series are taken for all grid cells classified as neighbours. If so, I suggest changing the statement to "we use model output of a single cell as representative for all neighbors".

p.8 Section 2.1.5: In my opinion, the reader should be informed about how the climatic output is processed to the coarsened resolution. This could be summarised in a single sentence by specifying that precipitation, evaporation and melt fluxes are cumulated and by reminding about Eq. 15-16 for the calculation of $T_E$.

p.9 l.243: Use "GrIS" instead of "*Greenland Ice Sheet*".

p.9 l.244: Please be more precise about "*several calibration sites*".

p.9 l.245: Change "*when simulated at five, ten, and twenty days*" to "when simulated at resolutions of five, ten, and twenty days".

p.9 l.246: I am not sure to understand how the residuals in $dFAC/dt$ are computed. Are $dFAC/dt$ values computed at each time step (five, ten, twenty days) or is only the total change in $FAC$ considered?

p.9 l. 247: Are the mean snow accumulation and skin temperature good predictors of residuals in $dFAC/dt$ in the regression model? Could the authors provide summary statistics of the fit?

p.9 l.262: "*the 151 depth-density profiles (stage 1)*" but in Section 2.1.3, the authors mention that they reach 141 depth-density profiles for stage 1.

p.9 l.264-265: Is the regression performed with the mean annual climate of the RCI or the mean annual climate of the entire MERRA-2 climatic forcing?

p.9 l.266-267: I do not understand the iterative removal process. If points with the largest residuals are iteratively excluded and the model is subsequently re-evaluated, there will always be points having residuals outside of the 99[th] percentile. I am probably missing something.

p.9 l.271: Change "*surface mean temperature*" to "mean surface temperature".

p.9 l.272: Specify "we capture more than 50% of the variability for measurements used in the calibration".

p.10 l.282-284: Use the abbreviations "GrIS" and "AIS".

p.10 l.294: I believe that "*Sect. 2.4*" should be changed to "Sect. 2.1.4".

p.10 Eq. (21): "*Eq. (21)*" refers to two different equations. The references to Eq. (21) in the text should subsequently be adjusted.

p.11 l.320: Please clarify what is meant by "*the threshold if determined by one evaluator alone*".

p.11 l.321: I think it is important to insist on the $DDF$ being different for each grid cell. Thus, I suggest changing "*and the DDF calibrated to that threshold*" to "and the grid cell specific $DDF$s calibrated to that threshold".

p.11 l.328: I suggest changing "*realistic magnitudes*" to "realistic annual magnitudes".

p.11 l.329: Again, I suggest changing "*and the DDF calibrated to that threshold*" to "and the grid cell specific $DDF$s calibrated to that threshold".

p.12 l.353: Change "*against observations*" to "against the calibration data set".

p;12 l.354: Clarify what is meant by "*shared variance*". Should it be "explained variance"?

p.12 l.356-357: I have some doubts about the values given for % decrease in model error. I believe that the authors calculate them as $\frac{MAE}{mean(values)}$, which is not the same as $mean\left(\frac{|error|}{value}\right)$.

p.12 l.357: Remove "*Interestingly*". Every reader might not consider it as interesting, although I certainly do.

p.12 l.362: Typo, "*a*" should be "an".

p.13 l.382: I suggest changing "*locally*" to "local".

p.13 Eq. (22): This equation is already given as Eq. (5).

p.14 Section 3.2.1: In my opinion, an interesting and valuable extra-contribution of this study would be to quantify the extension of the GrIS ablation area. That is, how does the extent of the area with $SMB < 0$ has increased post-2003 with respect to the RCI? I leave it to the authors to decide whether to include it in the manuscript or not.

p.14 l.405: Change "*Figure 11a*" to "Figure 11b".

p.14 l.407: Change "*statistical difference*" to "significant difference".

p.14 l.415-416: Is "*followed*" the appropriate word? It seems to me that the decrease in runoff and the increased precipitation are simultaneous.

p.14 l.419-420: Clarify to which period the "*gains*" and "*increases*" refer to. Since 2003 or post-RCI?

p.14 l.425: Typo, there is no verb in this sentence.

p.15 l.428: I think that another word than "*yet*" should be used here.

p.15 l.435-438: If the authors compare grounded- and floating-ice numbers, they should clarify that they consider absolute values here because their extents are very different.

p.15 l.449: I think there might be an error about the value "*142 km$^3$*". Here, the authors use it to quantify the post RFI annual net volume loss, but the same value is given below for the post-2003 period.

p.15 l.458: Consider replacing "*Like the GrIS, the change in FAC is 3 times larger than SMB*" with "The change in FAC is more than 3 times larger than SMB".

p.16 l.463-464: Change "*the height and volume changes begin and end with zero*" to "the height and volume changes in our model experiments begin and end with zero". It is important that the reader understands that this feature is due to a modelling assumption and is not necessarily representative of reality.

p.16 l.473: I think that there should not be a dash between "best" and "fit".

p.16 l.483-488: Please note that the Arthern et al. (2010) model was not developed to capture compaction of the very low density firn layers. The shallowest depth range for which it was calibrated for is 0-5 m. The densification process of very low density fresh snow is governed by different mechanisms, which are likely not well captured by firn densification models.

p.17 l.493: "*GrSMBMIP*" is not defined. I think that the sentence would be clear even without the abbreviation.

p.17 l.494: I think that "*results*" should be singular.

p.17 l.497: Note that the study of Wang et al. (2016) shows that MERRA has a similar bias than other models concerning SMB in Antarctica.

p.17 l.517: "**" can maybe be updated.

Figures: In general, for all figures using different colour scales for the GrIS and AIS, please make sure to add a statement such as "note the different colour scales" in the caption.

p.22 Figure 1: I believe that the open circles were not used in any of the calibration steps of this study. If so, they should be removed from the maps or the statement "*The open circles are calibration site locations*" should be modified.

p. 26-27 Figures 5-6: Please provide the period over which the mean annual climatic values are considered (because MERRA-2 and M2R12K do not have the same time span).

p.32-33 Figures 11-12: Increase the size of the labels of the subfigures a.

p.34-35 Figures 13-14: If possible, increase the size of the axes-labels and of the legends.

References used in this review:

Arthern, R. J., Vaughan, D. G., Rankin, A. M., Mulvaney, R., and Thomas, E. R.: In situ measurements of Antarctic snow compaction compared with predictions of models, J. Geophys. Res.-Earth, 115, 1–12, https://doi.org/10.1029/2009JF001306, 2010.

Fausto, R. S., Box, J. E., Vandecrux, B., van As, D., Steffen, K., MacFerrin, M., Machguth H., Colgan W., Koenig L. S., Mc-Grath D., Charalampidis, C., and Braithwaite, R. J.: A Snow Density Dataset for Improving Surface Boundary Conditions in Greenland Ice Sheet Firn Modeling, Front. Earth Sci., 6, 51, https://doi.org/10.3389/feart.2018.00051, 2018.

Fettweis, X., Hofer, S., Krebs-Kanzow, U., Amory, C., Aoki, T., Berends, C. J., Born, A., Box, J. E., Delhasse, A., Fujita, K., Gierz, P., Goelzer, H., Hanna, E., Hashimoto, A., Huybrechts, P., Kapsch, M.-L., King, M. D., Kittel, C., Lang, C., Langen, P. L., Lenaerts, J. T. M., Liston, G. E., Lohmann, G., Mernild, S. H., Mikolajewicz, U., Modali, K., Mottram, R. H., Niwano, M., Noël, B., Ryan, J. C., Smith, A., Streffing, J., Tedesco, M., van de Berg, W. J., van den Broeke, M., van de Wal, R. S. W., van Kampenhout, L., Wilton, D., Wouters, B., Ziemen, F., and Zolles, T.: GrSMBMIP: Intercomparison of the modelled 1980–2012 surface mass balance over the Greenland Ice sheet, The Cryosphere Discuss., https://doi.org/10.5194/tc-2019-321, in review, 2020.

Hanna, E., Huybrechts, P., Cappelen, J., Steffen, K., Bales, R. C., Burgess, E., McConnell, J. R., Peder Steffensen, J., Van den Broeke, M., Wake, L., Bigg, G., Griffiths, M., and Savas, D.: Greenland Ice Sheet surface mass balance 1870 to 2010 based on Twentieth Century Reanalysis, and links with global climate forcing, J. Geophys. Res.-Atmos., 116, D24121, doi:10.1029/2011JD016387, 2011.

Herron, M. and Langway, C.: Firn densification: an empirical model, J. Glaciol., 25, 373–385, https://doi.org/10.3189/S0022143000015239, 1980.

Koenig, L. and Montgomery, L.: Surface mass balance and snow depth on sea ice working group (SUMup) snow density subdataset, Greenland and Antarctica, 1950–2018, Arctic Data Center, https://doi.org/10.18739/A26D5PB2S, 2019.

Ligtenberg, S. R. M., M. M. Helsen, and M. R. van den Broeke : An improved semi-empirical model for the densification of Antarctic firn, The Cryosphere, 5(4), 809–819, doi:10.5194/tc-5-809-2011, 2011.

Machguth, H., Macferrin, M., van As, D., Box, J. E., Charalampidis, C., Colgan, W., Fausto, R. S., Meijer, H. A. J., Mosley-Thompson, E., and van de Wal, R. S. W.: Greenland meltwater storage in firn limited by near-surface ice formation, Nat. Clim. Chang., 6, 390–393, https://doi.org/10.1038/nclimate2899, 2016.

Medley, B., and Thomas, E. R.: Increased snowfall over the Antarctic Ice Sheet mitigated twentieth-century sea-level rise. Nature Climate Change, 9, 34–39. https://doi.org/10.1038/s41558-018-0356-x, 2019.

van den Broeke, M., Bus, C., Ettema, J. and Smeets, P.: Temperature thresholds for degree-day modelling of Greenland ice sheet melt rates, Geophysical Research Letters, 37(18), 2010.

Verjans, V., Leeson, A. A., Nemeth, C., Stevens, C. M., Kuipers Munneke, P., Noël, B., and van Wessem, J. M.: Bayesian calibration of firn densification models, The Cryosphere, 14, 3017–3032, https://doi.org/10.5194/tc-14-3017-2020, 2020.

---

## Referee Comment (RC2) · Anonymous Referee #2 · 16 Dec 2020

GENERAL

First of all, I would like to apologize for the time it has taken for me to provide this review. This was an oversight on my part and in no way reflects a lack of interest in this paper or the subject.

This manuscript by Medley et al. discusses a new modeling framework that uses a combination of the MERRA-2 reanalysis and the Community Firn Model to arrive at estimates of SMB and its individual components over the Greenland and Antarctic Ice Sheets for the period 1980-2019.

It is an important and welcome contribution to the existing regional climate models and firn models available for estimating ice-sheet mass balance by means of altimetry or

the input-output method. It is encouraging that more of these models become available, and MERRA-2 combined with CFM is presented as a comprehensive modeling framework.

There are three main concerns with this manuscript, which I hope can be addressed by the authors in a thorough revision. But I am sorry to say that this probably entails a substantial effort. In fact, it would have been better from the start to divide this work into a paper that evaluates MERRA-2 over ice sheets, and a paper that deals with the CFM for elevation change and runoff estimates.

1) Evaluation of MERRA-2 against independent observations.

In a global sense, MERRA-2 has been evaluated in a large paper by Gelaro et al. (2017). However to my knowledge, the performance of MERRA-2 over Greenland and Antarctica has not been assessed against observations, like it has been done extensively for e.g. MAR (Fettweis et al., 2013, Agosta et al., 2019), HIRHAM5 (Mottram et al., 2017), RACMO (Van Wessem et al., 2018, Noël et al., 2018) or CESM for climate variables ánd individual SMB components (temperature, snowfall, surface melt, albedo, cloud cover, firn temperature, runoff, etc.). This paper does present SMB and its components but they are not verified, evaluated, or compared against existing estimates. This is worrisome, since MERRA-2 serves as the input for CFM which computes the surface elevation changes needed to correct for ICESat and other altimeters. Thus, the authors will have to make sure that the reader can be confident about the MERRA-2 output over the ice sheets.

2) Uncertainty analysis

The paper lacks uncertainty analyses. Uncertainty intervals of the SMB components are a standard deviation of the sample of annual values. But there is no estimate of the uncertainty of annual values, established for example by independent evaluation against observations (see 1). For the volume and height changes, no uncertainty estimates are provided. Most effort in presenting numbers is not in the numbers themselves, but in quantifying the confidence that we put in them. This is also crucial for establishing an uncertainty estimate of mass loss determined by altimetry. A robust uncertainty analysis must be added to the paper.

Quite a number of processes could be identified that give an uncertainty. Regarding the surface elevations changes from CFM, probably the most important ones are the uncertainty in SMB, snowfall, surface melt, and the assumption that the RCI period is representative for the long-term climate (see comments by reviewer 1).

Adding an uncertainty estimate allows for a fair judgement of the numbers presented in this paper, and for an comparison with existing literature (see also point 3), like for example the volume change estimates from Kuipers Munneke et al., which at first sight appear to be larger over Greenland than from MERRA-2.

3) Embedding in existing literature

While I believe that MERRA-2/CFM is a very valuable addition to the suite of models that are currently available to estimate SMB and firn changes over the ice sheets, the literature from those existing models is largely ignored in this paper. It would be insightful if the numbers in this study are compared to existing estimates of SMB and firn volume change (MAR, RACMO, CESM, HIRHAM, ...). Also it would be fair to refer to previous work into, e.g., quantifying seasonal cycles of firn processes (Ligtenberg et al., 2012), or volume loss over Greenland.

SPECIFIC COMMENTS

Page 1 line 10: suggest to include the time frame of the simulations in the abstract: "new simulations of firn processes (1980-2019)"

Page 5 line 126: how is temperature initialized for locations with SMB<0?

Page 7 line 191: why was the functional form of equations (11) and (12) chosen to be an Arrhenius-style rate equation? Is there a reason to suppose that the misfit between modeled and observed densification rates should take this form?

Page 7 line 211: it is interesting to see that $E_0$ is found to be zero. It implies that the correction factor $R_0$ reduces to $b^{\beta_0}$, i.e. with no dependence on temperature. For the reader, it would be interesting to show this result rather than the current figure 4. The present figure 4 is of limited interest, since the correction procedures forces the data points to the 1:1 line by design. It would be more interesting to show the validity and behavior as an Arrhenius plot:

$$R_0 = b^{(\beta_0)} \exp(-E0/RT)$$

$$\ln(R_0) = \beta_0 \ln(b) - E_0 (1/RT)$$

With $E_0 = 0$, plotting $\ln(R_0)$ against $\ln(b)$ would show $\beta_0$ as the gradient and 0 as the intercept of a linear plot.

For $R_1$ it is more difficult. A normal Arrhenius plot would look like

$$\ln(R_1) = (-E_1/R) * (1/T) + \beta_1 * \ln(b)$$

which is a linear $y = mx + d$ with $1/T$ as x and $\beta_1 * \ln(b)$ as a constant. However, the latter is not constant but depends on b. Perhaps it is possible to select a subset of $\sim$20 firn cores with a narrow range in b so that a constant value for $(\beta_1 * \ln(b)$ can be assumed, and $\ln(R_1)$ can be plotted against $1/T$ as a linear function.

Page 9 line 270: I can understand that you have looked only at the significantly related predictors, and not at their physical interpretation. But I think that mean northward wind speed is an awkward predictor especially in the Greenland context. Could the performance of the model be retained while dropping $V_0$?

Page 10 line 297: I am struggling with the fact that no meltwater-related output from MERRA-2 was used in this manuscript, only snowfall. Is MERRA-2 not designed to provide this output? Is it of insufficient quality? Was it forgotten? Regarding runoff, was the runoff output from MERRA-2 used, or from the firn model? This remains unclear. In any case, is the runoff from the firn model comparable and similar to the runoff from MERRA-2?

Page 10 line 300: The degree-day modeling approach is confusing, and quite rough. The approach by Van den Broeke et al. (2010) was devised only in cases where daily mean temperatures temperatures are available, to compensate for the lack of representation of the daily cycle. I have tried to understand why such low $T_0$ are required to get a good match at an annual scale. I guess this is only possible if MERRA-2 has a strong cold bias, or a very poor representation of the daily temperature cycle, or if the degree-day approach actually fails over the entire range of melt-temperature combinations on both ice sheets. I would like the authors to comment on this. More generally, I wonder if surface melt in MERRA-2 is poorly represented, or if it was forgotten or otherwise impossible to obtain this output variable.

Page 13 line 382: a map showing the absolute change in FAC would be interesting here.

Page 14, section 3.2: To my opinion, it is important to emphasize that the surface melt estimates from Greenland and Antarctica are not independent numbers based on the outcomes of a physical model. Rather, these are the results of a degree-day model calibrated to observations (Antarctica) and another model (MAR, Greenland). Especially for Antarctica, it is presented (lines 427 and further) as if the melt estimates are an independent result of MERRA-2 and CFM, whereas in reality the degree-day method is tuned to reproduce the numbers by Trusel et al. over a part of the RCI.

Page 15, line 444: RFI -> RCI (?)

===

Agosta et al., 2019. Estimation of the Antarctic surface mass balance using the climate model MAR (1979-2015) and identification of dominant processes, The Cryosphere

Fettweis et al., 2013. Estimating the Greenland ice sheet surface mass balance contribution to future sea level rise using the regional atmospheric climate model MAR, The Cryosphere

Gelaro et al., 2017. The modern-era retrospective analysis for research and applications , version 2 (MERRA-2), Journal of Climate

Kuipers Munneke et al., 2015. Elevation change of the Greenland Ice Sheet due to surface mass balance and firn processes, 1960–2014, The Cryosphere

Ligtenberg et al., 2012. Quantifying the breathing of the Antarctic Ice Sheet, GRL

Mottram et al., 2017. Surface mass balance of the Greenland icesheet in the regional climate model HIRHAM5 : Present state and future prospects. Low Temperature Science. Series A. Physical Science.

Noël et al., 2018. Modelling the climate and surface mass balance of polar ice sheets using RACMO2 - Part 1 : Greenland (1958-2016), The Cryosphere

Trusel et al., 2013. Satellite-based estimates of Antarctic surface meltwater fluxes, GRL

Van den Broeke et al., 2010. Temperature thresholds for degree-day modelling of Greenland ice sheet melt rates, GRL

Van Wessem et al., 2018. Modelling the climate and surface mass balance of polar ice sheets using RACMO2, part 2: Antarctica (1979-2016), The Cryosphere

---

## Author Comment (AC1) · 12 Oct 2021

*Response to RC1. The authors provide responses to the reviewer comments in italic serif typeface throughout whereas the reviewers comments remain in normal sans serif typeface.*

This study investigates a modelling approach to estimate ice sheet wide time series of Surface Mass Balance (SMB) and Firn Air Content (FAC) evolution on both the Greenland and Antarctic ice sheets (GrIS and AIS). Using a set of firn cores, the authors recalibrate a firn densification model and establish a new formulation for surface snow density. The MERRA-2 atmospheric reanalysis product is used to compute climatic conditions and to force the firn densification model. The importance of different SMB components as well as their evolution is analysed for both ice sheets. The authors partition ice sheet wide volume changes associated with surface processes between mass and FAC changes, showing that the latter dominates seasonal variability, while the former dominates multi-annual trends.

I believe that this study demonstrates a comprehensive approach to estimate ice sheet SMB and FAC evolution. The modelling framework is robust, well-explained and is undoubtedly a great contribution to the firn modelling community. The use of MERRA-2 is also noteworthy, because this is the first assessment of ice sheet SMB using this product. The authors propose innovative ways to deal with challenges associated with decades-long, large scale simulations, and their results demonstrate an extensive work to perform these simulations. The objective of the study has direct implications for ice sheet mass balance assessments performed via satellite-based altimetry techniques or via the input-output method. As for any model-based study, assumptions and simplifications had to be made. I list my reservations concerning some aspects of the approach in this review. Given the quality of the study, I am confident that a slightly revised version of the manuscript will be accepted for future publication. I realise that modifying the modelling approach to account for any suggestion from the reviewers would subsequently require to re-run the simulations. I do not consider this necessary to address the reservations I raise. However, I expect the authors to provide strong justifications or to better acknowledge limitations with respect to my reservations in the revised manuscript.

I have separated my review in Specific comments requiring more clarity in the manuscript and/or strong justifications from the authors in their response, and Technical comments related to the structure of the text. Despite my numerous comments, I strongly encourage the authors to re-submit the manuscript after the revisions have been made.

*We thank Dr. Verjans for their support of the work presented and the time dedicated to providing a very thorough review that has substantially improved our research and publication quality. We apologize for the lengthy wait for the response; both reviews provided substantive feedback for improvement of our work, which required additional work. We acknowledge Dr. Verjans' review revolved more around adding clarifications and limitations to the modelled work rather than actually re-running simulations. As they will see, we have re-run the simulations accommodating many of the comments within both their review as well as that of reviewer 2, which we believe have led to an improvement in the quality of work.*

Specific Comments

1) The surface density formulation ($\rho 0$)

Constraining surface snow density is both important and difficult. The authors took the approach of using a very large range of possible predictor variables and a single formulation for both ice sheets. My first concern relates to the neglect of melt sites. The formulation is calibrated only to the set of firn cores previously selected for the dry firn compaction calibration. However, surface snow density values

are also used in the percolation and ablation areas of the GrIS, which are rejected from the calibration selection but account for a predominant part of the GrIS total area (see Figure 1). The estimated $\rho 0$ values are particularly low in the GrIS southwest, which is the highest melt area. This disagrees with observational studies (e.g. Machguth et al., 2016; Fausto et al., 2018). Why did the authors decide to use the same selection criteria for the surface snow density calibration as for the dry firn compaction calibration? Surface snow density should be accurately represented, including for higher melt areas. Underestimating surface snow density there can lead to overestimation of the firn retention capacity, and thus lower runoff values. I believe that this largely explains why the GrIS-wide runoff estimate of this study is on the lower end of the GrIS SMB intercomparison (Fettweis et al., 2020). A similar effect can be perceived on the Antarctic Peninsula, which has $\rho 0$ values in the lowest range of the AIS.

*The reviewer brings up an excellent concern regarding our choice of eliminating sites that experience significant melt from our constraint on surface density. We have opted to include **all** observations of density, irrespective of their melt rates, in our calculation of surface density. The initial reason that the melt-rich sites were excluded was under an interpretation of "new snow" as "dry snow." The thought process being that snow is deposited at an initial dry density and then subsequently modified as melt occurs. However, we realize that this surface density is more of a long-term integrated mean density, sometimes being dry and other times wet; thus, melt should be considered when using such a simplified surface density scheme. The new parameterization provides more realistic results when compared to observations, and we use the references provided to acknowledge their contribution.*

My second concern relates to the use of the northward wind speed in the formulation. This climatic variable has very different physical meanings in various areas of the AIS (i.e. wind from the inner continent or from the shore), and even more between AIS and GrIS. I am thus skeptical about the physical sense to include it in the parameterisation. Furthermore, given the few GrIS cores used in the calibration, I believe that including the northward wind speed may undermine the validity of the parameterisation on the GrIS.

*Both reviewers expressed concern regarding the lack of physical basis for the use of northward winds. We agree with the reviewers and have reformulated the relationship using total windspeed rather than just northward.*

Finally, the model performs worse for the lowest $\rho 0$ values. It should be mentioned that these correspond to surface densities most critical for FAC calculations because firn layers of low density have high FAC values.

*Yes. The simple regression is not capable of reproducing the lowest initial densities, which we express on Line 273. We have added a line in the section on FAC (Section 3.1) to express more clearly the impact of overestimation of surface density on our model results.*

2) The use of the effective temperature ($TE$)

It is well-explained in Section 2.1.3 that stage-1 firn compaction rates cannot be assumed to depend simply on the mean annual temperature. Computing an effective temperature is a novel approach that accounts for the impact of high temperatures on compaction rates, but that leads to other potential problems. Firstly, I think that more details should be given about Equations (15) and (16). In Eq. (16), is $TE$ meant to be $\overline{TE}$ ? It is stated that Eqs. (15) and (16) are only used for stage-1 densification, thus should $Ec$ be $Ec0$ ? Is $\overline{k}$ the average of all hourly $k$ values of the climatic forcing?

*We thank the reviewer for the close inspection of the equations presented and indeed, they were lacking some key details to fully understand the calculations performed. Because of the concerns regarding the potential "downstream" impacts of the use of an effective temperature, we have elected to re-run the simulations using mean temperatures rather than mean effective temperatures. We have added a section that compare the results between the two approaches with simulations at daily resolution. The use of a traditional mean temperature better matched the 1-day simulations better than using the effective mean. Thus, we have eliminated use of the effective temperature. We do, however, add in the text that there are likely other ways in which an effective mean could be utilized in future work.*

Secondly, several shortcomings related to the use of $TE$ should be mentioned. As far as I understand, $k$ represents the compaction rate of a firn layer at the skin temperature. The temperature signal is dampened in depth and $k$ thus overestimates the compaction rate of the whole stage 1 firn. In turn, this leads to an overestimation of the effective temperature. Also, the use of $TE$ as temperature forcing in the temporal coarsening of the climate input can have a significant impact on the firn temperature profile in the simulations. In the CFM, a newly deposited layer has its temperature set at the temperature of the time step (thus $TE$ in this approach). Subsequently, the layer is buried and carries this temperature signal, causing advective heating. I guess that $TE$ is significantly higher than the mean skin temperature for a large majority of the grid cells and coarsened time steps. As such, advective heating is significantly greater, which in turn leads to overestimated firn compaction rates. I do not know how much the 5-, 10- and 20-days $TE$ values differ from their mean temperature and thus how strong this bias in compaction rates is.

*The reviewer is entirely correct that effective temperatures will be larger than traditional mean temperatures; thus, the modeled temperatures at depth are larger than if they were modeled with mean temperatures. Thus, use of the temperature profile from the initial simulations would provide temperatures that are not realistic. Thus, we abandoned the effective temperature, but do note that this could be implemented in a future CFM release by simulating both effective and physical temperatures with time.*

3) Densification Model Calibration (Section 2.1.3)

3a) The assumption that the "logarithm of the firn density profile with depth is approximately linear" was stated by Herron and Langway (1980) but never mathematically verified. I would appreciate if the authors could validate their use of this assumption for their calibration process. I suggest that summary statistics are provided to evaluate this validity. It should be straightforward to compare point density measurements of the firn core dataset to corresponding density values taken from the estimated log-profile (and re-converted to kg m-3 units). I ask the authors to provide RMSE and R2 values of the fit for both stage-1 measurements and stage-2 measurements. These statistics should be computed before removal of the measurements via the iterative 3-sigma edit. I would welcome any valid alternative way to validate this assumption put forward by the authors.

*The reviewer raises another reasonable concern regarding the approach to using linear fits to the logarithm of the firn depth-density profile. In order to address the concern, we have executed a comparative analysis of the fit statistics between deriving a linear fit to the log-profile as well as the original density profiles. We believe that we have designed it as the reviewer suggested, including keeping all sites (rather than just using those that remained after the 3-sigma edit. The results are presented in Section 2.1.3 and we have added a plot comparing the RMSE/R2 values between the linear fits to the actual density profile against the log(density) profile.*

3b) I am not sure to understand how "the firn density measurements and model output are binned into half-meter depth increments to obtain similar sampling intervals before slopes are estimated". Are all measurements (resp. model outputs) averaged in intervals of 0.5 m and the slopes computed on these 0.5 m averaged density points?

*We bin the observations to 0.5 meter increments as well as the respective model output. Slopes are then calculated on these binned profiles (both observations and model output). We have reworded the sentence on Line 167 to better express the actual technique.*

3c) There are mathematical inconsistencies when substituting Eq. (11) in Eq. (18). The final formulation of the firn model assumes: $\bar{b}^{\beta_0} \times b = b^{1+\beta_0} \ exp ( −60000 \ RT ) \times exp ( −Ec_0 \ RT\bar{\bar{E}} ) = exp ( −60000−Ec_0 \ RT )$ Both these assumptions are mathematically wrong. I understand that these are made for practical purposes, but they should at least be mentioned in the manuscript. Similar concerns hold for the stage-2 formulation (substituting Eq. (12) in Eq. (19)), even though they are less critical because $\bar{b} \approx b$ and $\bar{T} \approx T$ in deeper firn.

*The reviewer is absolutely correct; we had to make assumptions regarding how the calibration translates into modification to the densification rates, which we did not fully expand. We have clarified the assumptions made (as well as the equations presented) to ensure mathematical correctness.*

3d) Why do the authors reject sites falling in a same grid cell? And how do they choose which depth-density profile to exclude? They could very well compute two different pairs $(R_0, R_1)$ within a single grid cell. This would illustrate the natural small-scale heterogeneity of firn structure.

*We actually do not reject sites that fall within the same grid cell and have added more context to the sentence beginning on Line 159. In reality, what we meant to convey is that we do actually compare **all** observed profiles with model output, but that the number of model simulations is fewer than the total amount of observations because several observations fall within the same grid cell. We realize that the language used was vague and have tried to be more explicit in our description.*

3e) Why is the intercept forced to 0 in the regression? Is it to make the estimation of the parameters well-determined?

*The choice to set the intercept to zero was to ensure modification to the functional form presented by Arthern et al. (2010) and limit the possibility of overdetermination. Basically, we wanted to ensure that the observed variability was linked to an atmospheric control rather than an unknown. We have added justification to this choice in the text on Line 193.*

3f) Is $E_0$ exactly 0? Or was it sufficiently close to 0 to set it equal to 0?

*Excellent question. The uncertainty in the linear fit was larger than the prediction, so we assume that the E0 does not differ significantly from 60kj per mol, or rather, the observations are not sufficient to resolve a significant deviation from 60kj per mol. We have clarified how we came to the results in Equation 17.*

4) The degree-day model (Section 2.2.1)

4a) In their study, van den Broeke et al. (2010) used $T_0 < 273.15$ K with the justification that: "on days with a negative average T2m, the method predicts zero melt if T0 = 273.15 K is used, while melt may have occurred during a short period. This problem may be avoided by applying the method to hourly T2m values or by applying a lower value for T0". Because hourly $T2m$ values are used in this study, citing

van den Broeke et al. (2010) to justify the choice of $T0< 273.15$ K is inappropriate. Also, this raises the question of the physical sense of using $T0< 273.15$ K. Should the calibration not rather fix $T0=273.15$ K and tune $DDF$ only?

*We appreciate the concern from the reviewer as it is one of the largest assumptions made in this work and can be improved in future studies. The reviewer is correct that we should not cite van den Broeke et al. (2020) to justify use of a T0 < 273.15 as they were using daily means whereas we use hourly data. We did, however, learn from van den Broeke et al. (2010) that DDFs become more realistic at lower T0 threshold, which is something that we found in our analysis as well. We have refined Section 2.2.1 to acknowledge the reference in a more appropriate manner. When using T0 at 273.15K, we ended up with unrealistically large DDFs over regions that experience little melt. As we moved to lower T0, the values normalized more, and we use an evaluation of how well the model predicts the training data to select the most realistic model. The use of a degree-day model is not the most physically robust model; we have attempted to expand on the limitations. Additionally, we do not explore the potential for temperature bias within MERRA-2, which would also bias the temperature threshold needed to reflect melt.*

4b) The selection of the best $T0$ threshold depends on maximizing $r2$ and minimizing $RMSE$. However, it is not explained which particular $r2$ and $RMSE$ are considered since many grid cells are used. It is only in the caption of Figure 7 that the authors mention that "the median $r2$ and $RMSE$ of every grid cell" are used. This should be stated in the main text. Also, I wonder about the relevance of the choice of the median values. Most of the grid cells have very low melt rates. It is not important to capture the low melt rates with great accuracy. It is much more important to capture melt rates of the grid cells in high melt areas. Thus, why choosing the median?

*We agree that we need to provide more context in regards to how the curves in Figure 7, so we have added additional explanations in Section 2.2.1 to make it more clear how we performed our analysis, but also explain many of the limitations. We have used the median because we were more interested in representing most values, regardless of magnitude. When interpreting ice elevation change, while the largest melt rates impart the largest height changes, we must consider the relative importance of the changes. So even small melt rates can be relatively important outside of regions of fast-flowing ice and strong ice melt. We discuss this choice and present the T0 solution if we had used the mean to show they do not differ by much.*

4c) I ask the authors to provide the final ranges of $DDF$ values used for the GrIS and for the AIS.

*We have provided the distributions in a new figure and present the ranges in the text in Section 2.2.1.*

5) Wet firn compaction

Simulating wet firn compaction and liquid water processes is a major weakness of firn models. I certainly do not blame the authors for this and addressing this shortcoming is not the subject of this study. I appreciate that the results of the compaction model are also evaluated at high melt sites (Figure 8). I think it is important to also provide the bias of the compaction model at the zero-, moderate- and high-melt sites to know if the model tends to overestimate/underestimate densities in such melt conditions. Also, I believe that the text should remind the reader in the Discussion section about the wet compaction shortcoming and that it can have a significant impact on FAC results for the GrIS and ice shelves. As stated by the authors themselves, only a limited area of the GrIS satisfies the criteria used for the dry firn compaction calibration. This implies that firn compaction can only be expected to be well represented in that limited area.

*The reviewer brings up a very fair point, and we have carved out additional discussion within the FAC discussion on how poorly constrained wet firn processes are over the ice sheets. We appreciate that this lack of knowledge translates really challenges our ability to model FAC changes, especially over the Greenland Ice Sheet, which experiences a significant amount of melt. We still use the results but have made it clear that the results are limited significantly by our lack of understanding of wet snow/firn processes and that the field needs improved process studies to build better models.*

6) The Reference Climate Interval (RCI)

In Section 3.2, the authors are perfectly right: "The RCI is ideally representative of long-term steady-state conditions". However, when they evaluate the assumption of their RCI choice, they only assess the "steady-state" aspect and neglect the "long-term" aspect. Indeed, the RCI should show no trend in any climatic field and this is thoroughly investigated for both the GrIS and the AIS RCIs. But the RCI should also be representative of the climate under which the firn column was established (i.e. of the past centuries in AIS). Some studies contradict the assumption that 1980- 2019 is representative of the long-term AIS climate and that there are very likely some pronounced regional trends (e.g. Medley and Thomas, 2019). Similarly, in Greenland, regional long-term trends may exist (e.g. Hanna et al., 2011). This impacts the spin-up process because the initial firn column should be in equilibrium with the past climate. Again, such difficulties are inherent to firn simulations because reliable climate forcing covers only the recent decades. Thus, one cannot expect the authors to have a solution to this particular problem. But I would like this limitation to be mentioned in the manuscript, as well as its potential impacts on the findings.

*The reviewer highlights a very important assumption that all firn evolution simulations must consider, and they rightfully point out that we do not fully convince the readers of the work as to our selection of RCI. We have added a "Limitations" section within the discussion to highlight this challenge as well as several other challenges we face when modeling firn column processes (surface density, degree-day modeling, etc.). While we cannot solve this problem, we do provide an evaluation of the potential impact of this selection. We have done several additional simulations for select sites to show how this choice impacts our modeled FAC changes.*

7) Comparing SMB and FAC components

Seasonal variability in height is shown to be driven more by FAC than by snow mass. However, FAC gain/loss is essentially governed by snowfall amounts. For example, if we assume 1 m i.e. accumulation over a given month and a surface snow density of 300 kg m-3 , the corresponding FAC gain is ~2 m. In other words, without considering compaction, one should expect FAC variability to be around 2 times larger than SMB variability. The values found in this study are around 3 and show the additional effect of seasonally varying compaction rates. But the reader should be explicitly informed about the direct dependence of FAC variability on the SMB variability and, as a consequence, about its expected larger magnitude. Most of the change in FAC is not simulated by firn densification models but is determined by the climatic forcing. In regards to this aspect, I find the statement in the Conclusion line 505 misleading ("Thus, determination of seasonal mass change using satellite altimetry requires a substantial FAC correction, highlighting the importance of firn densification models, especially when investigating shorter intervals of change as not being mindful of the seasonal cycles of SMB and FAC can generate large biases.").

*We entirely agree that FAC variability is driven predominantly by snow accumulations (or lack thereof), which include both air and ice mass. Our intent was not to mislead, but rather highlight the difference between changes in "air" versus "ice" because of the importance for satellite altimetry studies of ice-sheet mass balance. Specifically, we must remove the changes in air to measure the mass balance. How does one convert SMB to height change? You must have a density, which is a large unknown as the reviewer has already pointed out. We have reworded many of the sentences discussing this to clarify that SMB controls ice and air, whereas firn processes control air content. In regards to the sentence of interest, we have clarified to combine surface mass balance AND firn models importance.*

Technical comments

*All changes suggested by the reviewer were included except those with specific responses.*

p.2 l.39: "few hundred meters", I am not sure that the firn column can be that deep (e.g. Ligtenberg et al., 2011), please provide a reference.

p.2 l.53: Make sure to consistently use either "solid earth" or "solid-earth" throughout the manuscript.

p.3 l.71: I suggest adding a statement underlining the sensitivity of Eq. (3), such as "Mass balance estimates are highly sensitive to small errors in the height change measurements and in the modelled firn signal."

p.3 l.72: I think that "Variable rates of the height change due to compaction" should be replaced by "Height changes due to variable rates of compaction".

p.3 l.73-76: I suggest not introducing the variables $dhc/dt$, $dhm/dt$ and $dha/dt$ because these are not used in the remainder of the manuscript.

p.3 l.80: SMB and FAC appear in the wrong order: "air thickness and the thickness of ice: surface mass balance (SMB) and firn air content (FAC), respectively"

p.3 l.86: Add a comma: "(…) mass fluxes at the surface, including (…)"

p.4 Eq. (6): There is a typo in the equation, which should have $\rho i$ in the denominator: $FAC = \int \rho i - \rho(z) \, \rho i \, dz \, z \rho i \, 0$

p.4 l.110: Why do the authors simulate grain-size evolution?

*It was simulated as a test. We have clarified that the grain sizes were simulated, but are not likely useable because we were testing the capabilities.*

p.5 l.141: "subset of 256 published firn depth-density profiles" The authors should provide a little more detail about the dataset of firn cores used in this study. I suppose that the SUMup dataset is used. If this is the case, the authors should cite the work of Koenig and Montgomery (2019) (https://doi.org/10.18739/A26D5PB2S). If other datasets are used, they should also be cited. All the references can be provided in the section Code and data availability or in the section Acknowledgements.

*We did not use the SUMup dataset, but rather compiled data from the literature. It was a complete oversight to not reference the data. We have now included the references.*

p.6 l.156-157: I do not understand the point of this sentence. The authors introduce a model in which grain growth is only a function of mean annual temperature, which is also the case for the model presented above. Do they mean that Arthern et al. (2010) actually developed two different models? However, the model in which grain growth does not depend on the mean annual temperature is not the one calibrated in this study. Please clarify the purpose of this sentence.

p.6 l.160: Typo, change "form" to "from".

p.6 l.171: "did not contain more than 7 data points for that stage" before or after the 3-sigma edit?

*Before the 3-sigma edit. We have clarified the language to express that.*

p.6 l.174: Note that stage 1 and stage 2 were not previously defined, which might be confusing for readers less familiar with firn densification models.

*Thank you for the comment. We have introduce the stages earlier.*

p.7 Eq. (14): Typo, there should be no space in $ln$.

p.8 l.215: Change "equations" to "Eq.".

p.8 l.216: I think it is worth mentioning the good agreement of the calibrated coefficients with the calibration of Verjans et al. (2020), despite using very different statistical techniques. This reinforces the reliability of the calibrated dry densification model.

*We agree and have included it in our revision.*

p.8 l.217: Change "equations" to "Eq.".

p.8 l.221: Remove the italic from "any".

p.8 l.223: Define "peripheral ice".

p.8 l.233: What is meant by "interpolate between these neighbors"? I believe that the same SMB and FAC time series are taken for all grid cells classified as neighbours. If so, I suggest changing the statement to "we use model output of a single cell as representative for all neighbors".

p.8 Section 2.1.5: In my opinion, the reader should be informed about how the climatic output is processed to the coarsened resolution. This could be summarised in a single sentence by specifying that precipitation, evaporation and melt fluxes are cumulated and by reminding about Eq. 15-16 for the calculation of $TE$.

p.9 l.243: Use "GrIS" instead of "Greenland Ice Sheet".

p.9 l.244: Please be more precise about "several calibration sites".

p.9 l.245: Change "when simulated at five, ten, and twenty days" to "when simulated at resolutions of five, ten, and twenty days".

p.9 l.246: I am not sure to understand how the residuals in $dFAC/dt$ are computed. Are $dFAC/dt$ values computed at each time step (five, ten, twenty days) or is only the total change in $FAC$ considered?

*We have run all of Antarctica at 5-day resolution, so this analysis is no longer relevant.*

p.9 l. 247: Are the mean snow accumulation and skin temperature good predictors of residuals in $dFAC/dt$ in the regression model? Could the authors provide summary statistics of the fit?

*See prior comment.*

p.9 l.262: "the 151 depth-density profiles (stage 1)" but in Section 2.1.3, the authors mention that they reach 141 depth-density profiles for stage 1.

p.9 l.264-265: Is the regression performed with the mean annual climate of the RCI or the mean annual climate of the entire MERRA-2 climatic forcing?

*The mean annual climate of the RCI. We have clarified.*

p.9 l.266-267: I do not understand the iterative removal process. If points with the largest residuals are iteratively excluded and the model is subsequently re-evaluated, there will always be points having residuals outside of the 99th percentile. I am probably missing something.

*We have clarified the process. We do iteratively remove (and recalculate statistics) because one or two extreme outliers skew the statistics. We present the results when throwing out outliers in a non-iterative fashion as well to show it doesn't impact the results.*

p.9 l.271: Change "surface mean temperature" to "mean surface temperature".

p.9 l.272: Specify "we capture more than 50% of the variability for measurements used in the calibration".

p.10 l.282-284: Use the abbreviations "GrIS" and "AIS".

p.10 l.294: I believe that "Sect. 2.4" should be changed to "Sect. 2.1.4".

p.10 Eq. (21): "Eq. (21)" refers to two different equations. The references to Eq. (21) in the text should subsequently be adjusted.

p.11 l.320: Please clarify what is meant by "the threshold if determined by one evaluator alone".

p.11 l.321: I think it is important to insist on the $DDF$ being different for each grid cell. Thus, I suggest changing "and the $DDF$ calibrated to that threshold" to "and the grid cell specific $DDF$s calibrated to that threshold".

p.11 l.328: I suggest changing "realistic magnitudes" to "realistic annual magnitudes".

p.11 l.329: Again, I suggest changing "and the $DDF$ calibrated to that threshold" to "and the grid cell specific $DDF$s calibrated to that threshold".

p.12 l.353: Change "against observations" to "against the calibration data set".

p;12 l.354: Clarify what is meant by "shared variance". Should it be "explained variance"?

p.12 l.356-357: I have some doubts about the values given for % decrease in model error. I believe that the authors calculate them as $MAE\ mean(values)$, which is not the same as $mean\ (\ |error|\ value\ )$.

*We calculate the mean absolute error as the mean(abs(error)). The MAE numbers presented in Figure 4 are compared to the values presented on Lines 355-357 to generate the percentages. Specifically, Figure*

*1a shows MAE = 0.030 while the mean observed rate is 0.066, which we call a mean absolute error of 45%. We have attempted to clarify how these numbers were generated.*

p.12 l.357: Remove "Interestingly". Every reader might not consider it as interesting, although I certainly do.

p.12 l.362: Typo, "a" should be "an".

p.13 l.382: I suggest changing "locally" to "local".

p.13 Eq. (22): This equation is already given as Eq. (5).

p.14 Section 3.2.1: In my opinion, an interesting and valuable extra-contribution of this study would be to quantify the extension of the GrIS ablation area. That is, how does the extent of the area with $SMB <$ 0 has increased post-2003 with respect to the RCI? I leave it to the authors to decide whether to include it in the manuscript or not.

*We have included this analysis as it is an interesting contribution to provide.*

p.14 l.405: Change "Figure 11a" to "Figure 11b".

p.14 l.407: Change "statistical difference" to "significant difference".

p.14 l.415-416: Is "followed" the appropriate word? It seems to me that the decrease in runoff and the increased precipitation are simultaneous.

p.14 l.419-420: Clarify to which period the "gains" and "increases" refer to. Since 2003 or post-RCI?

p.14 l.425: Typo, there is no verb in this sentence.

p.15 l.428: I think that another word than "yet" should be used here.

p.15 l.435-438: If the authors compare grounded- and floating-ice numbers, they should clarify that they consider absolute values here because their extents are very different.

p.15 l.449: I think there might be an error about the value "142 km3 ". Here, the authors use it to quantify the post RFI annual net volume loss, but the same value is given below for the post-2003 period.

p.15 l.458: Consider replacing "Like the GrIS, the change in FAC is 3 times larger than SMB" with "The change in FAC is more than 3 times larger than SMB".

p.16 l.463-464: Change "the height and volume changes begin and end with zero" to "the height and volume changes in our model experiments begin and end with zero". It is important that the reader understands that this feature is due to a modelling assumption and is not necessarily representative of reality.

p.16 l.473: I think that there should not be a dash between "best" and "fit".

p.16 l.483-488: Please note that the Arthern et al. (2010) model was not developed to capture compaction of the very low density firn layers. The shallowest depth range for which it was calibrated for is 0-5 m. The densification process of very low density fresh snow is governed by different mechanisms, which are likely not well captured by firn densification models.

p.17 l.493: "GrSMBMIP" is not defined. I think that the sentence would be clear even without the abbreviation.

p.17 l.494: I think that "results" should be singular.

p.17 l.497: Note that the study of Wang et al. (2016) shows that MERRA has a similar bias than other models concerning SMB in Antarctica.

p.17 l.517: "" can maybe be updated.

Figures: In general, for all figures using different colour scales for the GrIS and AIS, please make sure to add a statement such as "note the different colour scales" in the caption.

p.22 Figure 1: I believe that the open circles were not used in any of the calibration steps of this study. If so, they should be removed from the maps or the statement "The open circles are calibration site locations" should be modified.

p. 26-27 Figures 5-6: Please provide the period over which the mean annual climatic values are considered (because MERRA-2 and M2R12K do not have the same time span).

p.32-33 Figures 11-12: Increase the size of the labels of the subfigures a.

p.34-35 Figures 13-14: If possible, increase the size of the axes-labels and of the legends.

References used in this review:

Arthern, R. J., Vaughan, D. G., Rankin, A. M., Mulvaney, R., and Thomas, E. R.: In situ measurements of Antarctic snow compaction compared with predictions of models, J. Geophys. Res.-Earth, 115, 1–12, https://doi.org/10.1029/2009JF001306, 2010.

Fausto, R. S., Box, J. E., Vandecrux, B., van As, D., Steffen, K., MacFerrin, M., Machguth H., Colgan W., Koenig L. S., Mc-Grath D., Charalampidis, C., and Braithwaite, R. J.: A Snow Density Dataset for Improving Surface Boundary Conditions in Greenland Ice Sheet Firn Modeling, Front. Earth Sci., 6, 51, https://doi.org/10.3389/feart.2018.00051, 2018.

Fettweis, X., Hofer, S., Krebs-Kanzow, U., Amory, C., Aoki, T., Berends, C. J., Born, A., Box, J. E., Delhasse, A., Fujita, K., Gierz, P., Goelzer, H., Hanna, E., Hashimoto, A., Huybrechts, P., Kapsch, M.-L., King, M. D., Kittel, C., Lang, C., Langen, P. L., Lenaerts, J. T. M., Liston, G. E., Lohmann, G., Mernild, S. H., Mikolajewicz, U., Modali, K., Mottram, R. H., Niwano, M., Noël, B., Ryan, J. C., Smith, A., Streffing, J., Tedesco, M., van de Berg, W. J., van den Broeke, M., van de Wal, R. S. W., van Kampenhout, L., Wilton, D., Wouters, B., Ziemen, F., and Zolles, T.: GrSMBMIP: Intercomparison of the modelled 1980–2012 surface mass balance over the Greenland Ice sheet, The Cryosphere Discuss., https://doi.org/10.5194/tc-2019-321, in review, 2020.

Hanna, E., Huybrechts, P., Cappelen, J., Steffen, K., Bales, R. C., Burgess, E., McConnell, J. R., Peder Steffensen, J., Van den Broeke, M., Wake, L., Bigg, G., Griffiths, M., and Savas, D.: Greenland Ice Sheet surface mass balance 1870 to 2010 based on Twentieth Century Reanalysis, and links with global climate forcing, J. Geophys. Res.- Atmos., 116, D24121, doi:10.1029/2011JD016387, 2011.

Herron, M. and Langway, C.: Firn densification: an empirical model, J. Glaciol., 25, 373–385, https://doi.org/10.3189/S0022143000015239, 1980.

Koenig, L. and Montgomery, L.: Surface mass balance and snow depth on sea ice working group (SUMup) snow density subdataset, Greenland and Antarctica, 1950–2018, Arctic Data Center, https://doi.org/10.18739/A26D5PB2S, 2019.

Ligtenberg, S. R. M., M. M. Helsen, and M. R. van den Broeke : An improved semi-empirical model for the densification of Antarctic firn, The Cryosphere, 5(4), 809–819, doi:10.5194/tc-5-809-2011, 2011.

Machguth, H., Macferrin, M., van As, D., Box, J. E., Charalampidis, C., Colgan, W., Fausto, R. S., Meijer, H. A. J., Mosley-Thompson, E., and van de Wal, R. S. W.: Greenland meltwater storage in firn limited by near-surface ice formation, Nat. Clim. Chang., 6, 390–393, https://doi.org/10.1038/nclimate2899, 2016.

Medley, B., and Thomas, E. R.: Increased snowfall over the Antarctic Ice Sheet mitigated twentieth-century sea-level rise. Nature Climate Change, 9, 34–39. https://doi.org/10.1038/s41558-018-0356-x, 2019.

van den Broeke, M., Bus, C., Ettema, J. and Smeets, P.: Temperature thresholds for degree-day modelling of Greenland ice sheet melt rates, Geophysical Research Letters, 37(18), 2010.

Verjans, V., Leeson, A. A., Nemeth, C., Stevens, C. M., Kuipers Munneke, P., Noël, B., and van Wessem, J. M.: Bayesian calibration of firn densification models, The Cryosphere, 14, 3017–3032, https://doi.org/10.5194/tc-14- 3017-2020, 2020

---

## Author Comment (AC2)

*Response to RC2. The authors provide responses to the reviewer comments in italic serif typeface throughout whereas the reviewers comments remain in normal sans serif typeface.*

GENERAL

First of all, I would like to apologize for the time it has taken for me to provide this review. This was an oversight on my part and in no way reflects a lack of interest in this paper or the subject.

This manuscript by Medley et al. discusses a new modeling framework that uses a combination of the MERRA-2 reanalysis and the Community Firn Model to arrive at estimates of SMB and its individual components over the Greenland and Antarctic Ice Sheets for the period 1980-2019.

It is an important and welcome contribution to the existing regional climate models and firn models available for estimating ice-sheet mass balance by means of altimetry or the input-output method. It is encouraging that more of these models become available, and MERRA-2 combined with CFM is presented as a comprehensive modeling framework.

There are three main concerns with this manuscript, which I hope can be addressed by the authors in a thorough revision. But I am sorry to say that this probably entails a substantial effort. In fact, it would have been better from the start to divide this work into a paper that evaluates MERRA-2 over ice sheets, and a paper that deals with the CFM for elevation change and runoff estimates.

*We thank the reviewer for taking the time to present a thorough review of our submitted manuscript. We appreciate that the reviewer sees value in our work, and we have worked to satisfy all the reviewer's concerns to improve the quality of the research and manuscript. We think the reviewer's insight has largely improved the manuscript, and we thank them accordingly.*

1) Evaluation of MERRA-2 against independent observations.

In a global sense, MERRA-2 has been evaluated in a large paper by Gelaro et al. (2017). However to my knowledge, the performance of MERRA-2 over Greenland and Antarctica has not been assessed against observations, like it has been done extensively for e.g. MAR (Fettweis et al., 2013, Agosta et al., 2019), HIRHAM5 (Mottram et al., 2017), RACMO (Van Wessem et al., 2018, Noël et al., 2018) or CESM for climate variables ánd individual SMB components (temperature, snowfall, surface melt, albedo, cloud cover, firn temperature, runoff, etc.). This paper does present SMB and its components but they are not verified, evaluated, or compared against existing estimates. This is worrisome, since MERRA-2 serves as the input for CFM which computes the surface elevation changes needed to correct for ICESat and other altimeters. Thus, the authors will have to make sure that the reader can be confident about the MERRA-2 output over the ice sheets.

*The reviewer is entirely correct that we must ensure the quality of the forcing data to understand its strengths and limitations in representing polar climate conditions. As the reviewer points out, a full analysis of multiple variables over both ice sheets is an extensive undertaking; thus, we present SMB analysis of MERRA-2, but also state that MERRA-2 has been extensively evaluated over the ice sheets and have added additional citations which should provide the reader more confidence in the use of MERRA-2. Input for the CFM is net accumulation, melt, rain, and temperature. We provide an evaluation of SMB as best we can replicate recent SMB model intercomparison exercises for each ice sheet presented in this journal (i.e., Mottram et al., 2021 and Fettweis et al., 2020). We provide citations*

*regarding temperature evaluations from MERRA-2 that show its capability in providing realistic air temperatures for both ice sheets.*

2) Uncertainty analysis

The paper lacks uncertainty analyses. Uncertainty intervals of the SMB components are a standard deviation of the sample of annual values. But there is no estimate of the uncertainty of annual values, established for example by independent evaluation against observations (see 1). For the volume and height changes, no uncertainty estimates are provided. Most effort in presenting numbers is not in the numbers themselves, but in quantifying the confidence that we put in them. This is also crucial for establishing an uncertainty estimate of mass loss determined by altimetry. A robust uncertainty analysis must be added to the paper.

Quite a number of processes could be identified that give an uncertainty. Regarding the surface elevations changes from CFM, probably the most important ones are the uncertainty in SMB, snowfall, surface melt, and the assumption that the RCI period is representative for the long-term climate (see comments by reviewer 1).

Adding an uncertainty estimate allows for a fair judgement of the numbers presented in this paper, and for an comparison with existing literature (see also point 3), like for example the volume change estimates from Kuipers Munneke et al., which at first sight appear to be larger over Greenland than from MERRA-2.

*The reviewer brings up an excellent point regarding the importance of also providing uncertainties. We have added a section detailing our uncertainty analysis, which includes insights into the comparisons with observations for SMB (and how it translates into the model uncertainty). We have also generated perturbation ensemble CFM runs in which both climate variables and firn model parameters as perturbed at several locations across the ice sheet to evaluate the impact on FAC and thickness change through time. These uncertainties provide the reader realistic bounds on our knowledge of the firn evolution stemming from uncertainties in both the forcing, firn model parameters, as well as choice of RCI, which as the reviewer rightly points out helps with intercomparison of two models.*

3) Embedding in existing literature

While I believe that MERRA-2/CFM is a very valuable addition to the suite of models that are currently available to estimate SMB and firn changes over the ice sheets, the literature from those existing models is largely ignored in this paper. It would be insightful if the numbers in this study are compared to existing estimates of SMB and firn volume change (MAR, RACMO, CESM, HIRHAM, ...). Also it would be fair to refer to previous work into, e.g., quantifying seasonal cycles of firn processes (Ligtenberg et al., 2012), or volume loss over Greenland.

*We appreciate the importance of model intercomparison, which is why we compared our results to those from two recent SMB model intercomparison exercises to show that our SMB estimates fall within the range of several different models (rather than comparing to individual models; Lines 490-499). While an intercomparison with other models is an interesting analysis, it falls outside the scope of this paper, which is to present the new results from MERRA-2 and CFM. Future work will involve a comparison of volume change products.*

*Not referencing the work of Ligtenberg et al., 2012 was an oversight on our end, and we have now included it in our paper. If the reviewer could be more specific regarding omitted literature on "volume loss over Greenland," we would greatly appreciate it. Because this paper is not generating model simulations of SMB, but rather using it from an existing global model, we didn't extensively compare to existing papers that describe various SMB model updates and improvements. The focus of the work is rather on the CFM modelling, which is new. We do, however, add references to some of the recent SMB model work as suggested by the reviewer to point readers to additional resources if they are interested. We believe that the additions will better contextualize the existing work, and how this effort builds on an existing strong research community.*

SPECIFIC COMMENTS

Page 1 line 10: suggest to include the time frame of the simulations in the abstract: "new simulations of firn processes (1980-2019)"

*Done.*

Page 5 line 126: how is temperature initialized for locations with SMB<0?

*The initial ice column is isothermal, set at the mean annual temperature. We have clarified the sentence.*

Page 7 line 191: why was the functional form of equations (11) and (12) chosen to be an Arrhenius-style rate equation? Is there a reason to suppose that the misfit between modeled and observed densification rates should take this form?

*This was based off the functional form of the densification equations 9-10. Densification follows an Arrhenius type dependence on temperature whereas the overburden or accumulation rate is assumed to be linearly dependent (exponent equal to one). We took on the functional form from the source equation rather than adding our own expectation of the functional form because it makes interpretation of the impact on the densification parameters more straightforward rather than adding a new form to the calibration on top. We have discussed this choice and its implications more without the text. We explored a linear form on the calibration, but the model fit was worse than using the functional form of the densification equations.*

Page 7 line 211: it is interesting to see that $E_0$ is found to be zero. It implies that the correction factor $R_0$ reduces to $b^{\{beta_0\}}$, i.e. with no dependence on temperature. For the reader, it would be interesting to show this result rather than the current figure 4. The present figure 4 is of limited interest, since the correction procedures forces the data points to the 1:1 line by design. It would be more interesting to show the validity and behavior as an Arrhenius plot:

$R_0 = b^{(beta_0)} exp(-E0/RT)$

$ln(R_0) = beta_0 \, ln(b) - E_0 \, (1/RT)$

With $E_0 = 0$, plotting $ln(R_0)$ against $ln(b)$ would show $beta_0$ as the gradient and 0 as the intercept of a linear plot.

For $R_1$ it is more difficult. A normal Arrhenius plot would look like

$ln(R_1) = (-E_1/R) * (1/T) + beta_1 * ln(b)$

which is a linear y = mx + d with 1/T as x and beta_1*ln(b) as a constant. However, the latter is not constant but depends on b. Perhaps it is possible to select a subset of ~20 firn cores with a narrow range in b so that a constant value for (beta_1*ln(b) can be assumed, and ln(R_1) can be plotted against 1/T as a linear function.

*Thank you for the suggestion to expand more on how the calibration procedure impacts the original densification rates. We plot R_0 and R_1 as a contour plot with temperatures on the x-axis and accumulation rates on the y-axis, and colors represent the calibration coefficients. The reviewer is indeed correct that there for the first stage of densification, E_0 = 0 implies the calibration does not have any dependence on the temperature. For Stage two, there is temperature dependence. We also note a recent paper that independently came to near similar results using a different technique (Verjans et al., 2020), which we reference now as well.*

Page 9 line 270: I can understand that you have looked only at the significantly related predictors, and not at their physical interpretation. But I think that mean northward wind speed is an awkward predictor especially in the Greenland context. Could the performance of the model be retained while dropping V_0?

*Yes. Use of northward wind only was an oversight (and something that the other reviewer noted as well), so we no longer use northward wind in our surface density calculation.*

Page 10 line 297: I am struggling with the fact that no meltwater-related output from MERRA-2 was used in this manuscript, only snowfall. Is MERRA-2 not designed to provide this output? Is it of insufficient quality? Was it forgotten? Regarding runoff, was the runoff output from MERRA-2 used, or from the firn model? This remains unclear. In any case, is the runoff from the firn model comparable and similar to the runoff from MERRA-2?

*MERRA-2 simulates liquid water processes, including melt flux, infiltration, refreezing, and runoff; however, actual melt flux variables were not saved within a data collection. Only runoff is available. We use the runoff provided by the firn model because the snowpack physics used over the ice sheets was derived for mountain snowpack environments, not polar ice sheet conditions and have made that clear in Section 2.2, and we also make it clear when we discuss how SMB is calculated in Section 3.2 (i.e., we specifically mention that runoff from the CFM simulations is used). We provide a comparison of runoff derived from the CFM with that from MERRA-2 as well.*

Page 10 line 300: The degree-day modeling approach is confusing, and quite rough. The approach by Van den Broeke et al. (2010) was devised only in cases where daily mean temperatures temperatures are available, to compensate for the lack of representation of the daily cycle. I have tried to understand why such low T_0 are required to get a good match at an annual scale. I guess this is only possible if MERRA-2 has a strong cold bias, or a very poor representation of the daily temperature cycle, or if the degree-day approach actually fails over the entire range of melt-temperature combinations on both ice sheets. I would like the authors to comment on this. More generally, I wonder if surface melt in MERRA-2 is poorly represented, or if it was forgotten or otherwise impossible to obtain this output variable.

*We have reworded much of the section on degree-day modeling to help make clear how the actual melt rates were calculated. There is unfortunately no way to recover surface melt from MERRA-2, which is why we use the degree-day approach. We are completely transparent that this is a major weakness of the work, especially when considering the Greenland Ice Sheet. We instead use the degree-day approach as a best approximation. We agree that we should have more discussion into the implications for such an*

*approach, as well as the derivation, and we also compare plots of the derived melt with the training datasets. It's important to note that the reasons could be many in number why the numbers needed are less than 273K, including either MERRA-2 model deficiencies (as the reviewer points out) but also MAR model deficiencies as well. We have added context and refined the Section 2.2.1 on degree-day modeling, including reiteration of the model deficiencies and its role on firn volume change.*

Page 13 line 382: a map showing the absolute change in FAC would be interesting here.

*We have added a map of FAC change.*

Page 14, section 3.2: To my opinion, it is important to emphasize that the surface melt estimates from Greenland and Antarctica are not independent numbers based on the outcomes of a physical model. Rather, these are the results of a degree-day model calibrated to observations (Antarctica) and another model (MAR, Greenland). Especially for Antarctica, it is presented (lines 427 and further) as if the melt estimates are an independent result of MERRA-2 and CFM, whereas in reality the degree-day method is tuned to reproduce the numbers by Trusel et al. over a part of the RCI. Page 15,

*We appreciate the concern of the reviewer. We have made note that only snowfall, evaporation, and rainfall come from MERRA-2, melt is forced to match output from other work, and runoff is derived via the CFM using that calibrated melt model.*

line 444: RFI -> RCI (?)

*Yes. We have corrected this typo.*

===

Agosta et al., 2019. Estimation of the Antarctic surface mass balance using the climate model MAR (1979-2015) and identification of dominant processes, The Cryosphere

Fettweis et al., 2013. Estimating the Greenland ice sheet surface mass balance contribution to future sea level rise using the regional atmospheric climate model MAR, The Cryosphere

Gelaro et al., 2017. The modern-era retrospective analysis for research and applications , version 2 (MERRA-2), Journal of Climate

Kuipers Munneke et al., 2015. Elevation change of the Greenland Ice Sheet due to surface mass balance and firn processes, 1960–2014, The Cryosphere

Ligtenberg et al., 2012. Quantifying the breathing of the Antarctic Ice Sheet, GRL

Mottram et al., 2017. Surface mass balance of the Greenland icesheet in the regional climate model HIRHAM5 : Present state and future prospects. Low Temperature Science. Series A. Physical Science.

Noël et al., 2018. Modelling the climate and surface mass balance of polar ice sheets using RACMO2 - Part 1 : Greenland (1958-2016), The Cryosphere

Trusel et al., 2013. Satellite-based estimates of Antarctic surface meltwater fluxes, GRL

Van den Broeke et al., 2010. Temperature thresholds for degree-day modelling of Greenland ice sheet melt rates, GRL

Van Wessem et al., 2018. Modelling the climate and surface mass balance of polar ice sheets using RACMO2, part 2: Antarctica (1979-2016), The Cryosphere

---

## Referee Report (RR1)

Review of Medley et al. by Vincent Verjans.

The study of Medley et al. estimates Surface Mass Balance (SMB) and Firn Air Content (FAC) evolution over the time period 1980-2021, on both the Greenland and Antarctic ice sheets (GrIS and AIS). This study represents a comprehensive modeling effort to derive model estimates of elevation changes associated to surface processes. The authors have thoroughly reworked the manuscript since its first version. This includes revision of some modeling aspects (use of effective temperature, degree day modeling, surface density parameterization), uncertainty analysis, extensive comparison of SMB against state-of-the-art studies, and improved discussion of limitations. I believe that the authors have appropriately addressed my comments from the first round of revisions, as well as those of reviewer 2. This review is separated in minor comments and technical comments. My minor comments address some aspects that should be clarified, or about which I raise some secondary reservations with respect to the approach and/or judgment of the authors. My technical comments only address structural and language aspects of the manuscript. I sincerely appreciate the thorough work of the authors to address all the comments from the first round of reviews. Provided some minor issues in the updated manuscript are addressed by the authors, I encourage the publication of this study in The Cryosphere.

Minor comments
1) The interpretation of the Reference Climate Interval (RCI)
1.a)
The authors define the RCI as "*ideally representative of long-term steady-state conditions*". I believe that this interpretation is slightly wrong. The RCI is used to develop the initial model firn column, from which transient experiments over the period of interest (1980-2021) start. As such, ideally, the initialization should be computed with the true climate forcing of the decades and centuries preceding 1980. This is true regardless of whether the long-term conditions were in steady-state (i.e., without trends) or not. In other words, the "perfect" RCI should not represent steady-state conditions if the true conditions were not in steady-state prior to 1980.
As the authors rightly point out:
"*we only have a spatiotemporally complete understanding of polar climate conditions arguably since the beginning of the satellite era (1979 and onwards). Thus, we make assumptions regarding how that firn column will respond to modern conditions (…)*"
In light of this incomplete knowledge, using steady-state conditions over the RCI is only the most reasonable assumption possible to make, but not a necessary condition for a valid firn model initialization procedure. For this reason, I believe that the authors should revisit their discussion of the RCI and of the appropriateness of their assumptions in Sections 2.1.2, 3.2.1, and 3.2.2. It should be clearer that steady-state conditions over the RCI are used in order to isolate effects of climatic deviations from the RCI on firn column changes. But such steady-state conditions are not representative of true conditions, and the true changes in firn thickness are influenced by the unsteady nature of past climate conditions. This is already partly explained in the Discussion section, but I find that earlier statements in the manuscript are misleading. Furthermore, the Discussion section only discusses past trends on AIS, whereas the existence of past trends is also true for GrIS (e.g., Hanna et al., 2011), which should also be mentioned.
1.b)
In the Discussion section:
"*we expect our results as a lower bound for trends, and future work investigating the impact of these reconstructed trends would help to quantify the resulting uncertainty in height changes due to long-term climate change*"
Either I misunderstand this statement, or I respectfully disagree with the authors about the lower bound for the trends. The presence of spatial trends in past climate does not mean that assuming a steady-state RCI necessarily causes an underestimation of recent trends in firn thickness change. That is because recent trends are relative changes compared to firn column dynamics during the RCI. For example, if the RCI overestimates SMB in past climate with respect to reality, a decrease in SMB in the recent past

would cause a stronger surface lowering than what happened in reality. That is because the relative decrease in SMB in the simulations would be larger than the relative decrease in SMB in reality. Thus, computed trends are not necessarily a lower bound. Instead, an RCI not representative of the true past climate can result in both an over- or under-estimation of current trends, depending on the particular biases of the RCI assumptions (which vary in space, and are different for different variables).

2) The uncertainty analysis
In general, I appreciate the effort to perform the uncertainty analysis in this revised version of the manuscript. The method is robust, and the uncertainty values add a lot of value to the Figures 13, 16, and 18. I would like however to bring three minor points to the attention of the authors.
2.a)
It is not entirely clear to me how the authors sampled the values of the variables in their uncertainty analysis. They state that:
"*we sampled the 2-sigma Gaussian distribution error in the modelled initial density (rho_0) and the calibration parameters (alpha_0, alpha_1, E_c0, E_c1) and perturb the CFM parameters*" (Section 2.5.1)
"*The perturbation developed sample randomly within the 2-sigma bounds of the Gaussian Perturbations and from a small number of Random Perturbations.*" (Table 1)
Does that mean that (i) they sample from a Gaussian distribution or (ii) from a Uniform distribution bounded by the 2-sigma bounds of the uncertainty?
I guess that they did (i), but in that case I do not understand why they limit the sampling to the 2-sigma bounds and thus exclude outliers from the uncertainty analysis. Alternatively, I may be misunderstanding the method. I recommend that the authors clarify and justify the method.
2.b)
Uncertainty in the parameters, and even more in the climatic variables, are certainly correlated. For example, SMB perturbations are, in reality, negatively correlated with perturbations in temperature. Similarly, the model parameters are not independent of each other. I understand that the authors decided not to constrain the appropriate correlation values, and their approach is sufficient in my view. However, I ask the authors to mention that there is some dependence between the perturbed variables, and that this is not accounted for in the uncertainty analysis.
2.c)
In Figure 12, there are 55 points plotted, whereas Section 2.5.1 mentions 45 sites for AIS and 18 sites for GrIS (i.e., total of 63 sites). Why is there a discrepancy of 8 sites?

Technical comments
In general, there are a lot of places where commas should be added for better clarity. I identify some of them in my Technical comments, but it would be good to pay attention to missing commas when the authors re-read the manuscript.
p.1 l.19-20
Change "*associated with surface mass balance*" to "associated with mass fluxes from surface processes". I want the readers to keep in mind that firn air content fluctuations themselves are largely governed by surface mass balance fluctuations.
p.2 l.44
Add "constant": "becomes approximately constant (917 kg m-3)".
p.2 l.62
Add comma: "changes, yet".
p.4 Eq. (6)
Remove brackets from the numerator.
p.4 l.111
Typo: "*GSDC-FDMv1.2*".
p.5 l.135-136

"*This depth is divided by a burial rate (snowfall – sublimation – melt) to estimate the time needed to refresh the firn column for a given site*."
I believe that "*This depth*" should be replaced by "The cumulative mass until this depth".
p.6 l.157
Remove "*that*".
p.6 l.161
I suggest replacing "*total mass above*" by "cumulative accumulation above" because b does not account for mass removal via runoff.
p.6 l.169
Add "is": "and is based".
p.6 l.179
Add "the": "on the use".
p.7 l.183-185
The iterative 3-sigma edit method is not clear to me. "*removing individual density measurements with residuals to the linear model larger than 3-sigma*": Larger than 3-sigma of what? Is "*sigma*" here the root mean square deviation of all the individual density measurements with respect to the linear model? In this case, it should be clarified that "*sigma*" refers to the root mean square deviation.
p.7 l.195
Add a space: "(2010) model".
p.8 l.232
Add commas: "parameters, when plugged into Eqns. 11–12, provide".
p.8 l.235
Add comma: "will increase, while".
p.8 l.242:
Replace "*expectation*" by "assumption".
p.8 l.244
Add comma: "with increasing depth, T".
p.9 l.246:
Replace "*expectation*" by "assumption".
p.9 l.255-256
I am not sure that this sentence is correctly phrased.
p.10 l.280
Add commas: "that, on average, we".
p.10 l.281
Plural: "deviations".
p.10 l.289-290
Give RMSE of fit of rho_0 for all rho_0 values as well as for rho_0 values below 330 kg/m^3.
p.11 l.311
"*GEOS*" is not defined.
p.11 l.315
Change "*span*" to "spans".
p.12 l.347
"*normalized distance*": normalized to what? To the mean r^2 and RMSE of the grid cell?
p.12 l.350
Add "an": "an observation-based calibration data set".
p.12 l.360
Add "in": "more in Sect. 4.".
p.12 l.365-366
Add a comma: "0.13 kg m^-2 hr^-1 K^-1, while calibrated values".
p.13 l.369
Change "*complicates*" to "complicate".

p.13 l.372
Why set the DDFs lower than the lower bound of ice shelves to 0 and not to the lower bound itself?
p.13 l.374
Provide also the range of DDFs in Antarctica. And maybe give a brief comparison with the Greenland values.
p.13 l.382-383
Rephrasing needed.
p.13 l.385
Change "*reduced the mismatch*" to "caused a larger mismatch". Or clarify the sentence.
p.13 l.388-391
Yes, but mention that the calibrated firn model is nevertheless used in areas with meltwater percolation.
p.13 l.397
Add space: "For v1 and".
p.14 l.406
Change "*error analysis*" to "uncertainty analysis".
p.14 l.420
I recommend replacing "*densification rates are reduced under increasingly high accumulations*" by "the sensitivity of densification rates to increasing accumulation is reduced".
p.14 l.420
Change "*dramatic*" to "pronounced".
p.14 l.421
Add comma: "is increased, especially".
p.15 l.441:
Change "*or*" to "i.e.,".
p.15 l.446
Add "of variables": "time series of variables of critical importance".
p.15 l.454
Add "the": "Each of the perturbations".
p.15 l.455
Add a comma: "Gaussian distribution, except".
p.15 l.458-459
Snow accumulation should be defined as Sn-Ev for consistency with Eq. (5).
p.16 l.483
Make sure to use the same tense for the verbs.
p.16 l.495
Refer to Section 2.5.2 when introducing the observations of SMB.
p. 17 l.500
Add hyphen: "SMB-induced".
p.17 l.516-524
How did the number of observations reduce from 16427 to 1037? Is that because many observations fall within the same grid cell? And/or are there observations excluded because they do not span a long enough time period? Or for another reason?
p.17 l.530
Rephrasing needed.
p.17 l.531
What is meant by "*accumulated*"?
p.18 l.533
"*First, we determined the mean GSFC SMB over the exact observation interval.*"
This comparison is never analyzed or discussed in the remainder of the manuscript. I recommend removing it for the sake of clarity.
p.18 l.552

Add "on average": "the AIS firn column contains, on average, more air than the GrIS.".
p.18 l.560
Specify that runoff is an output of the CFM, and not of MERRA-2.
p.19 l.564
"*the firn column accommodated 40% of all liquid water*": give +/- annual variability of the percentage.
p.19 l.569
"*The RCI is ideally representative of long-term steady-state conditions*": see my Minor comment 1 about the "*steady-state*" aspect.
p.19 l.574
Change "*most likely*" to "significantly".
p.19 l.575
Change "*our choice of RCI (1980–1995) should not generate non-physical transients in our firn simulations*" to "our choice of RCI (1980–1995) should not generate transients associated with the initialization process in our simulations."
p.19 l.579
"*The firn column only accommodated 38% of liquid water*": give +/- annual variability of the percentage. Also, please specify if the decrease from 40% to 38% is statistically significant.
p.19 l.580
"*The ablation zone grew in area by 30%*": please specify how this was computed. Does this number come from a fitted trend on the area with SMB<0 over the annual time series?
p.19 l.588
"t*he firn only accommodated ~19% of all liquid water*": give +/- annual variability of the percentage.
p.20 l.594
"*majority (94%)*": give +/- annual variability of the percentage.
p.20 l.601-602
Change "*the choice of RCI is justified*" to "the firn column initialized over the RCI spin-up should be in equilibrium with steady climate conditions".
p.20 l.607
Typo: "*and*" should be "an".
p.20 l.613
Change "*cycles*" to "amplitudes".
p.20 l.619
Change "*skewed*" to "driven".
p.21 l.623
"*(86.3 ± 13.6 km^3 yr^-1 )*": what do these value refer to?
p. 21 l.626
Change "*component*" to "amplitude".
p. 21 l.646
Add "mean": "than the ensemble mean".
p.21 l.647
Add "observations with": "for observations with SMB > ~ -2".
p.22 l.654-655
Also give the value of the mean absolute relative bias for the sake of information.
p.22 l.659
Add "integrated": "suggests that integrated over the entire ice sheet".
p.22 l.675
Typo: "*GSFC the*".
p.23 l.688-690
Provide the results of the ensemble mean of Mottram et al. (2021) in Table 3. Otherwise, the reader cannot evaluate the comparison in a quantitative manner without going to the publication that is referenced.

p.23 l.695-697
Also give the value of the mean absolute relative bias for the sake of information.
p.23 l.704
"*exceeded by two models within the ensemble*": two of how many?
p.23 l.705
Change "*within*" to "in".
p.23 l.708
Add "compare": "We also compare".
p.24. l.725
Change "*largely in response to the overburden*" to "largely due to reduced sensitivity to increasing overburden".
p.24 l.727-728
"*Our calibration differs*": note that Verjans et al. (2020) took a similar approach.
p.24 l.742
Add comma: "firn densification model, which models".
p.24 l.766
Change "*recent past*" to "past prevailing conditions".
p.25 l.775
Add hyphen: "physically-based".
p.25 l.776-778
I believe that the most likely cause of the lower melt values is that the calibrated DDFs are capped at higher elevations after the calibration process. This should also be mentioned here.
p.25 l.782
Typo: "*heigh-elevation pats*".
p.25 l.782
Add comma: ", particularly".
p.26 l.785-786
Add an extra reservation by changing: "*to fully evaluate this improvement and highlight other potential future improvements.*" to "to fully evaluate this improvement, rule out possible compensating errors, and highlight other potential future improvements.".
p.26 l.800
Change "*e.g., fresh snowfall*" to "i.e., fresh snowfall".
p.26 l.802-803
"*we want to separate the climate model impact (SMB) from the firn model impact (FAC)*": I respectfully disagree with this statement. Firstly, air changes due to snowfall are governed by the climate model, not by the firn model. Secondly, the firn model takes as inputs the fields from the climate model, thus the effects from both models cannot be separated entirely.
p.26 l.803
"*While the SMB and FAC contributions to total firn volume change over multiannual time scales are somewhat comparable,*": I believe that this does not give an appropriate picture of the results given in Sections 3.3.1 and 3.3.2. I think that it would be better to add one or two extra sentences to quantify the different impacts of SMB and FAC on long-term volume changes on the GrIS and AIS.
p.27 l.824-825
Typo: "*within firn column*".
p.27 l.826
Add "in-situ": "measuring firn processes in-situ".
p.33 l.838-839
I believe that the authors calculate a series of 141 RMSE values for stage 1 and 76 RMSE values for stage 2, from which they calculate the lower quartile, median, and upper quartile RMSE. Similarly for r^2. Is that correct? If so, I recommend adding a statement such as: "For each observation, we use all the point measurements of density in depth to calculate the corresponding RMSE and r^2."

p.33 l.839:
Typo: "*observation*" should be plural.
p.33 Table B1
Change the column "*Density Profile*" to "Linear Density Profile".
p.34 l.847-848
"*using the daily MERRA-2 fields*": this contradicts the statement above "*native 1-hour resolution*". Are the fields from MERRA-2 hourly or daily?
p.34 l.848
Change "*label*" to "labeled".
p.34 l.854
Specify that n is taken equal to 5.
Figure 1
Explain the difference between red and black circles. Also, most red crosses are not clearly visible.
Figure 2
At first, it was not intuitive to me why the contour plot has the same color code as the closed circles. I recommend adding a statement such as: "The background contours represent the best fit to the coefficients from the calibration sites.".
Figure 4
"*those in grey reference the open circles*": I don't think that "reference" can be used as a verb.
Figure 5
I recommend plotting these figures as differences between MERRA-2 and M2R12K instead.
Figure 6
I recommend plotting these figures as differences between MERRA-2 and M2R12K instead.
Figure 7
"*as it maximizes the distance between the two curve*": that depends on the y-axis scales that are chosen for the respective curves. Instead, this should be reformulated as minimizing the normalized distance (and see my comment for p.12 l.347). Note also that "*curve*" should be plural.
Figure 11
Is it possible to show these plots as box plots? The information conveyed would be similar, but that would also allow the reader to see the outliers.
Table 1
Rephrasing of the caption is needed. And reference to Calonne (2019) should be to Calonne et al. (2019), and it is not given in the references.
Table 3 (l.1389)
Typo: "*difference*" should be "different".

References used in this review:
Hanna, E., Huybrechts, P., Cappelen, J., Steffen, K., Bales, R. C., Burgess, E., McConnell, J. R., Peder Steffensen, J., Van den Broeke, M., Wake, L., Bigg, G., Griffiths, M., and Savas, D.: Greenland Ice Sheet surface mass balance 1870 to 2010 based on Twentieth Century Reanalysis, and links with global climate forcing, J. Geophys. Res.- Atmos., 116, D24121, doi:10.1029/2011JD016387, 2011.

Verjans, V., Leeson, A. A., Nemeth, C., Stevens, C. M., Kuipers Munneke, P., Noël, B., and van Wessem, J. M.: Bayesian calibration of firn densification models, The Cryosphere, 14, 3017–3032, https://doi.org/10.5194/tc-14- 3017-2020, 2020.

---

## Author Response (AR2)

TC-2020-266 | Research article
Submitted on 12 Sep 2020

**Simulations of Firn Processes over the Greenland and Antarctic Ice Sheets: 1980–2021**

Brooke Medley, Thomas A. Neumann, H. Jay Zwally, Benjamin E. Smith, and C. Max Stevens

**Handling editor**: Nicolas Jourdain, nicolas.jourdain@univ-grenoble-alpes.fr

*We have made the effort to address most of the comments provided by Vincent Verjans, which has helped improve the quality of our manuscript. All changes to this version of the manuscript are in response to their comments as well as updating the numbers and figures to incorporate new model simulations with the CFM bug fixed. Thus, we have re-run our model with the bug fixed and have updated all the numbers accordingly. The bug was very minor, and therefore, the our results/discussion have not changed.*